# Aberrant phase separation and nucleolar dysfunction in rare genetic diseases

Martin A. Mensah[1,2,3,32], Henri Niskanen[4,32], Alexandre P. Magalhaes[4], Shaon Basu[4], Martin Kircher[5,6], Henrike L. Sczakiel[1,2,3], Alisa M. V. Reiter[1], Jonas Elsner[1], Peter Meinecke[7], Saskia Biskup[8], Brian H. Y. Chung[9], Gregor Dombrowsky[10,11], Christel Eckmann-Scholz[12], Marc Phillip Hitz[10,11], Alexander Hoischen[13,14], Paul-Martin Holterhus[15], Wiebke Hülsemann[16], Kimia Kahrizi[17], Vera M. Kalscheuer[3], Anita Kan[18], Mandy Krumbiegel[19], Ingo Kurth[20], Jonas Leubner[21], Ann Carolin Longardt[22], Jörg D. Moritz[23], Hossein Najmabadi[17], Karolina Skipalova[1], Lot Snijders Blok[14], Andreas Tzschach[24], Eberhard Wiedersberg[25], Martin Zenker[26], Carla Garcia-Cabau[27], René Buschow[28], Xavier Salvatella[27,29], Matthew L. Kraushar[4], Stefan Mundlos[1,2,3,30], Almuth Caliebe[6], Malte Spielmann[3,6,31,33✉], Denise Horn[1,33✉] & Denes Hnisz[4,33✉]

Thousands of genetic variants in protein-coding genes have been linked to disease. However, the functional impact of most variants is unknown as they occur within intrinsically disordered protein regions that have poorly defined functions[1–3]. Intrinsically disordered regions can mediate phase separation and the formation of biomolecular condensates, such as the nucleolus[4,5]. This suggests that mutations in disordered proteins may alter condensate properties and function[6–8]. Here we show that a subset of disease-associated variants in disordered regions alter phase separation, cause mispartitioning into the nucleolus and disrupt nucleolar function. We discover de novo frameshift variants in *HMGB1* that cause brachyphalangy, polydactyly and tibial aplasia syndrome, a rare complex malformation syndrome. The frameshifts replace the intrinsically disordered acidic tail of HMGB1 with an arginine-rich basic tail. The mutant tail alters HMGB1 phase separation, enhances its partitioning into the nucleolus and causes nucleolar dysfunction. We built a catalogue of more than 200,000 variants in disordered carboxy-terminal tails and identified more than 600 frameshifts that create arginine-rich basic tails in transcription factors and other proteins. For 12 out of the 13 disease-associated variants tested, the mutation enhanced partitioning into the nucleolus, and several variants altered rRNA biogenesis. These data identify the cause of a rare complex syndrome and suggest that a large number of genetic variants may dysregulate nucleoli and other biomolecular condensates in humans.

Monogenic and common diseases are frequently associated with mutations in transcriptional regulatory proteins, including DNA-binding transcription factors. However, the functional impact of the majority of such mutations is unknown, and many complex diseases still lack a clear underlying genetic component[1,9–12]. We initially set out to identify the molecular basis of brachyphalangy, polydactyly and tibial aplasia/hypoplasia syndrome (BPTAS; Online Mendelian Inheritance in Man database identifier: 609945), an extremely rare complex malformation syndrome with an as yet unknown molecular aetiology[13–19]. During the study, five individuals (I1–I5) were diagnosed with BPTAS. All five exhibited a distinct skeletal phenotype, including short and malformed lower limbs characterized by tibia aplasia or hypoplasia, preaxial polysyndactyly and contractures of large joints (Fig. 1a,b). In all five individuals, anomalies of the upper limbs were less severe compared with those of the lower limbs, and included brachydactyly or brachyphalangy of fingers with an irregular finger length (Fig. 1a,b).

Short radius and ulna and contractures or pterygia of the elbow joints were present in four out of five individuals. All individuals with BPTAS diagnosed during our study or described in previous reports also presented with distinct craniofacial, neurological and genitourinary features. Phenotypic findings are summarized in Supplementary Table 1. Detailed clinical and family histories are provided in Supplementary Note and Extended Data Fig. 1.

## De novo *HMGB1* frameshifts in BPTAS

We performed genome sequencing of I1 and detected a potentially pathogenic variant: the heterozygous frameshift NM_002128.7(*HMGB1*): c.556_559delGAAG;p.(Glu186Argfs*42) in the final exon of *HMGB1* (Extended Data. Fig. 2a). *HMGB1* encodes a highly conserved, low-specificity DNA binding factor associated with cell signalling[20], cell motility[21,22], base excision repair[23,24] and chromatin looping[25].

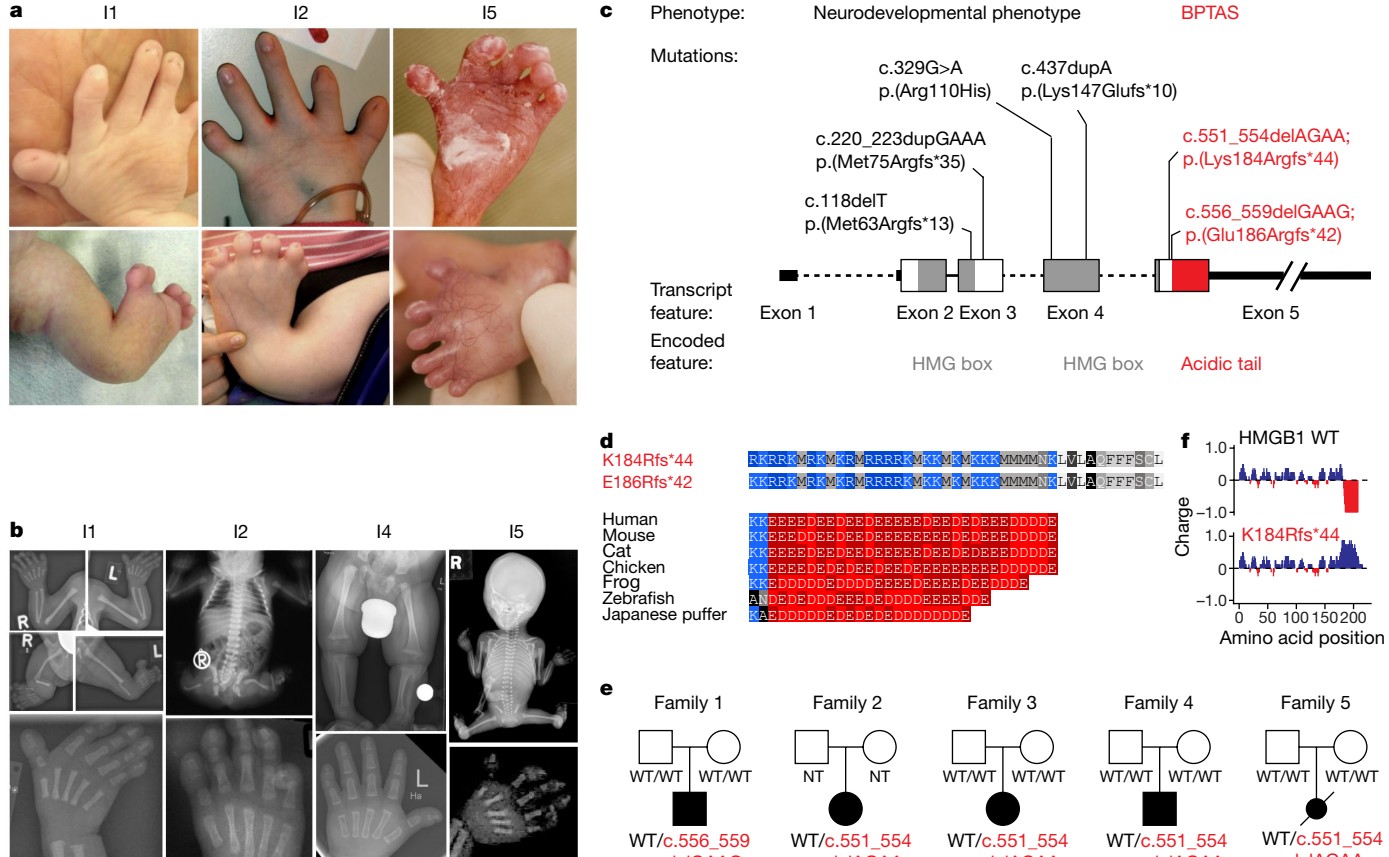

**Fig. 1 | De novo frameshifts in *HMGB1* cause BPTAS. a**, Photographs of individuals diagnosed with BPTAS. Top row, hands of I1, I2 and I5. Note brachydactyly, irregular finger length and hypoplasia of the nails. Bottom row, lower extremities of I1, I2 and I5, presenting with malformed legs, joint contractures, preaxial polysyndactyly and hypoplasia of the nails. **b**, Radiograms of I1, I2, I4 and I5. Top far left, limb radiograms (at newborn age) of I1 showing brachydactyly and brachyphalangy, tibial aplasia, hypoplastic fibulae and preaxial polysyndactyly. Top middle left, babygram of I2. Note tibial aplasia, hypoplastic and absent fibulae, hypoplastic pelvic bones and hypoplastic right femur. Top middle right, lower extremities of I4 (at 6 months) showing asymmetric shortness of tibiae and fibulae. Top far right, fetogram of I5 showing tibial aplasia, hypoplastic and absent fibulae, hypoplastic pelvic bones and contractures of joints. Bottom row, hand radiograms of I1, I2 (both at newborn age), I4 (at 6 months)

and of I5 (at 21 weeks of gestation). Note the short middle phalanges and short proximal phalanges of the thumbs. **c**, Pathogenic frameshift variants in the acidic tail of HMGB1 in the individuals with BPTAS reported in this article are highlighted in red. Previously reported variants associated with developmental delay are in black. Note the genotype–phenotype correlation: C-terminal frameshifts result in BPTAS, whereas other variants lead to a neurodevelopmental phenotype. **d**, Amino acid sequence of the C terminus of HMGB1 in individuals with BPTAS and in selected vertebrates. Acidic residues glutamate and aspartate are shaded in red, basic residues arginine and lysine are shaded in blue. Note the replacement of the conserved acidic tail in individuals with BPTAS. **e**, Family pedigrees. Individuals with BPTAS are highlighted with black boxes, and the genotypes are below the boxes. **f**, Charge plots of WT and mutant HMGB1. I, individual; L, left; NT, not tested; R, right; WT, wild type.

Sanger sequencing of I1 and his parents confirmed the presence of the frameshift variant and revealed de novo occurrence (Fig. 1c–e and Extended Data Fig. 2b). Sanger sequencing of *HMGB1* in I2 and I3, and trio exome sequencing of I4 identified a similar de novo heterozygous frameshift: NM_002128.7(*HMGB1*):c.551_554delAGAA;p.(Lys184Argfs*44), a variant also detected in a previously described female fetus[26] (I5) (Fig. 1c–e and Extended Data Fig. 2c,d). Sequencing of cDNA from a lymphoblastoid cell line derived from peripheral blood cells from I3 confirmed the presence of both wild-type and mutant *HMGB1* transcripts (Extended Data Fig. 2e). The two frameshift mutations result in almost identical, positively charged sequences (Fig. 1d,f).

## Altered HMGB1 phase separation in vitro

To investigate the potential pathogenic role of the frameshift variant in HMGB1, we first explored structural and sequence features of the wild-type and mutant proteins. HMGB1 is a low-specificity DNA-binding protein that contains two HMG boxes that are responsible for DNA binding[27] and a C-terminal acidic tail (Fig. 2a,b). The acidic tail is

predicted to be intrinsically disordered and resides within an approximately 60-amino-acid long conserved intrinsically disordered region (IDR), as revealed by AlphaFold2 and PONDR analyses (Fig. 2a,b and Extended Data Fig. 3a). Both algorithms predicted a slight propensity of the C-terminal portion of the IDR to assume a helical conformation in the frameshift mutant HMGB1 (Fig. 2a,b and Extended Data Fig. 3b). This prediction was confirmed by circular dichroism experiments on synthetic peptides that corresponded to the C-terminal 80–90 amino acid region (Extended Data Fig. 3c–e). IDRs of numerous proteins, including transcription factors, co-activators (for example, Mediator) and RNA polymerase II (RNAPII), contribute to phase separation by mediating multivalent low-affinity interactions[6,28–31]. Therefore, we hypothesized that the potentially BPTAS-causing frameshift may alter the phase-separation capacity of HMGB1.

To test the phase-separation capacity of HMGB1, we purified recombinant HMGB1 proteins tagged with enhanced green fluorescent protein (eGFP) and examined their behaviour in vitro. Wild-type full-length HMGB1 formed droplets in the presence of a crowding agent (10% polyethylene glycol (PEG)), and the number and size of droplets scaled with

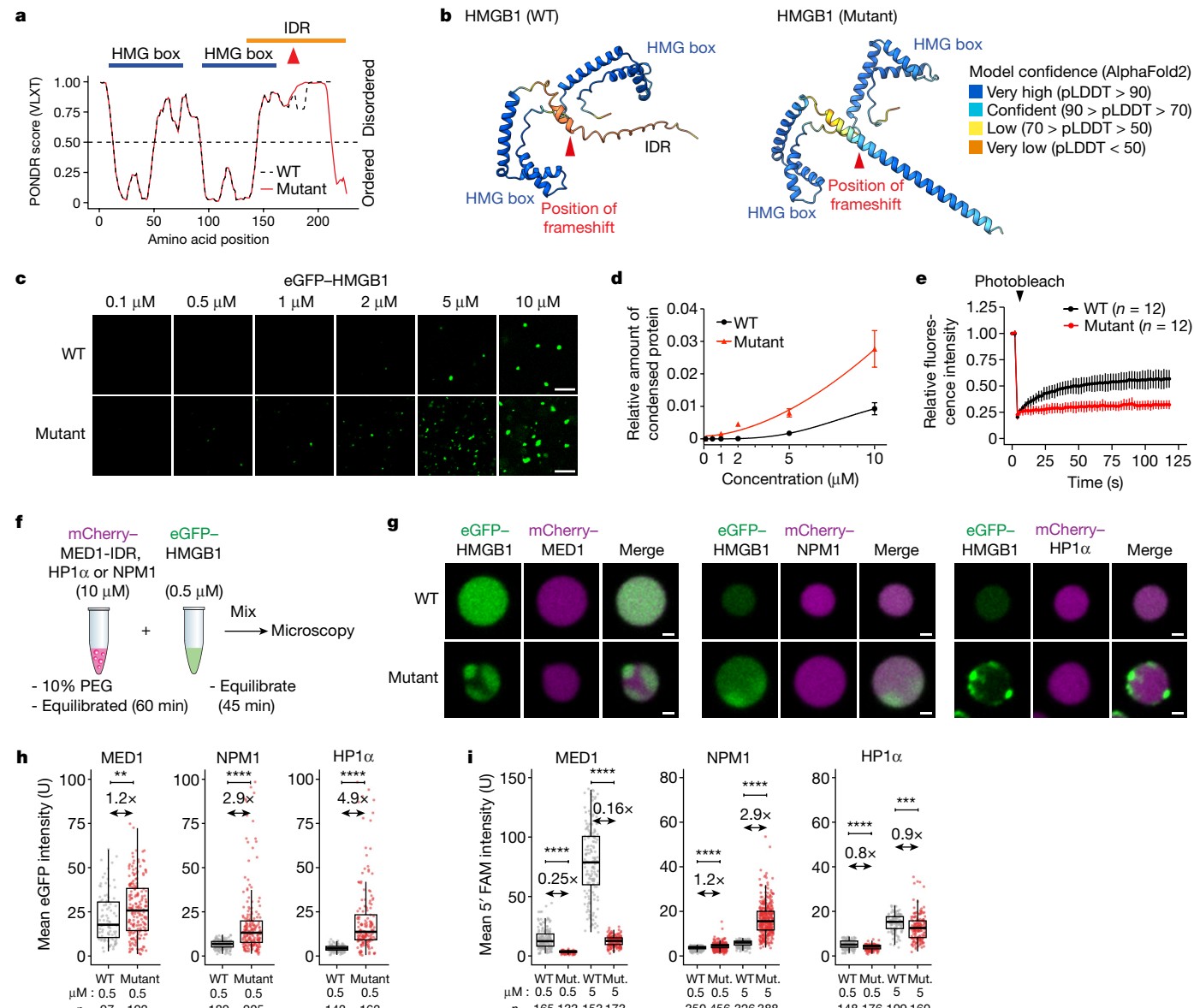

**Fig. 2 | A BPTAS-causing frameshift alters HMGB1 phase separation in vitro.**
**a**, Graph plotting the intrinsic disorder of HMGB1. Red arrowhead shows the
position of the BPTAS frameshift. The position of the IDR is highlighted with an
orange bar and the position of HMG boxes with blue bars. **b**, Structures of WT
and mutant HMGB1 predicted with AlphaFold2. Colours ranging from blue to
orange depict the per-residue measure of local confidence (pLDDT) for the model.
**c**, Representative images from droplet formation assays of eGFP–HMGB1
variants at the indicated concentrations. The experiment was repeated three
times, with similar results obtained. **d**, Quantification of the relative amount
of condensed protein at the indicated concentrations. Data displayed as the
mean ± s.d. **e**, Relative fluorescence intensity of the bleached area from
eGFP–HMGB1 condensates before and after photobleaching. Data displayed as
the mean ± s.d. **f**, Scheme of co-droplet assays. **g**, Representative images of
eGFP–HMGB1 proteins mixed with preassembled mCherry-labelled MED1-IDR,
HP1α or NPM1 droplets. **h,i**, Quantification of eGFP (**h**) and 5′ FAM (**i**) fluorescence
intensity in mCherry-labelled MED1-IDR, HP1α and NPM1 droplets mixed with
full-length mEGFP–HMGB1 proteins (**h**) or 5′ FAM–HMGB1-IDR peptides (**i**).
Fold change values between the mean intensities of WT and mutants (Mut.) are
indicated above the plot. Median is shown as a line within the boxplot, which
spans from the 25th to 75th percentiles. Whiskers depict a 1.5× interquartile
range. $P$ values are from two-tailed Welch's $t$-test. **$P < 1 \times 10^{-2}$, ***$P < 1 \times 10^{-3}$,
****$P < 1 \times 10^{-4}$. Scale bars, 5 μm (**c**) and 10 μm (**g**).

the concentration of the protein (Fig. 2c,d and Extended Data Fig. 4a–c).
The droplets were spherical, settled on the surface and occasionally
underwent fusion (Supplementary Video 1), which are hallmarks of phase
separation[32]. By contrast, the frameshift mutant HMGB1 formed amor-
phous condensates that appeared at a lower saturation concentration
(Fig. 2c,d) and, after photobleaching, recovered fluorescence slower than
wild-type HMGB1 droplets (Fig. 2e). Similar results were observed using
synthetic peptides that corresponded to the C-terminal 80–90 amino
acid region of wild-type and frameshift mutant HMGB1 (Extended Data
Fig. 4d–g). These results indicate that the frameshift in HMGB1 enhances
condensate formation and alters condensate properties in vitro.

Mammalian nuclei contain numerous biomolecular condensates,
for example, the nucleolus, heterochromatin, co-activator and RNAPII
condensates[4,5]. IDRs play important roles in the partitioning of pro-
teins into nuclear condensates[4,5]. We therefore tested whether the
frameshift mutation alters the partitioning of HMGB1 into nuclear
condensates. Using purified marker proteins, we assembled the fol-
lowing model condensates: recombinant mCherry-tagged MED1
IDR droplets as an in vitro model for Mediator co-activator conden-
sates[29,31,33]; mCherry-tagged HP1α droplets as an in vitro model for
heterochromatin[34]; and mCherry-tagged NPM1 droplets as an in vitro
model for the granular component of the nucleolus[35]. Wild-type and

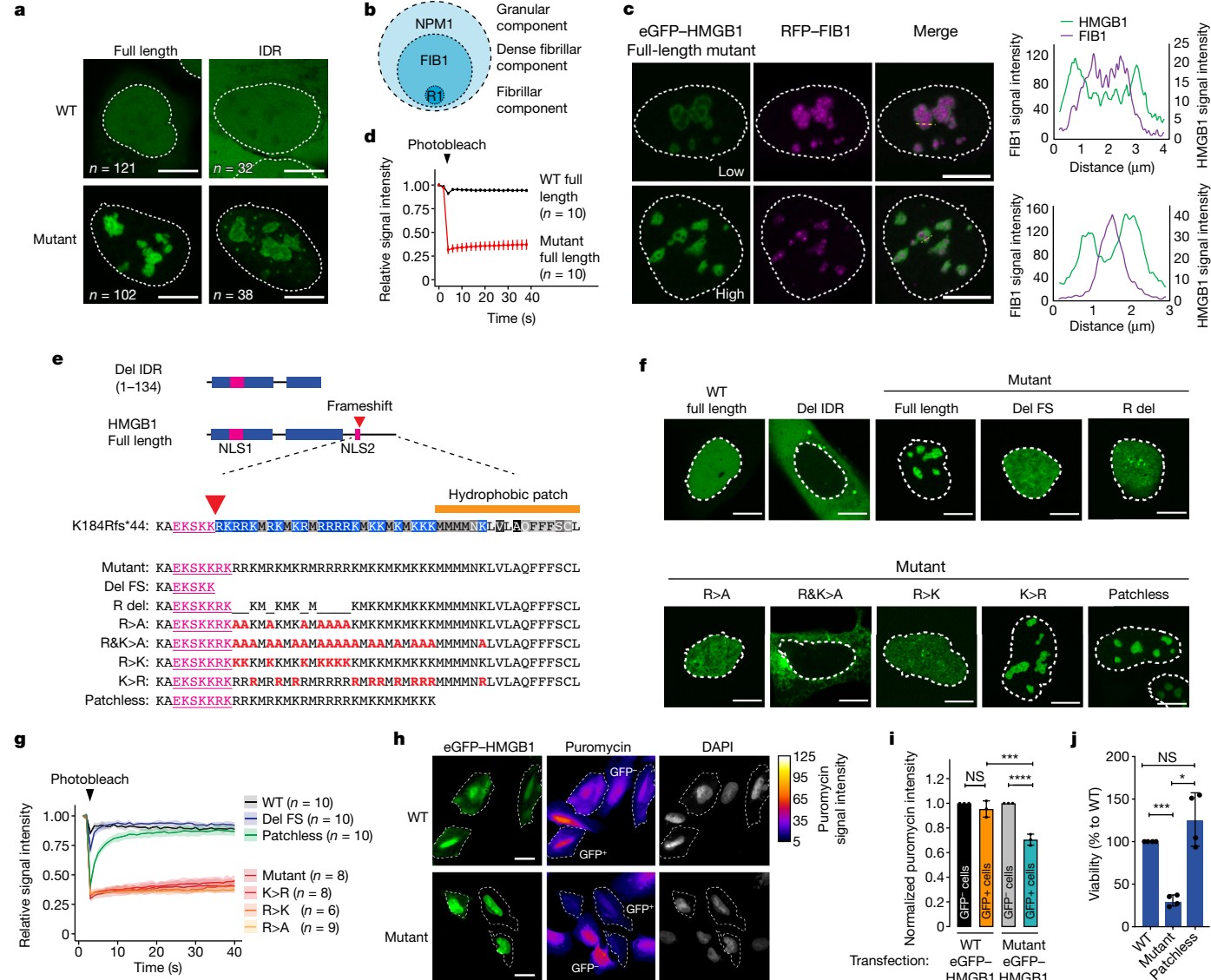

**Fig. 3 | Mutant HMGB1 replaces the granular component of the nucleolus in vivo. a**, Representative images of live U2OS cells expressing eGFP–HMGB1 proteins. Nuclear area revealed by Hoechst staining is shown as dashed white lines in **a**, **c** and **f**. **b**, Model of the nucleolus. R1, RNA polymerase I. **c**, Left, representative images of U2OS cells expressing RFP–FIB1 and mutant eGFP–HMGB1. Right, fluorescence intensity profiles from the region highlighted by the dashed yellow line. Low and high indicate nuclei with a relatively low or high amount, respectively, of the mutant protein. **d**, Relative fluorescence intensity of eGFP–HMGB1 before and after photobleaching. Data displayed as the mean ± s.d. **e**, Schematic and sequence representation of HMGB1 variants. Blue bars, HMG boxes. NLS, nuclear localization signal. Red arrow marks the position of the frameshift mutation (K184Rfs*44) and red letters highlight mutagenized amino acids. **f**, Representative images of live U2OS cells expressing

the indicated eGFP–HMGB1 variants. **g**, Relative fluorescence intensity of eGFP–HMGB1 variants before and after photobleaching. Data are displayed as a line for the mean signal, with the shaded region representing ± s.d., *n* = number of cells examined. **h**, Representative images from puromycin-staining experiments with U2OS cells ectopically expressing eGFP–HMGB1 proteins. The puromycin signal was used to trace the cell area to highlight GFP⁺ cells with a dashed line. **i**, Normalized puromycin intensities displayed as the mean ± s.d. from three independent biological replicate experiments. ***$P$ < 0.0002, ****$P$ < 0.0001 by one-way ANOVA. **j**, Quantification of the viability of cells expressing the indicated HMGB1 proteins. Data displayed as individual points from independent biological replicates (*n* = 4). Bar charts show mean ± s.d. *** $P$ = 0.0005, * $P$ = 0.0177 by one-way ANOVA. Scale bars, 10 μm (**a**,**c**,**f**) or 20 μm (**h**).

mutant HMGB1 proteins were then added to the droplets (Fig. 2f). Wild-type eGFP-tagged HMGB1 partitioned into all three model condensates, with the highest partitioning observed in MED1-IDR droplets (Fig. 2g,h). The mutant HMGB1 protein displayed enhanced partitioning into NPM1 droplets (threefold compared with wild type, $P$ < 1 × 10⁻⁵, Welch's *t*-test) and to some extent in HP1α droplets (Fig. 2g,h). Mutant HMGB1 also tended to form dense foci within the MED1-IDR, HP1α and NPM1 droplets over time that appeared sequestered to the surface of the droplets (Fig. 2g). Enhanced partitioning into NPM1 condensates and foci formation were also observed using a 5′-carboxyfluorescein (5′ FAM)-labelled synthetic HMGB1 IDR mutant peptide, tested at

multiple concentrations (Fig. 2i and Extended Data Fig. 4h–j). These results reveal that mutant HMGB1 exhibits enhanced partitioning into NPM1 condensates in vitro.

## Nucleolar HMGB1 mispartitioning in vivo

We next sought to investigate the condensate behaviour of mutant HMGB1 in human cells. As primary culturable cells from individuals with BPTAS were not available, we ectopically expressed eGFP-tagged HMGB1 in U2OS cells. Wild-type HMGB1 displayed diffuse nuclear localization in live cells (Fig. 3a and Extended Data Fig. 5a,b). By contrast,

mutant HMGB1 localized to discrete nuclear inclusions (Fig. 3a, Extended Data Fig. 5a,b and Supplementary Video 2). Ectopic expression of the mutant HMGB1 IDR also led to the formation of nuclear inclusions in live U2OS cells (Fig. 3a and Extended Data Fig. 5a–c), which indicated that the replaced IDR of the mutant HMGB1 is responsible for its altered subnuclear localization. Nuclear inclusions were observed in several other human cell types expressing mutant HMGB1 (Extended Data Fig. 5d).

Mutant HMGB1 nuclear inclusions frequently contained cavities and resembled nucleoli. Nucleoli are phase-separated multiphasic condensates that contain an outer granular component enriched in NPM1 and an inner dense fibrillar component enriched in FIB1 (ref. [35]) (Fig. 3b). To gain initial insights into the nature of the mutant HMGB1 nuclear inclusions, we expressed FIB1 tagged with red fluorescent protein (RFP–FIB1) and eGFP–HMGB1 in live U2OS cells. The cavities in the mutant HMGB1 inclusions tended to encapsulate FIB1 (Fig. 3c). Fluorescence recovery after photobleaching (FRAP) experiments revealed that the HMGB1 shell displayed arrested dynamics around the FIB1 cores (Fig. 3d and Extended Data Fig. 5e). These results suggest that the mutant HMGB1 inclusions may be abnormal, arrested nucleoli.

To further probe the identity of mutant HMGB1 nuclear inclusions, we performed immunofluorescence against various nuclear proteins known to form condensates. The immunofluorescence analyses revealed that mutant HMGB1 inclusions were distinct from RNAPII and MED1 puncta, nuclear speckles and heterochromatin (Extended Data Fig. 5f–h). However, they overlapped with NPM1 and FIB1 (Extended Data Fig. 5h). The NPM1 signal within the HMGB1 inclusions inversely correlated with the HMGB1 signal (Pearson's $r = -0.70$) (Extended Data Fig. 5i). Moreover, the amount of diffuse NPM1 outside nucleoli correlated with the amount of HMGB1 in the inclusions (Pearson's $r = 0.50$) (Extended Data Fig. 5j). These results indicate that the mutant HMGB1 inclusions replace the NPM1-enriched granular component of nucleoli.

Targeted mutagenesis experiments revealed that arginine residues in the mutant HMGB1 tail drive nucleolar mispartitioning, and a hydrophobic patch drives nucleolar arrest. Various mutant HMGB1 sequences were expressed in live U2OS cells. A HMGB1 protein lacking the entire IDR (Del IDR) or the sequence after the frameshift position (Del FS) was not enriched in the nucleolus (Fig. 3e,f). Deletion of arginine residues (R del), substitution of arginine residues with alanine residues (R>A), substitution of arginine and lysine residues with alanine residues (R&K>A), and substitution of arginine residues with lysine residues (R>K) within the sequence created by the frameshift led to failure of the mutant protein to partition into the nucleolus (Fig. 3e,f). Furthermore, deletion of the short hydrophobic patch at the C terminus of the frameshifted sequence (Patchless) did not alter nucleolar mispartitioning (Fig. 3e,f and Extended Data Fig. 5k), but it did rescue the arrested dynamics of the mutant HMGB1 nucleoli assessed by FRAP (Fig. 3e,g). These results demonstrate that nucleolar mispartitioning of the frameshift mutant HMGB1 depends on arginine residues within the sequence created by the frameshift and that the hydrophobic patch contributes to nucleolar arrest.

## Mutant HMGB1 and nucleolar dysfunction

To test whether the nucleolar mispartitioning of HMGB1 affects nucleolar function, we investigated ribosomal RNA (rRNA) production using quantitative PCR with reverse transcription (RT–qPCR)[36]. The level of 28S rRNA in U2OS cells expressing the frameshift mutant HMGB1 was significantly reduced by about 1.5-fold ($P < 0.05$, Student's $t$-test) (Extended Data Fig. 6a). Ribosomal dysfunction was subsequently probed using an assay of nascent translation that measures puromycin incorporation[37]. U2OS cells expressing mutant eGFP–HMGB1 consistently displayed lower levels of puromycin intensity than non-transfected (that is, GFP⁻) cells and cells transfected with wild-type eGFP–HMGB1 (Fig. 3h,i and Extended Data Fig. 6b,c). Furthermore,

U2OS cells expressing the mutant HMGB1 exhibited substantially reduced viability after several days of culture compared with cells expressing wild-type HMGB1 ($P < 5 \times 10^{-4}$, one-way analysis of variance (ANOVA)) (Fig. 3j and Extended Data Fig. 6d). The reduced viability was associated with nucleolar arrest, as transfection of cells with the Patchless mutant did not compromise viability (Fig. 3j and Extended Data Fig. 6d). The findings of nucleolar mispartitioning, nucleolar arrest and viability were corroborated using cell lines expressing stably integrated eGFP–HMGB1 transgenes from a PiggyBac transposon (Extended Data Fig. 6e–j). These results indicate that the presence of the HMGB1 frameshift mutant in cells disrupts nucleolar function and is cytotoxic.

## ACMG classification of HMGB1 variants

The clinical and genetic information of the five individuals with BPTAS and the functional data were used to classify the *HMGB1* frameshift variants as pathogenic. This classification was made in accordance with the criteria of the American College of Medical Genetics and Genomics (ACMG)[38]. Both frameshifts observed in individuals with BPTAS result in the replacement of the highly conserved acidic tail of the protein (ACMG criterion PM1), and were classified as pathogenic by MutationTaster (ACMG criterion PP3). Notably, of the 43 nonsynonymous variants in the HMGB1 tail (1,123 alleles) listed in the gnomAD database (v.2.1.1)[39], only 4 variants (5 alleles) introduce amino acids other than aspartate and glutamate (ACMG criterion PM2) (Extended Data Fig. 2f,g). All previously described pathogenic *HMGB1* variants are associated with neurodevelopmental phenotypes without severe skeletal anomalies, which are therefore distinct from BPTAS (Fig. 1c and Supplementary Table 2), including a chromosomal microdeletion encompassing the *HMGB1* locus in an individual (I6) diagnosed in our study (Extended Data Fig. 2h–j, Supplementary Note and Supplementary Table 2). The functional data presented in this study suggest a deleterious effect (ACMG criterion PS3). In summary, the identification of almost the same (ACMG criterion PS4) *HMGB1* frameshift variants in five (shown to be de novo in four; ACMG criteria PS2 and PM6) unrelated individuals with the same ultrarare diagnosis of BPTAS (ACMG criterion PP4) argues for the classification of these variants as pathogenic (ACMG evidence level: 3S+2M+2P; Supplementary Note).

## Catalogue of variants in C-terminal IDRs

We then sought to investigate whether replacement of a disordered C-terminal tail with an arginine-rich basic tail and the consequent nucleolar mispartitioning and dysfunction could occur in other diseases. To this end, we generated a catalogue of genetic variants in intrinsically disordered tails of cellular proteins. First, we annotated 9,303 isoforms of 5,618 genes that have a C-terminal IDR consisting of at least 20 amino acids (Supplementary Table 3). We then identified genetic variants that occur in the disordered tails of the 5,618 genes annotated in the 1000 Genomes Project, ClinVar, COSMIC and dbSNP databases. These analyses revealed 249,464 genetic variants in C-terminal IDRs, including 10,023 truncating variants and 3,888 frameshifts that replace the C-terminal sequence with ≥20 amino acids (Fig. 4a, Extended Data Fig. 7a–c and Supplementary Tables 4 and 5). Of the 3,888 frameshifts, 426 were annotated as pathogenic in ClinVar, 763 were common variants curated in the 1000 Genomes Project and 189 in the dbSNP databases (Fig. 4b, Extended Data Fig. 7b,c and Supplementary Table 5). The frameshifts were associated with higher-than-average pathogenicity (Fig. 4a), and frameshifts were enriched for pathogenic variants (Fig. 4b and Extended Data Fig. 7c). Genes encoding transcription factors were highly enriched among those that contained C-terminal IDR mutations (Extended Data Fig. 7d). Among the 3,888 frameshift variants, 624 were predicted to result in a sequence consisting of at least 15% arginine residues, of which 101 were classified as pathogenic in ClinVar (Fig. 4b,c

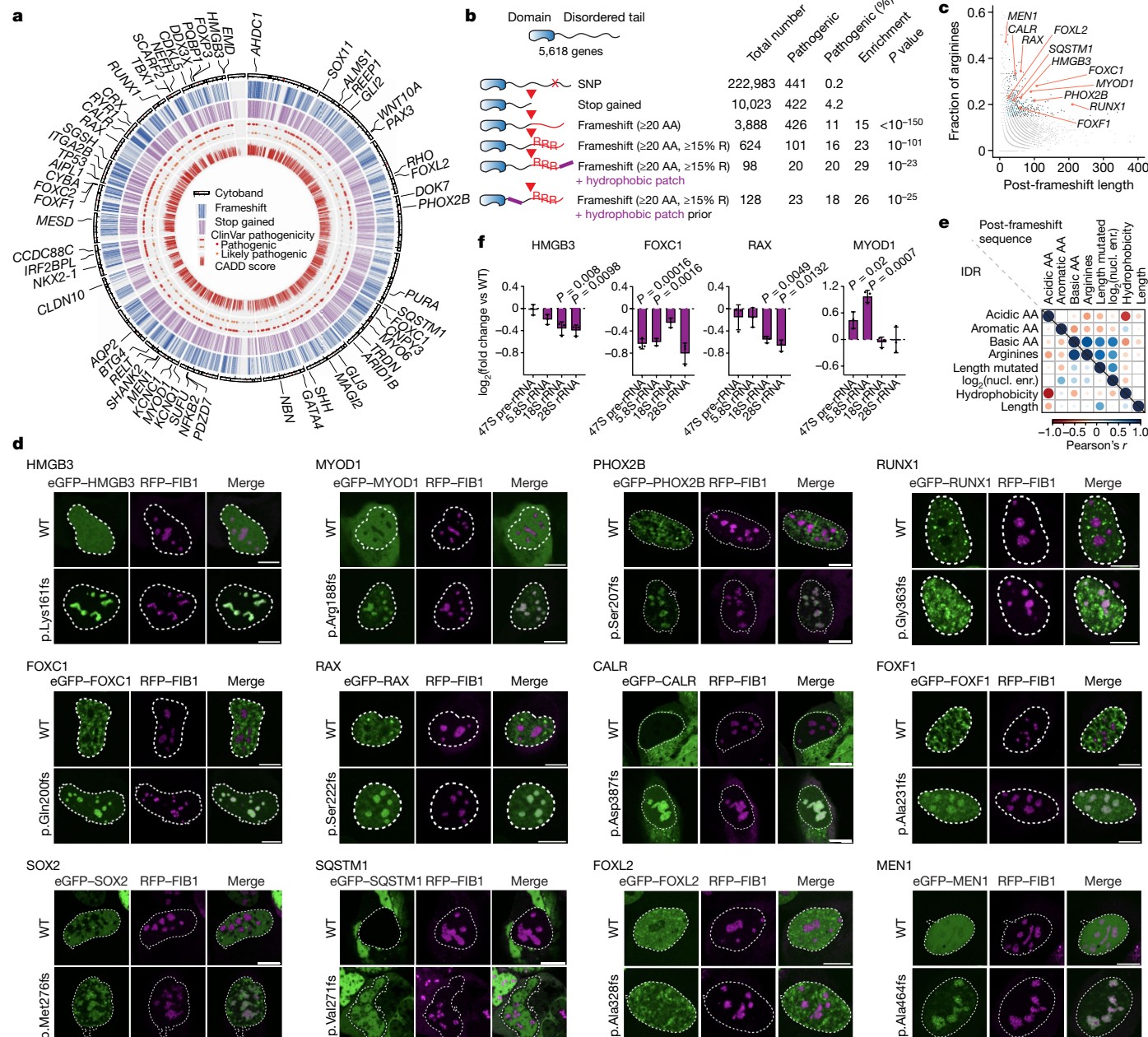

**Fig. 4 | A catalogue of variants in C-terminal IDRs reveals frameshifts associated with nucleolar mispartitioning and dysfunction. a**, Circos plot of the IDR variant catalogue. The circles indicate the location of genes that contain a truncation (stop gained) or frameshift variant in the dbSNP, 1000 Genomes Project, COSMIC and ClinVar databases. The highlighted genes contain a pathogenic frameshift that creates a sequence of ≥20 amino acids comprising ≥15% arginine residues. **b**, Summary statistics and features of variant types in C-terminal IDRs. *P* values are from hypergeometric tests. **c**, Identification of frameshifts creating a sequence of ≥20 amino acids that consist of ≥15% arginine residues. Plotted is the fraction of arginine residues against the length of the sequence created by the frameshift. The genes containing the variants selected for further validation are highlighted orange. Pathogenic gene variants are in blue. **d**, Representative images of U2OS cells co-expressing RFP–FIB1 and the indicated eGFP-tagged proteins. Nuclear area revealed by Hoechst staining is shown as dashed white lines. Mutations in the following genes are associated with the indicated conditions: microphthalmia (*HMGB3* and *RAX*); myopathy (*MYOD1*); congenital central hypoventilation (*PHOX2B*); myelodysplasia (*RUNX1*); Axenfeld–Rieger syndrome type 3 (*FOXC1*);

myelofibrosis (*CALR*); alveolar capillary dysplasia (*FOXF1*); anophthalmia/microphthalmia-oesophagealatresia syndrome (*SOX2*); Paget disease of bone 2, early-onset frontotemporal dementia and amyotrophic lateral sclerosis (*SQSTM1*); blepharophimosis, ptosis and epicanthus inversus (*FOXL2*); and hereditary cancer predisposing syndrome (*MEN1*). Scale bar, 10 μm. **e**, Nucleolar mispartitioning strongly correlates with the fraction of arginine residues in the frameshift sequence. Plotted are Pearson's correlation coefficients of the extent of nucleolar mispartitioning of mutant proteins with protein features of their IDRs (left triangle) and features of the sequences created by the frameshifts (right triangle). The colour corresponds to the value of Pearson's correlation coefficients, and the size of the circles is proportional to the *P* value of the Pearson's *r*. **f**, RT–qPCR analysis of rRNA species in U2OS cells expressing the indicated WT and mutant proteins. rRNA levels are normalized against an RNAPII transcript (*GAPDH*), and fold changes are calculated against the rRNA/*GAPDH* level measured in the cells expressing WT protein. Data are shown as mean ± s.d., *P* values are from two-tailed Welch's *t*-test. AA, amino acid; SNP, single nucleotide polymorphism; nucl. enr., nucleolar enrichment.

and Supplementary Fig. 2a–g). Overall, 29 out of 66 genes containing arginine-rich frameshift variants had a probability of loss-of-function intolerance (pLI) score of <0.05, which is consistent with a potential gain-of-function effect of the variants (Extended Data Fig. 7e). The variants were associated with various pathogenic conditions, including neurodevelopmental diseases and cancer predisposition (Extended Data Fig. 7f–h). Moreover, 98 of the frameshifts also created a sequence resembling the short hydrophobic patch encoded by the HMGB1 frameshift (Fig. 4b), and 128 of the frameshifts occurred in genes that contained at least one hydrophobic patch in their IDR (Fig. 4b). Overall, the catalogue revealed >200,000 variants in C-terminal IDRs, including 624 frameshifts that replace a C-terminal tail with an arginine-rich basic tail, of which 101 frameshifts were classified as pathogenic.

Genes containing pathogenic frameshift variants that create an arginine-rich basic tail were expressed in U2OS cells. As such frameshifts are highly enriched in genes that encode transcription factors, we selected nine transcription factors (HMGB3, FOXC1, FOXF1, MYOD1, RAX, RUNX1, SOX2, PHOX2B and FOXL2) and four additional proteins (MEN1, SQSTM1, CALR and DVL1) for functional testing (Fig. 4d, Extended Data Figs. 8 and 9a–d and Supplementary Fig. 3a–d). The frameshift mutants of 12 out of the 13 proteins formed nuclear inclusions that overlapped the FIB1–RFP-labelled dense fibrillar component of the nucleolus in live cells (Fig. 4d and Extended Data Fig. 9e–i). The extent of mispartitioning into the nucleolus strongly correlated with the length of the IDR sequence replaced by the frameshift and the fraction of arginine residues in the sequence created by the frameshifts (Fig. 4e and Supplementary Fig. 4a,b). For six variant proteins, cavities enriched in FIB1–RFP were apparent (Fig. 4d and Extended Data Fig. 10a). FRAP experiments showed that condensate properties for 7 out of the 13 variants were affected (Extended Data Figs. 9g and 10b). Six of the mutant proteins that showed significant nucleolar enrichment were further analysed. For four out of six, changes in the level of rRNA species in cells expressing the frameshift mutants were detectable (Fig. 4f and Extended Data Fig. 10c,d). These results indicate that disease-associated frameshifts that generate an arginine-rich basic tail in C-terminal IDRs can cause nucleolar mispartitioning and dysfunction.

## Discussion

We propose that disease-associated and common variants in disordered regions may alter phase separation and partitioning of proteins into biomolecular condensates. In particular, the results presented here indicate that frameshift variants that substantially increase the arginine content of various proteins lead to mispartitioning into the nucleolus and disruption of nucleolar function. Our data identified the replacement of the disordered tail with an arginine-rich basic tail in HMGB1 as the pathomechanism underlying BPTAS, a rare complex malformation syndrome[13]. The HMGB1 variant appears to encode a sequence that combines high arginine content, reminiscent of the phase-separation grammar of native nucleolar proteins[40], and a hydrophobic patch that predominantly contributes to nucleolar arrest and dysfunction (Fig. 3e–j). The frameshift therefore interferes with the 'molecular grammar' of phase separation encoded in *HMGB1*, and the resulting mutant protein disrupts condensate features and function of the nucleolus where it accumulates. The extent to which the minimal propensity of the HMGB1 mutant sequence to form a helix contributes to these effects remains to be tested.

We provided evidence that arginine-rich frameshifts occur in hundreds of proteins, which implies that there is a common mechanism for hundreds of disease-associated and common genetic variants with previously unknown functions. The organismal effects of such frameshifts are probably influenced by tissue-specific expression and haplosufficiency (or haploinsufficiency) of the genes in which they occur. For example, BPTAS is associated with a frameshift in *HMGB1* that is broadly expressed and haploinsufficient (pLI score of 0.83),

which is consistent with phenotypic features presenting in multiple organ systems and partially overlapping with those seen when the locus is deleted (Supplementary Note). Of note, mispartitioning into the nucleolus and nucleolar dysfunction have been reported for poly-(proline:arginine)-dipeptides produced by repeat-expanded variants of *C9orf72* linked to amyotrophic lateral sclerosis[36,41,42]. Aberrant phase separation and nucleolar dysfunction may therefore occur in a wide range of genetic conditions as a shared underlying molecular pathomechanism.

Finally, the IDR variant catalogue provides a resource for exploring further models of how disease-associated variants may alter biomolecular condensates. For example, the >10,000 variants that truncate a C-terminal IDR may inhibit biogenesis of condensates, and several such variants have been associated with condensate dissolution in cultured cells[43]. Disease-associated alanine repeat expansions in a few transcription factors have been shown to alter the composition of their condensates[6], and our catalogue contains >200 frameshift sequences consisting of at least 25% alanine residues. In summary, we propose that disruption of phase separation may frequently occur in genetic diseases. Further investigation of the underlying molecular basis may lead to future strategies that alter phase separation with therapeutic intent.

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

[1]Institute of Medical Genetics and Human Genetics, Charité–Universitätsmedizin Berlin, corporate member of Freie Universität Berlin and Humboldt-Universität zu Berlin, Berlin, Germany. [2]BIH Biomedical Innovation Academy, Berlin Institute of Health at Charité–Universitätsmedizin Berlin, Berlin, Germany. [3]RG Development and Disease, Max Planck Institute for Molecular Genetics, Berlin, Germany. [4]Department of Genome Regulation, Max Planck Institute for Molecular Genetics, Berlin, Germany. [5]Exploratory Diagnostic Sciences, Berlin Institute of Health at Charité–Universitätsmedizin Berlin, Berlin, Germany. [6]Institute of Human Genetics, University Hospitals Schleswig-Holstein, University of Lübeck and Kiel University, Lübeck, Kiel, Germany. [7]Institute of Human Genetics, University Medical Center Hamburg-Eppendorf, Hamburg, Germany. [8]Center for Genomics and Transcriptomics (CeGaT), Tübingen, Germany. [9]Department of Pediatrics and Adolescent Medicine, School of Clinical Medicine, LKS Faculty of Medicine, The University of Hong Kong, Pok Fu Lam, Hong Kong. [10]Department of Congenital Heart Disease and Pediatric Cardiology, University Hospital Schleswig-Holstein, Kiel, Germany. [11]Department of Medical Genetics, Carl von Ossietzky University, Oldenburg, Germany. [12]Department of Obstetrics and Gynecology, University Hospital Schleswig-Holstein, Kiel, Germany. [13]Department of Internal Medicine, Radboud Institute for Molecular Life Sciences, Radboud Expertise Center for Immunodeficiency and Autoinflammation and Radboud Center for Infectious Disease (RCI), Radboud University Medical Center, Nijmegen, The Netherlands. [14]Department of Human Genetics, Radboud University Medical Center, Nijmegen, The Netherlands. [15]Department of Pediatrics, Pediatric Endocrinology and Diabetes, University Hospital Schleswig-Holstein, Schleswig-Holstein, Germany. [16]Handchirurgie, Katholisches Kinderkrankenhaus Wilhelmstift, Hamburg, Germany. [17]Genetics Research Center, University of Social Welfare and Rehabilitation Sciences, Tehran, Iran. [18]Department of Obstetrics and Gynaecology, Queen Mary Hospital, Pok Fu Lam, Hong Kong. [19]Institute of Human Genetics, Universitätsklinikum Erlangen, Friedrich-Alexander-Universität Erlangen-Nürnberg (FAU), Erlangen, Germany. [20]Institute for Human Genetics and Genomic Medicine, Medical Faculty, RWTH Aachen University Hospital, Aachen, Germany. [21]Department of Pediatric Neurology, Charité–Universitätsmedizin Berlin, corporate member of Freie Universität Berlin and Humboldt-Universität zu Berlin, Berlin, Germany. [22]Department of Pediatrics, University Hospital Center Schleswig-Holstein, Kiel, Germany. [23]Department of Radiology and Neuroradiology, Pediatric Radiology, University Hospital Schleswig-Holstein, Kiel, Germany. [24]Institute of Human Genetics, Medical Center, University of Freiburg, Faculty of Medicine, University of Freiburg, Freiburg, Germany. [25]Zentrum für Kinder-und Jugendmedizin, Helios Kliniken Schwerin, Schwerin, Germany. [26]Institute of Human Genetics, University Hospital, Otto-von-Guericke University, Magdeburg, Germany. [27]Institute for Research in Biomedicine (IRB Barcelona), The Barcelona Institute of Science and Technology, Barcelona, Spain. [28]Microscopy Core Facility, Max Planck Institute for Molecular Genetics, Berlin, Germany. [29]ICREA, Passeig Lluís Companys 23, Barcelona, Spain. [30]BCRT-Berlin Institute of Health Center for Regenerative Therapies, Berlin, Germany. [31]DZHK (German Centre for Cardiovascular Research), partner site Hamburg, Lübeck, Kiel, Lübeck, Germany. [32]These authors contributed equally: Martin A. Mensah, Henri Niskanen. [33]These authors jointly supervised this work: Malte Spielmann, Denise Horn, Denes Hnisz. ✉e-mail: Malte.Spielmann@uksh.de; Denise.Horn@charite.de; hnisz@molgen.mpg.de

## Methods

### DNA sequencing, array comparative genomic hybridization and qPCR

Genome sequencing and exome sequencing were performed using Illumina technology with a paired-end sequencing approach[26]. Genome sequencing data were filtered using VarFish. Information on excluded variants and filtering strategy are displayed in Extended Data Fig. 2a. Sanger sequencing and real-time qPCR were performed on a 3730 DNA analyzer (Thermo Fisher Scientific). Sanger sequencing of *HMGB1* from gDNA from individuals included in this study was performed using primers listed in Supplementary Table 6. For cDNA Sanger sequencing and RT–qPCR of I3, RNA was extracted from a patient and a control lymphoblastoid cell line using a Direct-zol RNA Miniprep kit (Zymo Research Europe). RNA was measured on a Nanodrop instrument (Thermo Fisher Scientific), and 1 µg of RNA was transcribed to cDNA using a RevertAid H Minus First Strand cDNA Synthesis kit (Thermo Fisher Scientific). Raw data of RT–qPCRs were analysed using the $2^{(-\Delta\Delta CT)}$ method normalized to *GAPDH*. For cDNA Sanger sequencing, the primers used for amplification and sequencing are listed in Supplementary Table 6. For RT–qPCR of cDNA from individuals included in this study, *HMGB1* and *GAPDH* primers are listed in Supplementary Table 6. Chromosomal microarray analysis was performed using a 4 × 180 k oligonucleotide slide from Agilent on a DNA microarray scanner (Agilent). Chromosomal microarray analysis results were confirmed by RT–qPCR. All procedures were performed using the manufacturers' protocols. All variants were annotated according to genome build hg19 and the *HMGB1* transcript NM_002128.7.

### Patient consent

Parental consent was obtained for all clinical and molecular studies of this article and for the publication of the relevant causative variants and of clinical photographs. Patient consent did not cover the release of personal sequence information other than the causative pathogenic variants. Therefore, whole-genome sequencing and exome sequencing data cannot be made publicly available. All studies and investigations were performed according to the declaration of Helsinki principles of medical research involving human participants, and the study was approved by the ethics committee of the Charité–Universitätsmedizin Berlin (EA2/087/15).

### Patient recruitment and clinical protocol

Individuals were recruited during routine patient care at five departments of genetics (Berlin, Kiel, Nuremberg, Schwerin, Hong Kong). Fetuses from spontaneous abortions were not systematically screened for BPTAS. No statistical methods were used to predetermine sample sizes. Investigators were not blinded and no randomization was used.

### Computer-aided facial phenotyping

Facial frontal images were analysed using the Face2Gene suite (v.20.1.4, https://www.face2gene.com). Face2Gene Clinic was used for computer-aided facial phenotyping[44]. We created a composite mask using Face2Gene Research. If several images of the same patient were available, the image depicting the individual at the oldest age was used for facial analysis by Face2Gene Clinic. Seven images of unrelated individuals diagnosed with BPTAS were taken from the literature (of those reported in ref. [15], only the father was included)[13–19]. In addition, I1 and I2 of the current study were included in the analysis. Each selected BPTAS image was used twice for Face2Gene Research analysis to reach more than the ten images necessary for composite mask creation (Extended Data Fig. 1s).

### AlphaFold predictions for protein structures

AlphaFold predictions were computed using an in-house implementation of AlphaFold[45] using v.2.0.0 from 16 July 2021. The preset parameter was set to --preset=casp14 to use all genetic databases and eight ensembles, matching the CASP14 prediction pipeline. Templates were restricted to those available before the CASP14 predictions using the parameter --max_template_date=2020-05-14. Models were rendered using UCSF ChimeraX (v.1.5)[46,47], colouring the structure with the pLDDT score. Multiple sequence analysis depth plots and per-model pLDDT sequence plots were made using custom scripts based on ColabFold notebook AlphaFold2 with MMseqs2 (ref. [48]). Predictions of *Mus musculus*, *Rattus norvegicus* and *Danio rerio* HMGB1(A) protein structures, shown in Extended Data Fig. 4a, are from the AlphaFold Protein Structure Database[45].

### Generation of DNA constructs for protein purification and expression in human cells

To generate plasmids for recombinant protein expression, *HMGB1* cDNA sequences containing the wild-type or NM_002128.7(*HMGB1*): c.551_554delAGAA;p.(Lys184Argfs*44) variant were ordered from Twist Bioscience. Full-length cDNAs and the regions encoding IDR sequences were cloned into a monomeric eGFP (meGFP)-pET45 backbone by Gibson assembly using NEBuilder HiFi DNA Assembly MasterMix (NEB); primers are listed in Supplementary Table 6. For the generation of pET45-mCherry–NPM1 and pET45-mCherry–HP1a, *NPM1* and *HP1A* open-reading frames were amplified from mouse cDNA using primers flanked with Gibson overhangs (sequences listed in Supplementary Table 6). The resulting amplicons were gel purified and cloned into pET45-mCherry (Addgene, 145279) linearized with AscI and HindIII restriction enzymes. For the generation of pET28-mCherry–MED1-IDR, mCherry was subcloned into the pET28-meGFP–MED1-IDR vector as previously described[6,31] using NcoI and BsrGI restriction sites.

To express monomeric eGFP–HMGB1 variants in mammalian cells, eGFP–HMGB1 sequences were subcloned from pET45-meGFP vectors into a pRK5-meGFP vector digested with AgeI and XbaI (Addgene, 18696); primers used are listed below. To express wild-type and frameshift variants of FOXC1, FOXF1, HMGB3, MYOD1, RAX, RUNX1, PHOX2B, CALR, SOX2, SQSTM1, FOXL2, MEN1 and DVL1, the following cDNA sequences were ordered from Twist Bioscience: NM_001453.3(*FOXC1*):c.599_617del;p.(Gln200Argfs*109), variant rs1057519478; NM_001451.3(*FOXF1*):c.691_698del;p.(Ala231Argfs*61), variant 692054; NM_005342.4(*HMGB3*):c.480_481dup;p.(Lys161Ilefs*55), variant rs431825172; NM_002478.5(*MYOD1*):c.557dup;p.(Arg-188Profs*90), variant rs1179926739; NM_013435.3(*RAX*):c.664del; p.(Ser222Argfs*63), variant rs1603388837; NM_001754.5(*RUNX1*): c.1088_1094del;p.(Gly363Alafs*229), variant 1013621; NM_004343.4 (*CALR*):c.1157_1158dup;p.(Asp387Argfs*44), variant COSV104394382; NM_003924.4(*PHOX2B*):c.618del;p.(Ser207Alafs*102), variant 658418; NM_023067.4(*FOXL2*):c.982del;p.(Ala328Profs*28), variant 369937; NM_003106.4(*SOX2*):c.828del;p.(Met276Ilefs*95) variant 986766; NM_003900.5(*SQSTM1*):c.810del;p.(Val271Serfs*41) variant 967349; NM_001370259.2(*MEN1*):c.1382_1389dup;p.(Ala464Argfs*98) variant 428075; NM_004421.2(*DVL1*):c.1505_1517del;p.(His502Profs*143). For genotype–phenotype correlations see Supplementary Note.

cDNAs were amplified with primers listed in Supplementary Table 6 and cloned into a pRK5-meGFP–HMGB1 vector using Gibson assembly after removing the *HMGB1* sequence with BsrGI and XbaI restriction enzymes. To test the contribution of arginine and lysine residues of the mutant HMGB1 sequence, cDNA sequences were ordered from Twist Bioscience, in which all arginine and lysine residues after Lys185 were replaced with alanine (R&K>A variant), all arginine residues after Lys185 were deleted (R del variant) or replaced with alanine or lysine (R>A and R>K, respectively, variants). cDNAs were amplified using the primers listed below and cloned into a pRK5-meGFP–HMGB1 vector as described above. To create truncated versions of HMGB1, in which the IDR (amino acids after Asn134), or the sequence after the frameshift position (del FS) or the hydrophobic patch of the mutant sequence (amino acids after Lys209) is deleted, cDNA was amplified from pRK5-meGFP-HMGB1

using the primers listed in Supplementary Table 6 and cloned back to a vector digested with BsrGI and XbaI as described above. All constructs were sequence-verified. Plasmids are available from Addgene (https://www.addgene.org/Denes_Hnisz/).

## Protein purification and peptide synthesis

Protein expression of mCherry constructs was performed as previously described[6,33], but with modifications to mCherry–MED1-IDR expression, which was performed in the presence of 400 µg ml[−1] kanamycin. Protein expression of meGFP–HMGB1 constructs was performed in Rosetta (DE3)pLysS cells (Sigma-Aldrich) in the presence of 25 µg ml[−1] chloramphenicol and 100 µg ml[−1] ampicillin. All bacterial pellets were stored at −80 °C. Pellets were resuspended in 20 ml of ice-cold buffer A (50 mM Tris pH 7.5, 500 mM NaCl, 20 mM imidazole and complete protease inhibitors (Sigma-Aldrich, 11697498001)), and cells were lysed using a Qsonica Q700 sonicator. Lysate was cleared by centrifugation at 15,500$g$ for 30 min at 4 °C, and proteins were purified using an Äkta avant 25 chromatography system and a complete His-Tag purification column (Merck, 6781543001). Columns were pre-equilibrated in buffer A, loaded with cleared lysate and washed with 15 column volumes of buffer A. Fusion proteins were eluted in 10 column volumes of elution buffer (50 mM Tris pH 7.5, 500 mM NaCl and 250 mM imidazole). Protein preparations were diluted in storage buffer (50 mM Tris pH 7.5, 125 mM NaCl, 1 mM DTT and 10% glycerol) and concentrated using 3000 MWCO Amicon Ultra centrifugal filters (Merck, UFC803024) and stored at −80 °C. After His-Tag column purification, meGFP–HMGB1 protein preparations were further purified using Superdex 200 10/300 GL columns (GE28-9909-44) and concentrated and stored as noted above. Elution profiles are shown in Extended Data Fig. 4a. We note that the mutant protein elutes at lower elution volumes, which indicates that it may form soluble oligomers and that the potential to form soluble oligomers may be associated with the slight propensity of the mutant IDR to form a helix (Extended Data Fig. 3c,d). Immunoreactivity of purified meGFP–HMGB1 proteins were evaluated by western blotting. Equal amounts of protein were diluted in NuPAGE LDS buffer (Thermo Fisher Scientific, NP0007) with NuPAGE sample-reducing agent and heated at 70 °C for 10 min. Samples were run using NuPAGE 4–12% Bis-Tris protein gels (Invitrogen, NP0321PK2) and transferred to a nitrocellulose membrane with an iBlot2 device. The membrane was blocked with 5% non-fat milk TBST for 1 h and incubated 1 h with anti-HMGB1 (Sigma-Aldrich, H9664) or anti-eGFP (Invitrogen, A-11122) antibodies diluted 1:1,000 in 5% non-fat milk TBST. Membranes were washed five times with TBST, incubated with HRP-conjugated donkey anti-rabbit antibody (1:2,000, Jackson Immuno Research, 711-035-152) for 1 h, washed five times in TBST and visualized using SuperSignal West Dura Extended Duration substrate (Thermo Scientific, 34075). The identity of the fusion protein products was confirmed by mass spectrometry.

Synthetic peptides with amino-terminal 5′ FAM-labelling for in vitro droplet formation assays (Fig. 2i and Extended Data Fig. 4d–i) and circular dichroism (CD) spectroscopy experiments (Extended Data Fig. 3c,d) were ordered for wild-type and mutant HMGB1 C-terminal sequences (Asp135 onwards) from ProteoGenix. The synthetic peptides had >90% purity.

## CD experiments

The synthetic peptides were dissolved in 20 mM sodium phosphate buffer, pH 7.4. The samples were centrifuged for 10 min at 15,000 r.p.m. to remove undissolved solid. The supernatant was extensively dialysed against 20 mM sodium phosphate buffer, pH 7.4, to remove traces of impurities from peptide synthesis. The protein concentration was determined by amino acid analysis. CD spectra were acquired on 10.6 µM samples in a Jasco 815 UV spectrophotopolarimeter at 278 K with a 1 mm optical path cuvette. Each spectrum is the result of 20 cumulative scans acquired at a scanning speed of 50 nm min[−1] with a data pitch of 0.2 nm (Extended Data Fig. 4c,d).

Reference CD spectra in Extended Data Fig. 4e are included from the Protein Circular Dichroism Data Bank[49]. The following reference proteins were used: myoglobin (blue)[50], with a DSSP α-helix of 73.9%; outer membrane protein g (OmpG, purple)[51], with a DSSP β-strand of 67.6%; and translocated actin recruiting phosphoprotein (Tarp, green)[52], with a DSSP loop of 71.0%.

## In vitro droplet formation experiments

For droplet formation experiments in Fig. 2c–e, proteins were diluted to desired concentrations in storage buffer, further diluted 1:1 in 20% PEG-8000 and mixed well with pipetting. Next, 10 µl of solution was immediately transferred on a chambered coverslip (Ibidi, 80826-96). Droplets were imaged using a LSM880 confocal microscope (Zeiss) with a ×63, 1.40 oil DIC objective. Images were acquired slightly above the solution interface; for FRAP experiments, images were acquired directly on the solution interface. Time series for FRAP experiments were acquired using 60 cycles of 2 s intervals, during which the eGFP signal was bleached using a 488 nm laser with 95% intensity after the second interval. FRAP was performed for at least ten droplets for both wild-type and mutant HMGB1 using 10 µM concentration. Recovery curves were fitted to a power-law model. For droplet assays using pre-assembled mCherry–HP1α, mCherry–MED1-IDR and mCherry–NPM1 condensates (Fig. 2g–i), mCherry-labelled proteins were diluted to 20 µM concentration in storage buffer, diluted 1:1 in 20% PEG-8000 and droplets were allowed to form for 1 h at room temperature, shielded from light. Next, eGFP–HMGB1 proteins or 5′ FAM-labelled synthetic IDR peptides were added to the desired concentration, thoroughly mixed and solutions were left to equilibrate for 45 min at room temperature, shielded from light. Droplets were imaged as described above. To test the contribution of RNA for the condensation propensity of HMGB1 IDR peptides, total RNA from V6.5 mouse embryonic stem cells was isolated using a Direct-zol RNA Miniprep kit and added in indicated concentrations into peptide dilutions. RNA–peptide dilutions were thoroughly mixed with pipetting, crowding agent was added and imaging was performed as described above.

## Cell culture

U2OS, HCT116 and HEK293T cells were cultured in DMEM with GlutaMAX (Thermo Fisher Scientific, 31966-021) supplemented with 10% FBS and 100 U ml[−1] penicillin–streptomycin (Gibco). MCF7 cells were cultured in RPMI-1640 supplemented with 20% FBS and 100 U ml[−1] penicillin–streptomycin (Gibco). Human induced pluripotent stem (iPS) cells ZIP13K2 (ref. [53]), were grown in mTeSR Plus (Stem Cell Technologies, 100-0276) on plates coated with 1:100 diluted Matrigel (Corning, 354234) in KnockOut DMEM (Thermo Fisher Scientific, 10829-018) and supplemented with 10 µM of the Rho kinase inhibitor Y-27632 (Abcam, ab120129) once detached during passaging. Cells were cultured at 37 °C with 5% $CO_2$ in a humidified incubator. All cell lines were tested negative for mycoplasma contamination. For live-cell imaging and immunofluorescence, cells were seeded on chambered coverslips (Ibidi, 80826-96). On the next day, cells were transfected using FuGENE HD (Promega) according to the manufacturer's instructions. Human iPS cells were transfected using Lipofectamine 3000 according to the manufacturer's instructions. For viability experiments, cells were cultured on 6-well plates. Transfection series were repeated at least twice for each experiment.

## RT–qPCR after expression of frameshift variants in U2OS cells

Cells were grown on 6-well plates, transfected with FuGENE HD according to manufacturer's instructions, and eGFP[+] cells were sorted by FACS 48 h after transfections and lysed in TRIzol reagent (Thermo Fisher Scientific). Experiments were performed in at least three biological replicates. RNA was extracted and cDNA synthesis was performed as described above, except that 125 ng of RNA was used. Primers are listed in Supplementary Table 6.

## Live-cell imaging

Cells were imaged 24 h after transfections using a LSM880 confocal microscope (Zeiss) equipped with an incubation chamber with 5% $CO_2$ and a heated stage at 37 °C. Images were acquired using a ×63, 1.40 oil DIC objective. To visualize cell nuclei, cells were incubated with 0.2 µg ml$^{-1}$ Hoechst (Thermo Scientific, 33342) at least 10 min before imaging. To visualize nucleoli in living cells, we expressed RFP–fibrillarin fusion proteins by transfecting cells with pTagRFP-C1-fibrillarin plasmid (Addgene, 70649) together with plasmids for eGFP–HMGB1 and other transcription factor variants.

FRAP experiments were performed for nucleolar regions in cells expressing wild-type or mutant eGFP–HMGB1, guided by the RFP–fibrillarin fluorescence channel. Time series for FRAP experiments were acquired using 20 cycles of 2 s intervals, during which the eGFP signal was bleached using a 488 nm laser with 85% intensity after the second interval. FRAP experiments with designed variants of HMGB1 and other frameshift variants were performed as described above, but using 85–100% laser intensities for bleaching with identical settings for each wild type–mutant comparison. Fluorescence intensities were acquired from around ten regions of interest from separate nuclei, quantified using ZEN Black 2.3 software and reported as relative values to the pre-bleaching time point.

Time-lapse imaging of mutant HMGB1 expressing U2OS cells was performed on a Screenstar microplate (Greiner bio-one, 655866) with Zeiss Celldiscoverer 7. Images were acquired fully automated with a Plan-ApoChromat ×20 objective, NA = 0.7 and 1× tubelense (Optovar) using 15 min intervals and a camera binning of 1 × 1 pixel in 8-bit mode (Supplementary Video 2).

## Immunofluorescence

For fixed-cell immunofluorescence, cells were fixed 24 h after transfections with 4% PFA in PBS for 10 min. After two washes with PBS, cells were permeabilized by incubating 30 min with 0.5% Triton X-100 at room temperature, washed three times with PBS and blocked for 1 h with blocking buffer (1% BSA, 0.1% Triton X-100 in PBS) at room temperature. Samples were incubated with primary antibodies diluted in blocking buffer (1:500 rabbit anti-HP1α, Cell Signaling, 2616S; 1:500 rabbit anti-MED1, Abcam, ab64965; 1:500 rabbit anti-RNAPII, ab26721; 1:250 mouse anti-NPM1, Thermo Fisher Scientific, 32–5200; 1:100 mouse anti-FIB1, Santa Cruz, sc-374022; 1:200 mouse anti-SC35, Sigma-Aldrich, S4045) overnight in 4 °C with gentle agitation. After four washes with blocking buffer, samples were incubated with secondary antibodies (1:1,000 dilutions of Alexa Fluor 647 donkey anti-mouse or anti-rabbit antibodies, Jackson Immuno Research, 715-605-150 and 711-605-152) for 1 h at room temperature. Samples were washed two times with blocking buffer, incubated for 3 min with 0.25 µg ml$^{-1}$ DAPI (Invitrogen, D1306) in PBS and washed five times with PBS.

## Protein synthesis labelling by puromycylation

U2OS cells were seeded on 24-well plates (15,000 cells per well) on sterilized 13 mm glass coverslips pretreated with 0.2% gelatin. The next day, cells were transfected with meGFP–HMGB1 full-length wild-type or mutant constructs using FuGENE HD according to the manufacturer's instructions. After 24 h, pulse labelling of nascent peptide chains actively translated by the ribosome was performed by replacing the medium supplemented with 20 µM puromycin (Sigma Aldrich, P8833) for 15 min at 37 °C, 5% $CO_2$. Cells were then washed three times with cold PBS, followed by fixation with 4% formaldehyde (Roth, P087.5) at room temperature, with shaking, for 20 min. Fixative was removed, and cells were washed two times with PBS, followed by incubation in blocking solution (1× PBS, 5% v/v normal donkey serum, 1% w/v BSA, 0.1% w/v glycine and lysine) with shaking for 45 min at room temperature. Anti-puromycin (1:1,000, mouse, Sigma Aldrich, MABE343, RRID:AB_2566826) and anti-GFP (1:2,000, chicken, Abcam, ab13970,

RRID:AB_300798) primary antibodies were applied in blocking solution supplemented with 0.4% Triton-X-100 and incubated overnight with shaking at 4 °C. Cells were then washed three times with PBS for 5 min at room temperature, followed by secondary antibodies (1:250, Jackson ImmunoResearch, 488-anti-chicken, 703-545-155, RRID:AB_2340375; 647-anti-mouse, 715-605-151, RRID:AB_2340863) incubated in blocking solution with 0.4% Triton-X-100 shaking for 2 h at room temperature. After three PBS washes, cells were incubated in DAPI (1:2,500) in PBS for 30 min with shaking at room temperature, and washed with PBS an additional two times. Coverslips were removed from wells and sealed on poly-L-lysine slides (Thermo, J2800AMNZ) with ProLong Gold Antifade Mountant (Invitrogen, P36930). The experiment was performed in independent biological triplicates, with two to four technical replicate coverslips per conditions per experiment.

Coverslips were imaged using a Zeiss Celldiscoverer 7 running Zen Blue v.3.2 (Zeiss). All images were acquired in a fully automated fashion with a Plan-ApoChromat ×20 objective, NA = 0.95 and a ×2 tube lens (Optovar), and camera binning 2 × 2 pixels in 8-bit mode. The resulting lateral resolution ($xy$) is 0.227 µm pixel$^{-1}$. All images were acquired in tile regions of typically 20 × 20 individual tiles, resulting in 400 individual images per coverslip. Focus stabilization was achieved with an automated combined hardware and software focusing strategy at each second position (Fig. 3h,i and Extended Data Fig. 6b,c).

## Viability experiments

For viability experiments, cells were collected 24 h after transfections or doxycycline inductions and sorted for eGFP$^+$ cells using a FACS Aria II flow cytometer (BD Biosciences) with BD FACS Diva v.6.1.3. software. The FACS gating strategy is shown in Supplementary Fig. 6. One thousand cells per well were seeded on white microwell plates and were cultured for an additional 48 h. Viability was measured using a CellTiter-Glo 2.0 Cell Viability assay (Promega, G9242) according to the manufacturer's instructions. Measurements were done in three to five technical replicate wells and performed in four to five independent biological replicates. For imaging cells at the end of viability assay, 40,000 sorted cells were seeded per well on 24-well plates and imaged 48 h later with a Nikon Eclipse Ti2 microscope with a ×10 objective.

## Generation of doxycycline-inducible meGFP–HMGB1 transgenic cell lines

A PiggyBAC transposon system was used to integrate meGFP–HMGB1 wild-type and mutant sequences into U2OS cells. To generate the doxycycline-inducible expression cassette, *meGFP–HMGB1* cDNA was amplified from pRK5-meGFP–HMGB1 plasmids (primers listed in Supplementary Table 6), and Gibson assembly cloned into the backbone of a Caspex expression vector (Addgene, 97421) digested with NcoI and BsrGI restriction enzymes. Generated plasmids were transfected with a PiggyBAC transposase expression vector (SBI, PB210PA-1) into U2OS cells with FuGENE HD reagent according to the manufacturer's instructions using a molar ratio of 6:1 with meGFP–HMGB1 and transposase expression plasmids. Transfected cells were kept under puromycin (2 µg ml$^{-1}$) selection for 4 days, after which all untransfected control cells had died. Bulk populations of surviving cells were induced by adding 2 µg ml$^{-1}$ doxycycline (Sigma) and imaged 24 h after doxycycline treatments (Extended Data Fig. 6e–j). GFP$^+$ cells were sorted by FACS for viability experiments, which were performed as described above. Single-cell clones of meGFP–HMGB1 mutant-expressing U2OS cells was used for time-lapse imaging (Supplementary Video 2).

## Image analysis

For the detection of droplet regions for phase diagrams, we used the ZEN blue 3.2 Image Analysis and Intellesis software packages to analyse at least five images for each experimental condition. Image segmentation was performed using the Intellesis Trainable segmentation algorithm, which was trained on five representative images from

the image series to classify each pixel into the droplet area and image background. Regions of interest were automatically detected for the entire image series, and mean signal intensities for the eGFP or 5′ FAM channel and object areas for droplets and background are reported. In Fig. 2d, the phase-shifted fraction was calculated as the total area of detected droplets divided by the total area.

Data for dual-colour in vitro condensation experiments were acquired from 15–20 image fields for each condition (corresponding to Fig. 2g–i and Extended Data Fig. 4i,j) using ZEN Blue 3.2. For Extended Data Fig. 4j,k, droplets were first detected using triangle thresholding for light regions in the meGFP or 5′ FAM channel. For data analyses in Fig. 2h,i, droplets were detected using Otsu thresholding for light regions in the mCherry channel. Mean fluorescence intensity within droplet regions, area and diameter were then measured on both channels and plotted as described.

To quantify nuclear enrichment of eGFP–HMGB1, Hoechst stain was used to identify nuclei as the regions of interest using the ZEN Blue 3.2 zones of influence method. Images were automatically segmented with Otsu thresholding, parameters of which were adjusted on the basis of five representative images from the image series. The cytoplasmic region was defined as a ring surrounding the nucleus with a distance of 9 and a width of 29 pixels. Mean and standard deviation values for eGFP fluorescence intensity were recorded for nuclear and cytoplasmic regions, and nuclear enrichment, calculated as a ratio between the two, was plotted in Extended Data Fig. 5a. Cells with no expression (eGFP fluorescence intensity below 5) were excluded from the analysis.

To quantify the correlation between eGFP–HMGB1 fluorescence and NPM1 staining intensities inside and outside nucleoli, images from around 120 cells per condition were analysed using ZEN Blue 3.2 software. Images were first segmented to nuclear regions of interest with Otsu thresholding on the basis of DAPI channel intensity. Nuclei were further segmented to nucleolar regions of interest and regions outside the nucleoli, based on NPM1 staining intensity, using fixed thresholds that detected nucleoli in cells with high and low NPM1 intensities. Parameters were empirically set with ten representative images for each experimental set. Mean signal intensities for eGFP and NPM1 staining were recorded for each region of interest and reported as an average for each detected nucleus.

To quantify nucleolar enrichment of wild-type and frameshift variant proteins (Extended Data Fig. 9e), nuclear regions of interest were defined with Hoechst staining as outlined above and nucleolar regions with RFP–FIB1 intensity using two fixed thresholds that detect nucleoli in cells with high and low RFP–FIB1 expression. Mean signal intensities for eGFP were recorded, and nucleolar enrichment was plotted as $\log_2$(mean signal intensity for regions within nucleoli/mean intensity outside nucleoli). When imaging human iPS cells, nuclear regions of interest were eroded by 8 pixels to avoid signals at the nuclear periphery.

Data wrangling was performed in base R, and plots were generated using the ggplot2 package.

Image analysis for puromycylation experiments was performed using Zen Blue software v.3.4. DAPI was used to localize each cell. In brief, DAPI images were smoothed, an Otsu threshold was applied to binarize images and watershedding was used to separate neighbouring objects. The resulting nuclei masks were filtered to fit an area of 75–900 μm² and a circularity (sqrt(4 × area/π × FeretMax²)) of 0.6–1. The resulting primary objects were dilated with a total of 17 pixels, 3.9 μm. Puromycin and GFP signal intensities were quantified per cell. Puromycin intensity in each GFP⁺ cell was normalized by the mean puromycin intensity in GFP⁻ cells in the same image, for wild-type and mutant conditions, and plotted using R and GraphPad Prism, followed by comparisons for significant differences (one-way ANOVA) between condition means from biological replicates. A total of 37,979 single cells for mutant and 39,528 for wild-type conditions were identified and analysed (Fig. 3h,i and Extended Data Fig. 6b,c).

## C-terminal IDR identification

Prediction of IDRs was performed using metapredict (v.1.51)[54], a deep-learning-based predictor for consensus disordered sequences. The threshold score was set to 0.5, the minimum IDR length was set to 20 amino acids and the analysis was restricted to only GENCODE canonical or GENCODE basic isoforms. To complete the IDR catalogue, sequences from MobiDB[55] were added to the database. Protein coordinates for each IDR and Interpro domain were used to define the C-terminal IDR. Using a combination of custom scripts, the C-terminal IDR of each isoform was defined as any IDR that started 20 amino acids downstream of the start of the protein, to filter all disordered proteins. The region where the start of the IDR was downstream of the start of the most C-terminal domain was mapped.

## Variant identification and characterization

The resulting C-terminal IDR coordinates were then converted to genomic coordinates using the R package ensembldb[56] and the ensembl v.104 human annotation (v.2.22.0). The annotation version can affect the canonical isoforms that are selected for analysis, so the downstream analysis was locked to this version on Ensembl annotation. The resulting BED file was then used to filter ClinVar[57], COSMIC[58], dbSNP[59] and 1000 Genomes[60] to the designated genomic coordinates of the C-terminal IDR regions using BEDtools (v.2.30.0.)[61]. The resulting VCF file was filtered for protein-coding variant consequences using Ensembl Variant Effect Predictor (VEP, v.104)). The filtered VCF was then used to conduct downstream analysis using OpenCRAVAT[62] to annotate the variants for ClinVar annotation using the ClinVar and ClinGen[63] plugins, genomic frequencies using the 1000 Genome plugin, and CADD score[64,65] using the CADD plugin (v.1.6). The CADD score is a metric for the predicted effect of the variant on protein function (Fig. 4a). The same VCF file was also used to retrieve frameshift variant sequences using the Frameshift VEP plugin from pVACtools (v.3.1.0.)[66] and Downstream plugin for the stop gained sequences.

Sequences were then characterized using a combination of custom scripts to obtain protein sequence feature parameters based on local-CIDER (v.0.1.18.)[67] and biopython (v.1.79.)[68] packages. All scatter and violin plots were made using the R package ggplot2. The fraction of amino acids was defined as the sum of the count of amino acids over the sequence length. The acidic fraction was defined as the sum of aspartic acid and glutamic acid. The basic fraction was defined as the sum of arginine, lysine and histidine. The RK fraction was defined as the sum of arginine, lysine, and the aromatic fraction as phenylalanine, tyrosine and tryptophan. Hydrophobic patches were identified using custom regex expression (r'([CAVILMFYW]..?)<6,>') using hydrophobic amino acids as the dictionary, allowing 1 or 2 amino acid gap and 6 residue minimum match. Nucleolar signal prediction was caried using NoD program (v.1.0.0.) with the command line with default settings[69]. Characterization of nonsense-mediated mRNA decay of variants was done using a custom script. In brief, wild-type exon boundaries were retrieved from GENCODE and mapped to the wild-type coding sequence. An NMD sensitive zone was established for each wild-type sequence with the following rules: >100 bp downstream of starting codon and <51 bp of the second to last exon boundary. Variants with only one exon were marked 'NMD_escaping', then the stop codon coordinate of the variant was compared with the NMD sensitive zone coordinates and variants of which the stop codon did not overlap with the NMD sensitive zone were also marked as 'NMD_escaping'. All other variants were left empty.

Combined disordered and pLDDT score plots were plotted with the metapredict meta.graph_disorder function and pLDDT_scores parameter set to 'true', using v.2 of the metapredict network and v.7 of the pLDDT score prediction network.

Circos visualization of the variant catalogue was done using Circos implementation in R, and Granges package in R (Fig. 4a).

Enrichment analysis of pathogenic variants was done using hypergeometric nonaccumulative test with $N$ set as the full number of variants in the catalogue and $M$ set as the full set of pathogenic variants ($N = 249,468$ and $M = 1,805$). Reported $P$ values correspond to the calculated hypergeometric $P$ value and fold change as the number of pathogenic variants/expected number of pathogenic variants (Fig. 4b).

Sequence feature correlation matrices in Fig. 4e and Supplementary Fig. 4 were calculated using the cor package in R using Pearson parametric correlation test and plotted using the corrplot package in R. The $P$ value cut-off was set to 0.01. The fraction of mutated IDRs was defined as 1 − (frameshift position − IDR start)/IDR length. The *SQSTM1* wild-type sequence was excluded from correlation analysis because the wild-type isoform ENST00000510187.5 in our catalogue was replaced with isoform ENST00000389805.9 (NM_003900.5) in the imaging experiments owing to low transcript support level (TSL:5) for ENST00000510187.

Gene Ontology enrichment analysis (Extended Data Fig. 7d) for the variant type 'stop gained', 'frameshift' and 'ARG-rich FS' was done using gProfiler[70]. Multiple testing correction for $P$ values was done using the g:SCS method from g:Profiler.

Scores for the predicted disorder plotted in Fig. 2a and Extended Data Fig. 9a,c were obtained using PONDR (http://www.pondr.com). Charge plots in Fig. 1f and Extended Data Fig. 9b,d were prepared using EMBOSS Charge tool (https://www.bioinformatics.nl/cgi-bin/emboss/charge) with a window size of 8. Isoelectric points (pI) for post-frameshift sequences were calculated using Expasy compute pI tool (https://web.expasy.org/compute_pi/).

The DVL1 variant NM_004421.2(*DVL1*):c.1505_1517del was not part of the catalogue because the frameshift sequence from the canonical isoform used in ensembl v.104 did not fulfil all selection criteria. Instead, this variant was identified through a literature search that revealed Robinow syndrome-associated frameshift variants in the *DVL*1 gene that occur in a C-terminal IDR that generates arginine-rich sequences[71,72].

## Reporting summary

Further information on research design is available in the Nature Portfolio Reporting Summary linked to this article.

## Data availability

CD spectra have been deposited at the Protein Circular Dichroism Data Bank under the accession identifiers CD0006401000, CD0006401001, CD0006404000, CD0006404001. Genome and exome-wide summary statistics of I1, I4 and I5, and direct sequencing results of HMGB1 of I1–I4 and array comparative genomic hybridization and qPCR results of the *HMGB1* locus of I6 are made available in this manuscript (Extended Data Fig. 2). Patient consent did not cover the public release of personal sequence information other than the causative pathogenic variants. Therefore, the pathogenic variants are disclosed in this article, but individual-level whole-genome sequencing (WGS) and exome sequencing data cannot be made publicly available for reasons of data protection and patient privacy and are available only upon reasonable request from the corresponding authors. Access to individual-level sequencing data is subject to the policies and approval of the data protection officer of the institution that stores the patient data. WGS and Sanger sequencing data of I1 are stored at the Institute of Human Genetics, University Hospitals Schleswig-Holstein. WGS data of I4 is stored at the Center for Genomics and Transcriptomics (CeGaT) Tübingen. WGS data of I5, and Sanger Sequencing data of I2–I5, and qPCR and array comparative genomic hybridization data of I6 are stored at the Institute of Medical Genetics and Human Genetics, Charité–Universitätsmedizin Berlin. The respective servers are physically located in Germany.

## Code availability

Custom code is available at GitHub: https://github.com/hniszlab/HMGB1_2022; https://github.com/alexpmagalhaes/IDR-variant-catalog. Custom code and raw data for this study have been deposited at Zenodo (https://doi.org/10.5281/zenodo.7311150).

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

**Acknowledgements** We thank the patients and their families for participating in this study; V. Suckow, G. Hildebrand, V. Johnston and Y. Zhang for technical assistance; T. Aktas (MPI-MG) for comments on the manuscript; A. Papantonis (GAU, Göttingen) for advice on HMGB1 immunofluorescence; and U. Marchfelder and E. Weiß from the Flow Cytometry Facility of the MPI-MG for their assistance with cell sorting; D. Meierhofer and B. Lukaszewska-McGreal for performing mass spectrometry; and J. Naderi for mCherry constructs and human iPS cells. M.A.M. is a participant in the Digital Clinician Scientist Program and H.L.S. in the Junior Clinician Scientist Program founded by the late D. Dragun and funded by the Berlin Institute of Health and the Charité–Universitätsmedizin Berlin. This work was partially funded by the Max Planck Society. Work in the Hnisz Lab is supported by the Deutsche Forschungsgemeinschaft (DFG) Priority Program Grants HN 4/1-1 and HN 4/3-1 and a Worldwide Cancer Research grant (20-0232). M.S. is supported by grants from the DFG (SP1532/3-2, SP1532/4-1 and SP1532/5-1) and the Deutsches Zentrum für Luft- und Raumfahrt (DLR 01GM1925). S.M. was funded by

grant MU 880/16-1 from the DFG. H. Niskanen is supported by fellowships from the Orion Research Foundation and the Instrumentarium Science Foundation. C.G.-C. acknowledges a graduate fellowship from MINECO (PRE2018-084684) and X.S. acknowledges funding from AGAUR (2017 SGR 324), MINECO (PID2019-110198RB-I00) and the European Research Council (CONCERT, contract number 648201). IRB Barcelona is the recipient of a Severo Ochoa Award of Excellence from MINECO (Government of Spain).

**Author contributions** M.A.M., H. Niskanen and A.P.M. conceived and planned the study with input from S. Basu, S.M., M.S., D. Hnisz and D. Horn. M.A.M. managed the collection, analysis and interpretation of patient clinical and molecular data with M.S. and D. Horn. H. Niskanen designed and performed the cell biology experiments. M.L.K. performed the puromycin experiments. H. Niskanen designed and performed the biochemistry experiments with S. Basu. H. Niskanen, S. Basu and R.B. performed image analyses. A.P.M. performed AlphaFold2 modelling and built the variant catalogue with input from M.A.M., H. Niskanen and D. Hnisz. C.G.-C. performed the CD experiments. M.A.M., H. Niskanen, M.S., D. Hnisz and D. Horn wrote the manuscript with contributions from A.P.M. and S. Basu. M.S., D. Hnisz and D. Horn

supervised the study. All other authors contributed clinical or molecular data. All authors approved the final manuscript.

**Funding** Open access funding provided by Max Planck Society.

**Competing interests** D. Hnisz and X.S. are founders and scientific advisors of Nuage Therapeutics. The other authors declare no competing interests.

**Additional information**
**Correspondence and requests for materials** should be addressed to Malte Spielmann, Denise Horn or Denes Hnisz.

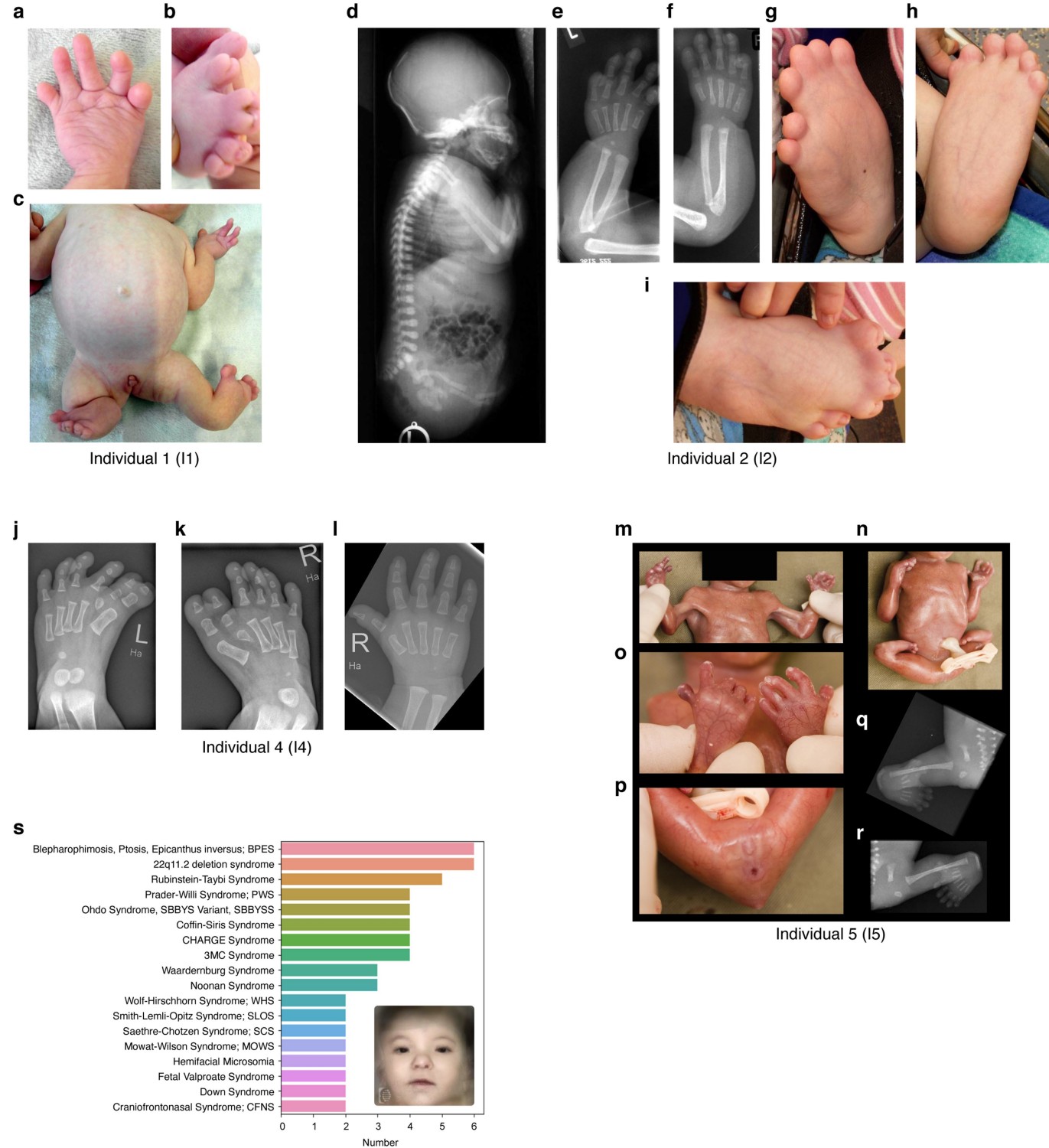

Individual 1 (I1)

Individual 2 (I2)

Individual 4 (I4)

Individual 5 (I5)

**s**

Blepharophimosis, Ptosis, Epicanthus inversus; BPES
22q11.2 deletion syndrome
Rubinstein-Taybi Syndrome
Prader-Willi Syndrome; PWS
Ohdo Syndrome, SBBYS Variant, SBBYSS
Coffin-Siris Syndrome
CHARGE Syndrome
3MC Syndrome
Waardernburg Syndrome
Noonan Syndrome
Wolf-Hirschhorn Syndrome; WHS
Smith-Lemli-Opitz Syndrome; SLOS
Saethre-Chotzen Syndrome; SCS
Mowat-Wilson Syndrome; MOWS
Hemifacial Microsomia
Fetal Valproate Syndrome
Down Syndrome
Craniofrontonasal Syndrome; CFNS

Number

**Extended Data Fig. 1 | Clinical findings in BPTAS individuals.** (**a-c**) I1 at age of 9 months. (**a**) Palmar view, left hand: brachydactyly and reduced creases of fingers. (**b**) Right foot: preaxial polysyndactyly, syndactyly between the second and third toes, increased soft tissue of distal toes in the dorso-ventral axis, hypoplastic/missing nails. (**c**) Malformed upper and lower limbs, contractures of large joints. (**d-i**) I2. (**d**) Lateral babygram (after birth): normal lateral spine apart from limb anomalies. (**e-f**) X-rays of upper extremities: contractures of the elbows, dislocation of the radius head, short radius and ulna, short middle phalanges. (**g-i**) I2, photos at age of 11 years. Plantar and dorsal views of the feet: preaxial polysyndactyly, hypoplastic nails. (**j-l**) I4. (**j**) Left foot, (**k**) Right foot, (**l**) Right hand. Radiograms of the feet at the age of 6 months showing symmetrical preaxial polysyndactyly. Radiogram of the right hand at age of 6 months: Note retarded bone age and short tubular bones, the middle phalanges are slightly more affected than the other ones. (**m-r**) I5 at 21 weeks of gestation. (**m**) Note webbed elbows. (**n**) Contractures of large joints. (**o**) Dorsal view of hands showing brachydactyly with hypoplastic nails. (**p**) Abnormal female genitalia. (**q-r**) Radiograms of the lower extremities and pelvis: hypoplastic iliac wings, lack of tibiae, hypoplastic fibulae, and preaxial polydactyly of feet. (**s**) Histogram of the syndromes suggested among the top-10 Face2Gene-suggestions of the images of individuals affected with BPTAS and the composite mask of this syndrome showing telecanthus and blepharophimosis.

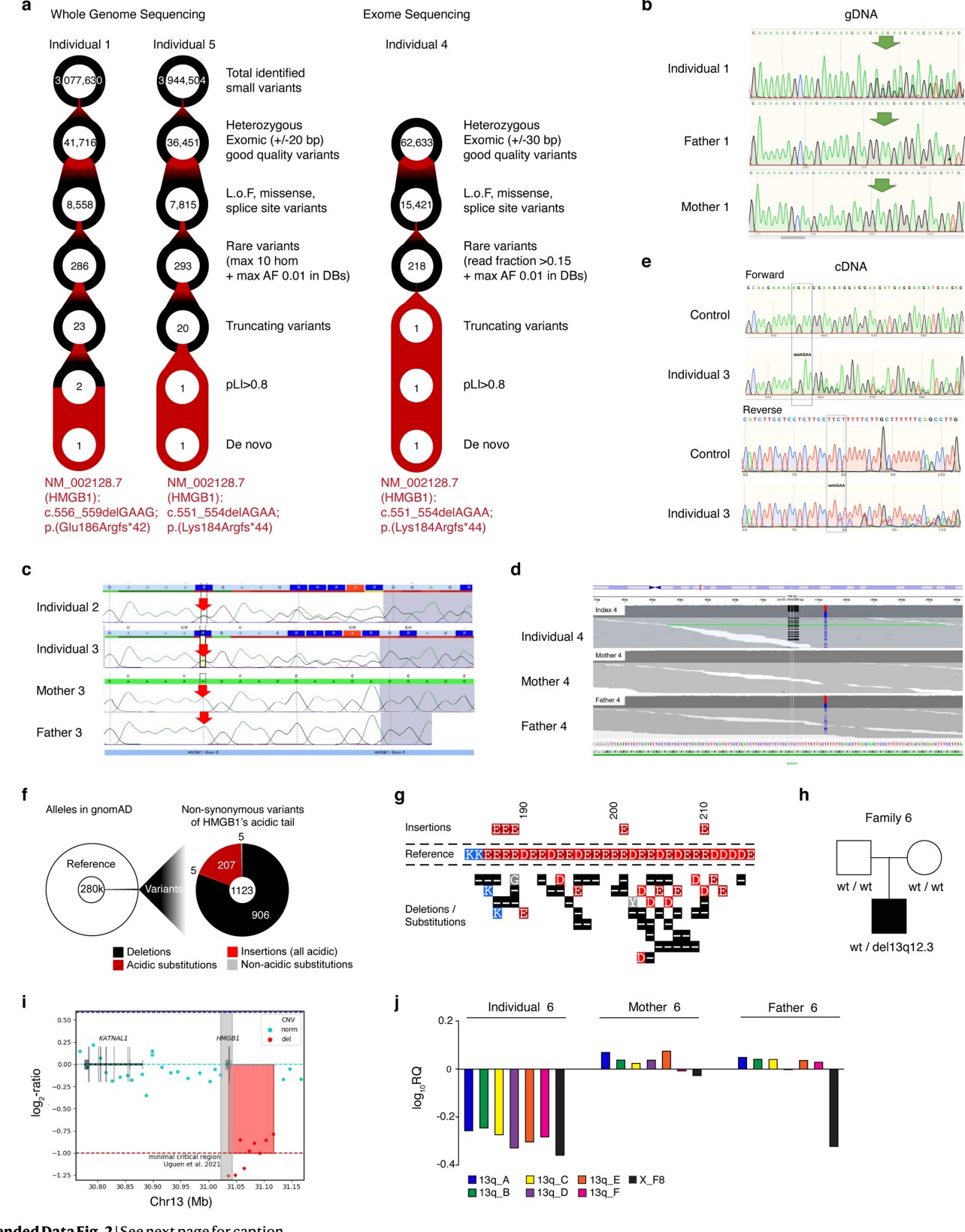

**Extended Data Fig. 2** | See next page for caption.

**Extended Data Fig. 2 | Patient genotyping and population-genetic data.**
(**a**) Variant detection and filtering scheme of the whole genome (I1 and I5) and exome sequencing data (I4). (**b**) gDNA Sanger sequencing data of *HMGB1* in I1. (**c**) gDNA Sanger sequencing data of *HMGB1* in I2, and I3, note identical de novo frameshift also found in I2 c.551_554delAGAA, de novo occurrence in I3. (**d**) Exome sequencing data of *HMGB1* in I4. Note identical frameshift as in I2 in I3, and de novo occurrence. (**e**) Sequencing of *HMGB1* cDNA in I3 and an unaffected control. Note the detection of both the wildtype and mutant cDNA in I3. (**f**) Allele counts of non-synonymous variants in HMGB1's acidic tail in the gnomAD database (v.2.1.1). Note that especially non-acidic substitutions are rare, and no frameshifts of *HMGB1* are listed in gnomAD. (**g**) Position of nonsynonymous variants in HMGB1's acidic tail. Note that most variants do not significantly shorten the uninterrupted succession of aspartic acid (D) and glutamic acid (E). The 4 non-acidic substitutions comprise merely 5 of 1123 non-synonymous alleles of the acidic tail listed in gnomAD. (**h**) Pedigree of family 6. Squares denote male, circles denote female individuals. Individuals diagnosed with BPTAS are highlighted with solid black boxes, and the genotypes are displayed below the boxes. WT = wildtype. (**i**) Microdeletion in 13q12.3 in I5. CMA data showing loss of *HMGB1* in I6. (**j**) qPCR showing de novo occurrence and revealing deletion of all exons, colored primers are positioned between the last non-deleted and first deleted oligo of the CMA, black primer X chromosomal control. Data are from one biological replicate.

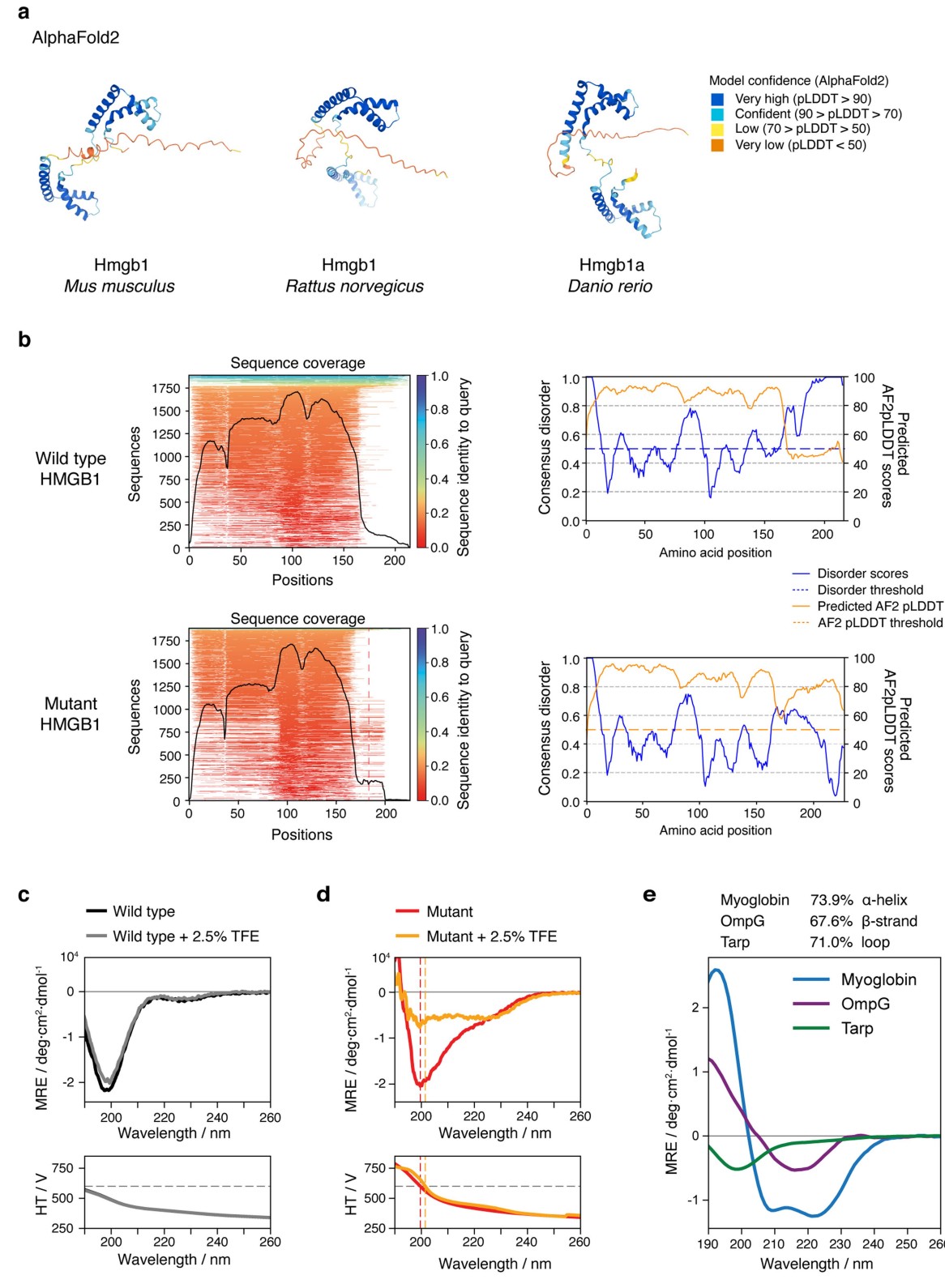

**Extended Data Fig. 3** | See next page for caption.

**Extended Data Fig. 3 | Computational and biochemical characterization of HMGB1.** (**a**) Predicted structures of Hmgb1 proteins from AlphaFold2 Protein Structure Database. Colors ranging from blue to orange depict the per-residue measure of local confidence for the model. (**b**) Left: MSA depth assessment for the sequences for quality assessment of the predicted HMGB1 models. Aligned sequences are colored by sequence identity. Sequence coverage frequency is depicted by a black line. The dotted red line marks the frameshift in the mutant. Right: Disorder analysis of wild type and mutant HMGB1 sequences using AlphaFold2 pLDDT scores (yellow) and Metapredict scores (blue). (**c**) Circular dichroism (CD) data of the WT HMGB1 IDR peptide in the absence (black) and in the presence (gray) of 2.5 % trifluoroethanol (TFE). On the upper panel, the CD spectra are shown as the mean residue ellipticity (MRE) as a function of wavelength. On the lower panel, the high-tension voltage (HT) values are shown as a function of wavelength. Vertical dotted lines indicate the wavelength value corresponding to HT = 600 V. (**d**) Circular dichroism (CD) data of the Mutant HMGB1 IDR peptide in the absence (red) and in the presence (orange) of 2.5 % trifluoroethanol TFE. On the upper panel, the CD spectra are shown as the mean residue ellipticity (MRE) as a function of wavelength. On the lower panel, the high-tension voltage (HT) values are shown as a function of wavelength. Vertical dotted lines indicate the wavelength value corresponding to HT = 600 V. (**e**) Representative CD spectra of α-helix, β-strand and disordered proteins, shown as the mean residue ellipticity (MRE) as a function of wavelength. The data was obtained from the Protein Circular Dichroism Data Bank (PCDDB) (see Methods).

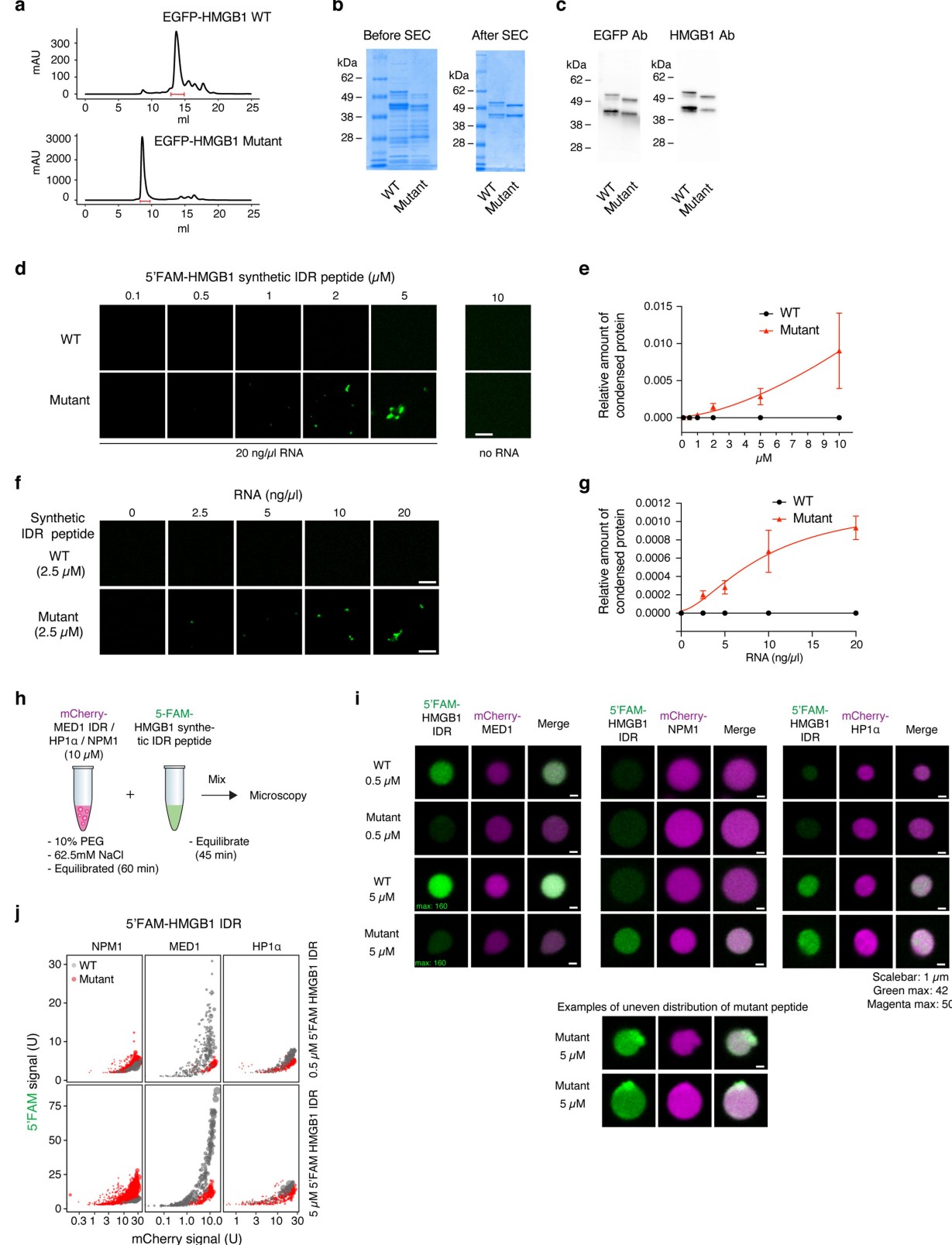

**Extended Data Fig. 4** | See next page for caption.

**Extended Data Fig. 4 | Mutant HMGB1 protein and synthetic IDR peptide.**
(**a**) Chromatograms from size-exclusion chromatography (SEC) of wild type and mutant mEGFP-HMGB1 fusion proteins. Selected fractions highlighted with red. (**b**) SDS-PAGE analysis of purified proteins after His-Tag column purification before SEC and after SEC purification steps. Analysis was performed once for each protein prep. (**c**) SDS-PAGE of SEC purified HMGB1 proteins followed by immunoblotting with anti-EGFP and anti-HMGB1 antibodies. Analysis was performed once for each protein prep. (**d**) Representative images from droplet formation assays performed with 5′FAM-labeled HMGB1-IDR variant peptides at indicated concentrations in the presence of 20 ng/µl RNA or without RNA. Experiment was replicated 2 times with similar results. (**e**) Quantification of the relative amount of condensed protein of 5′FAM-HMGB1-IDR peptides at the indicated concentrations. Data displayed as mean ± SD from 5 image fields examined per condition. (**f**) Representative images from droplet formation assays performed with 2.5 µM 5′FAM-labeled HMGB1-IDR synthetic peptides at indicated RNA concentrations. Experiment was replicated 2 times with similar results. (**g**) Quantification of the relative amount of condensed protein of 2.5 µM 5′FAM-HMGB1-IDR peptides at the indicated RNA concentrations. Data displayed as mean ± SD from 5 image fields examined per condition. (**h**) Scheme of co-droplet assays. (**i**) (top) Representative images of droplets formed by 5′FAM-HMGB1-IDR peptide mixed with pre-assembled mCherry-labeled MED1-IDR, HP1α or NPM1 droplets. (bottom) Example images from 5 µM 5′FAM-HMGB1 mutant peptide mixed with mCherry labeled NPM1 droplets with uneven distribution of the peptide within droplet. (**j**) Dual fluorescence plot quantification of 5′FAM and mCherry fluorescence intensities in mCherry-labeled NPM1, MED1-IDR or HP1α droplets mixed with synthetic HMGB1 IDR peptides. Each dot represents one droplet, and the size of the dot is proportional to the size of the droplet. Scale bars, 5 µm (**d,f**) or 1 µm (**g,i**).

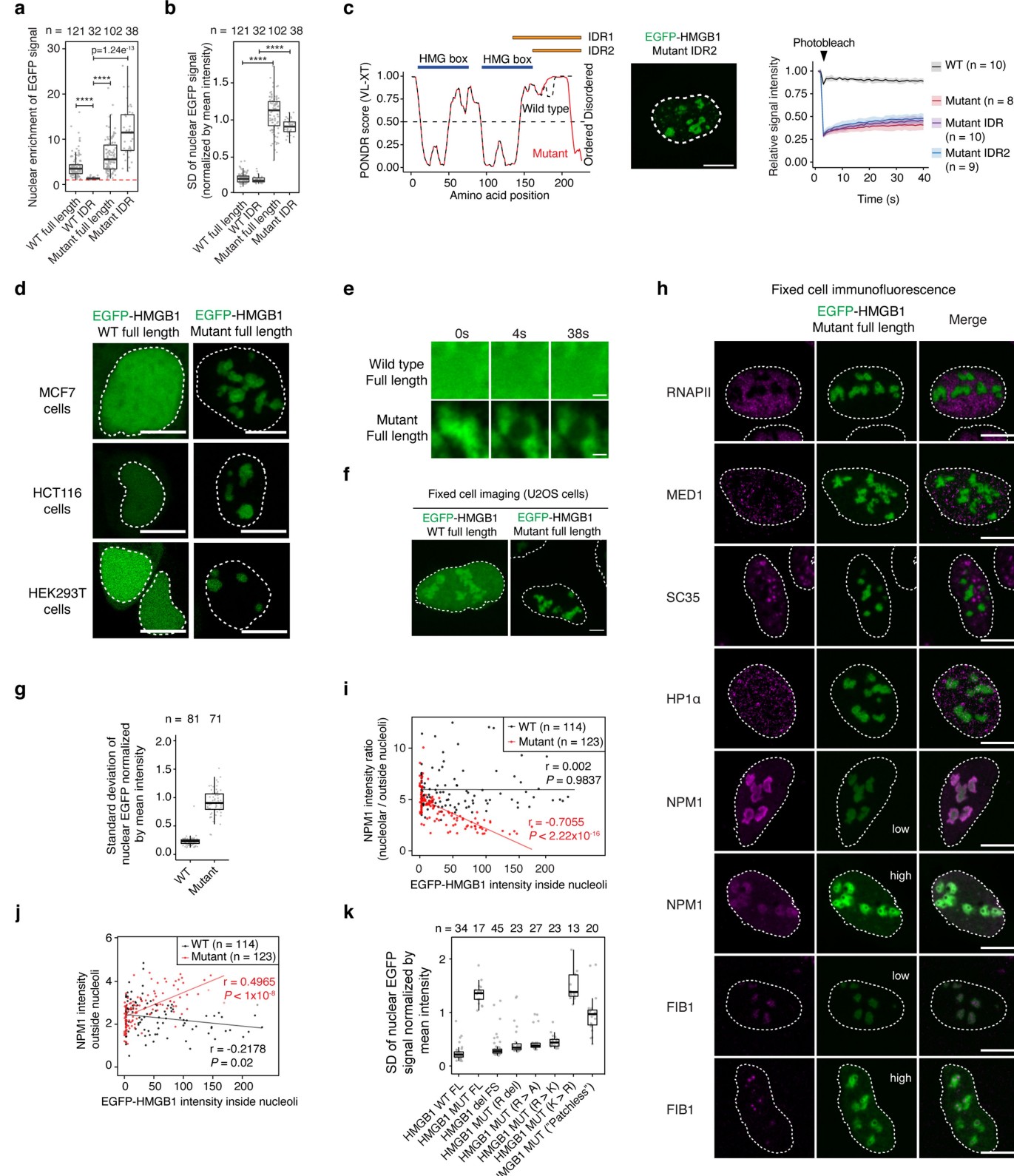

**Extended Data Fig. 5** | See next page for caption.

**Extended Data Fig. 5 | Mutant HMGB1 forms nuclear inclusions in human cells.** (**a**) Quantification of the nuclear enrichment of EGFP as the ratio of mean signal intensities inside and outside the nucleus. Red dashed line depicts a value of 1 (no enrichment). For all boxplots in this figure, the median is shown as a line within the boxplot, which spans from 25th to 75th percentiles. Whiskers depict a 1.5x interquartile range. **** $P < 2.2 \times 10^{-16}$, two-tailed Welch's t-test. (**b**) Quantification of nuclear inclusions as the standard deviation of nuclear EGFP fluorescence intensity normalized by mean intensity. **** $P < 2.2 \times 10^{-16}$, two-tailed Welch's t-test. (**c**) (left) Graph plotting the intrinsic disorder of HMGB1 predicted by PONDR VLXT algorithm. The positions two different IDR definitions are highlighted with orange bars and the position of HMG boxes with blue bars. IDR1 begins from Asn135 as defined by PONDR analysis and IDR2 begins from Ala164, excluding any sequence belonging to HMG box. (middle) Representative image from U2OS cells expressing EGFP-HMGB1-mutant-IDR2. Scale bar = 10 μm. (right) Relative fluorescence intensity of bleached EGFP-HMGB1 WT full-length, mutant full length and mutant IDRs 1 and 2 before and after photobleaching. Data is displayed with a line showing the mean and lighter shade represents ± SD. (**d**) Representative images of live MCF7, HCT116 and HEK293T cells expressing mEGFP-HMGB1 variants. The nuclear area is shown as dashed white lines. Scale bar = 10 μm. (**e**) Representative images of EGFP-HMGB1 within live U2OS cell nuclei before and after photobleaching. FRAP recovery quantified in (Fig. 3d). Scale bar = 1 μm. (**f**) Representative images of formaldehyde-fixed U2OS cells ectopically expressing EGFP-HMGB1 WT or mutant proteins. Scale bar = 5 μm. (**g**) Quantification of presence of nuclear inclusions in fixed cells in panel (f) represented as the standard deviation of nuclear EGFP fluorescence intensity normalized by mean intensity. (**h**) Immunofluorescence for RNAPII, MED1, SC35, HP1α, NPM1 and FIB1 in U2OS cells expressing full length mutant EGFP-HMGB1. (low) indicates a nucleus with a relatively low amount of the mutant protein, (high) indicates a nucleus with a relatively high amount of the mutant protein. Scale bar = 10 μm. (**i**) Quantification of the ratio between intra- /extranucleolar NPM1 intensity and EGFP-HMGB1 intensity inside nucleoli. $r$ = Pearson's correlation coefficient, $P$-value from a two-tailed t-test. (**j**) Quantification of the average NPM1 fluorescence outside the nucleoli and EGFP-HMGB1 intensity inside the nucleoli for the IF experiments shown in panel (h). $r$ = Pearson's correlation coefficient, p-value from a two-tailed t-test. (**k**) Quantification of nuclear inclusions in the panel of EGFP-HMGB1 mutants (Fig. 3e–g) as the standard deviation (SD) of nuclear EGFP signal normalized to the mean nuclear EGFP signal intensity. FL = full length.

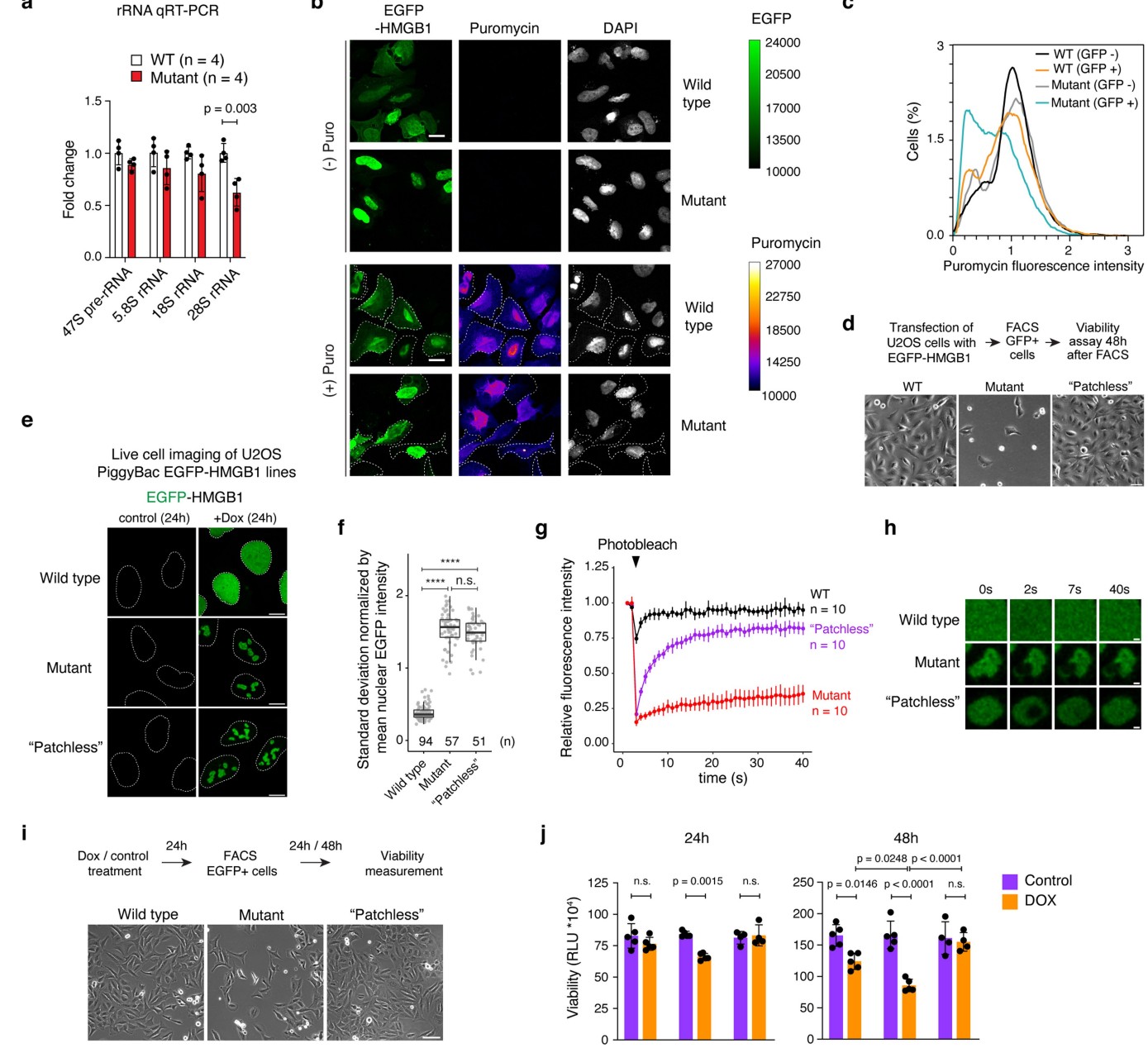

**Extended Data Fig. 6 | Additional characterization of HMGB1 mutant nuclear inclusions in U2OS cells.** (**a**) qRT-PCR analysis of rRNA species in U2OS cells expressing wild type and mutant HMGB1 variants. rRNA levels are normalized against an RNAPII transcript (*actin*), and the values are normalized against the rRNA/*actin* level measured in the cells expressing wild type HMGB1. Data is shown as mean +/− SD, *p < 0.05, two-tailed Student's t-test. (**b**) Representative images from Puromycylation experiments with U2OS cells ectopically expressing EGFP-tagged WT or mutant HMGB1 proteins with (Puro +) and without (Puro -) Puromycin pulse labeling. Scale bar = 20 μm. (**c**) Histograms depicting % of cells and their normalized puromycin intensities from EGFP+ and EGFP- cells ectopically expressing WT or Mutant full length HMGB1 combined from three independent puromycylation experiments. (**d**) (top) Scheme of the viability experiment. (bottom) Representative images of U2OS cells at the end of viability experiments (Fig. 3j). Scale bar = 50 μm. (**e**) Representative live cell imaging of U2OS cells with doxycycline inducible overexpression of EGFP-HMGB1 variants. Dashed lines show nuclear area defined by Hoechst staining. Scale bar = 10 μm.

(**f**) Quantification of nuclear inclusions as the standard deviation (SD) of nuclear EGFP signal normalized to the mean nuclear EGFP signal intensity. ****$P < 2.2 \times 10^{-16}$, two-tailed Welch's t-test, n = number of nuclei examined for each condition. (**g**) Relative fluorescence intensity of EGFP-HMGB1 before and after photobleaching with identical laser settings in cells described in (panels e-f). Data displayed as mean ± SD. (**h**) Representative images of EGFP-HMGB1 within live U2OS cell nuclei before and after photobleaching with identical laser settings, FRAP recovery quantified in (panel g). Scale bar = 2 μm. (**i**) (top) Scheme for experiments testing the viability of U2OS cells with Doxycycline-inducible expression of HMGB1 variants. (bottom) representative images from viability experiments 48h after sorting for GFP+ cells. Scale bar = 100 μm. (**j**) Quantification of viability of cells expressing the indicated HMGB1 proteins. Mean relative light units (RLU) displayed as individual points from independent biological replicate experiments (n = 5 for WT and Mutant, 4 for "Patchless"). Bar charts show the mean ± SD. p-values are from one-way ANOVA.

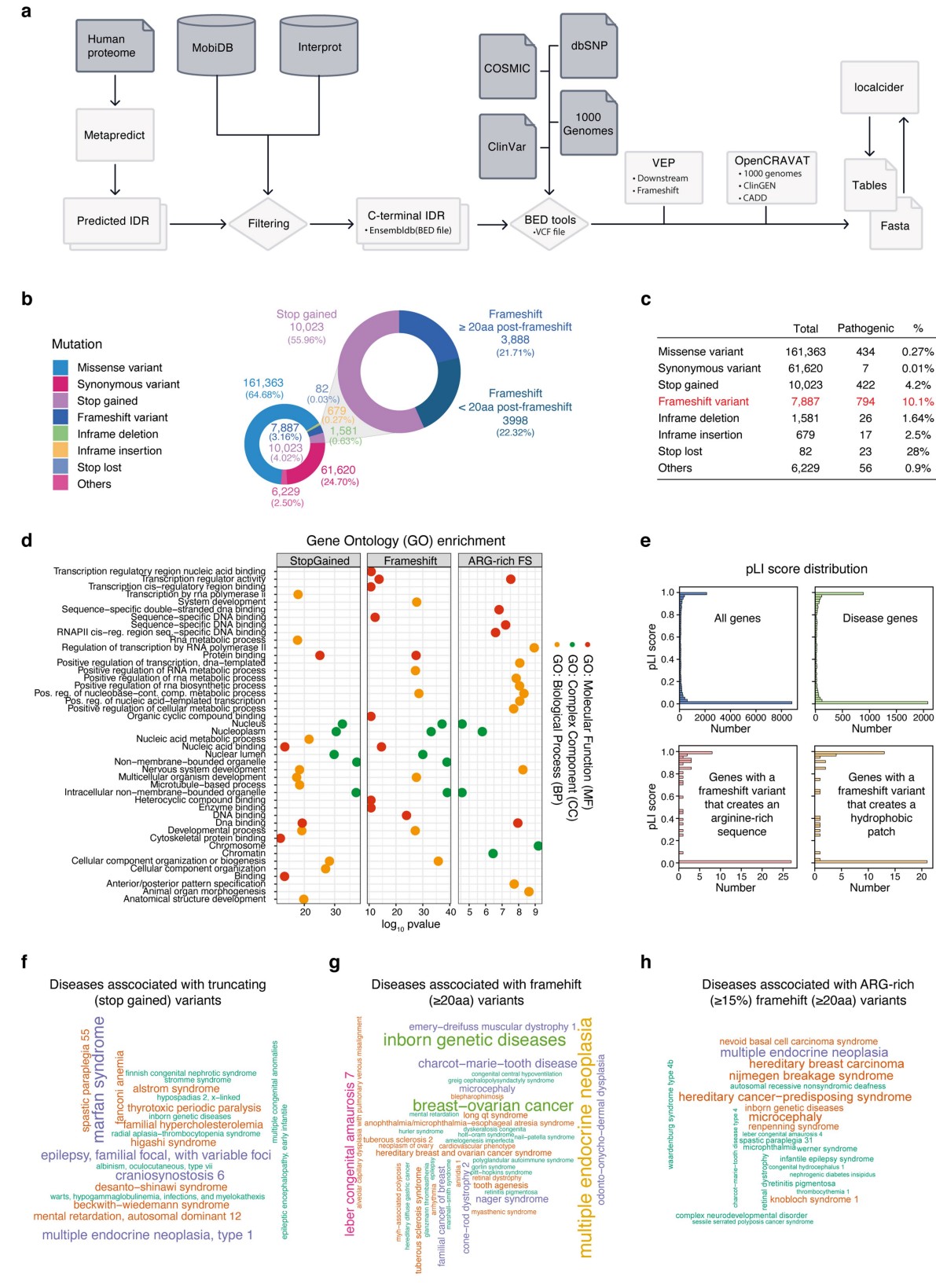

**Extended Data Fig. 7 |** See next page for caption.

**Extended Data Fig. 7 | Characterization of genetic variants in C-terminal IDRs.** (**a**) Scheme of the IDR catalog identification algorithm. (**b**) Summary of all variants identified in C-terminal IDRs. (**c**) Frameshift variants are enriched for pathogenic variants. (**d**) Gene Ontology (GO) term enrichment analysis of (left) genes that contain at least one 'stop gained' (i.e. truncating) mutation in the catalog; (middle) genes that contain at least one frameshift (≥20 amino acids) in the catalog; (right) genes that contain at least one frameshift (≥20 amino acids) that creates a sequence consisting of at least 15% arginines. (**e**) pLI score distributions for indicated gene sets. Disease genes: genes that have at least one "pathogenic", "likely pathogenic", or "conflicting interpretations" entry in ClinVar. (**f**) Word cloud plot of diseases associated with 'stop gained' (i.e. truncating) variants. Font size of words correlates with frequency of occurrence. (**g**) Word cloud plot of diseases associated with frameshift variants that create an at least 20 amino acid long sequence. Font size of words correlates with frequency of occurrence. (**h**) Word cloud plot of diseases associated with frameshift variants that create an at least 20 amino acid long sequence that consists of at least 15% arginines. Font size of words correlates with frequency of occurrence.

**>NM_005342.4(HMGB3**
KSKGKFDGAKGPAKVARKKVEEEDEEEEEEEEEEEEEDE

**>NM_005342.4(HMGB3)c.480_481dup**
ISRKESLMVQRVLLKLPGKRWKRKMKKRRRKKRRRRRRMNKETVYLSPCEYLE

**>NM_001453.3(FOXC1)**
QPPPAPPEQADGNAPGPQPPPVRIQDIKTENGTCPSPPQPLSPAAALGSGSAAAVPKIESPDSSSSSLSSGSSPPGSLPSARPLSLDGADSAPPPPAPSAPPPHHSQGFSVDNIMTSLRGSPQSA
AAELSSGLLASAAASSRAGIAPPLALGAYSPGQSSLYSSPCSQTSSAGSSGGGGGGAGAAGGAGGAGTYHCNLQAMSLYAAGERGGHLQGAPGGAGGSAVDDPLPDYSLPPVTSSSSSSLSHGGG
GGGGGGGQEAGHHPAAHQGRLTSWYLNQAGGDLGHLASAAAAAAAAGYPGQQQNFHSVREMFESQRIGLNNSPVNGNSSCQMAFPSSQSLYRTSGAFVYDCSKF

**>NM_001453.3(FOXC1):c.599_617del**
RSRPTATRPVRSRRPCASRTSRPRTVRAPRRPSPCPRPPPWAAAAPPRCPRSRAPTAAAAACPAGAAPRAACRRRGRSAWTVRIPRRRRPRPPPRRRTIARASAWTTS

**>NM_001451.3(FOXF1)**
AAAGEYPHHDSSVPASPLLPTGAGGVMEPHAVYSGSAAAWPPSASAALNSGASYIKQQPLSPCNPAANPLSGSLSTHSLEQPYLHQNSHNAPAELQGIPRYHSQSPSMCDRKEFVFSFNAMASSS
MHSAGGGSYYHQQVTYQDIKPCVM

**>NM_001451.3(FOXF1):c.691_698del**
RRVPAPRQLGARLPAAAHRRRWGHGAARRLLGLGGGLAALGVRGAQQRRLLYQAAAPVPL

**>NM_002478.5(MYOD1)**
RGGEHYSGDSDASSPRSNCSDGMMDYSGPPSGARRRNCYEGAYYNEAPSEPRPGKSAAVSSLDCLSSIVERISTESPAAPALLLADVPSESPPRRQEAAAPSEGESSGDPTQSPDAAPQCPAGAN
PNPIYQVL

**>NM_002478.5(MYOD1):c.557dup**
PRRRALQRRLRRVQPALQLLRRHDGLQRPPERRPAAELLRRRLLQRGAQRTQAREECGGVEPRLPVQHRGAHLHREPCGARPPAGGRAF

**>NM_013435.3(RAX)**
SPLGAGPGSGGGPAGGALPLESWLGPPLPGGGATALQSLPGFGPPAQSLPASYTPPPPPPPFLNSPPLGPGLQPLAPPPPSYPCGPGFGDKFPLDEADPRNSSIAALRLKAKEHIQAIGKPWQAL

**>NM_013435.3(RAX):c.664del**
RPSGRARAAVAGRLGARCRWSPGSGRRCRAGAPRRCRACRASGRRRRACLPATRHRRRLRPS

**>NM_001754.5(RUNX1)**
GIGIGMSAMGSATRYHTYLPPPYPGSSQAQGGPFQASSPSYHLYYGASAGSYQFSMVGGERSPPRILPPCTNASTGSALLNPSLPNQSDVVEAEGSHSNSPTNMAPSARLEEAVWRPY

**>NM_001754.5(RUNX1):c.1088_1094del**
ASACRPWARPRATTPTCRRPTPARRKRREARSKPARPPTTCTTAPRPAPTSSPWWAASARRRASCRPAPTPPPAPRCSTPASRTRATWWRPRAATATPPPTWRPPRAWRRPCGGPTEAPGLARLG
PAGRRLRLRARGPPVRDKPAGIPGPGPGHRPGAEGARRPGSRCRSGPRSLLRPEAHAAAVCWRPGPRGGVRGDAPRGCPPAPAPRGRAGKQTGRFPEGNCECF

**>NM_003924.4(PHOX2B):**
SCGANGGGGGGPSPAGAPGAAGPGGPGGEPGKGGAAAAAAAAAAAAAAAAAAAAAGGLAAAGGPGQGWAPGPGPITSIPDSLGGPFASVLSSLQRPNGAKAALVKSSMF

**>NM_003924.4(PHOX2B):c.618del**
AAGRMEAAAAGPARLELRGRRGPGAREANPARAAQQQRRRPRQRRRRQRQRRQLEAWLRLGALDKAGLPAPAPSPPSRIRLGVPSPASYLRSKDPTVPKPP

**>NM_004343.4(CALR)**
DDEDKDEDEEDEEDKEEDEEEDVPGQAKDEL

**>NM_004343.4(CALR):c.1157_1158dup**
RMMRTKMRMRRMRRTRRKMRRKMSPARPRTSCREACLQGWTEA

**>NM_023067.4(FOXL2)**
ATAAPPAPAPTSAPGLQFACARQPELAMMHCSYWDHDSKTGALHSRLDL

**>NM_023067.4(FOXL2):c.982del**
PPPRPRRPRPPVRRACSSLVPGSPSSP

**>NM_003106.4(SOX2)**
MYLPGAEVPEPAAPSRLHMSQHYQSGPVPGTAINGTLPLSHM

**>NM_003106.4(SOX2):c.828del**
IISPAPRCRNPPPPADFTCPSTTRAARCPARPLTAHCPSHTCEGRTANWRGEKFSKKNEGNGRGAKEESKKQHGENPVRSKRKRKKKNPITHSK

**>NM_003900.5(SQSTM1)**
VSPESSSTEEKSSSQPSSCCSDPSKPGGNVEGATQSLAEQMRKIALESEGRPEEQMESDNCSGGDDDWTHLSSKEVDPSTGELQSLQMPESEGPSSLDPSQEGPTGLKEAALYPHLPPEADPRLI
ESLSQMLSMGFSDEGGWLTRLLQTKNYDIGAALDTIQYSKHPPPL

**>NM_003900.5(SQSTM1):c.810del**
SLQRVPAQRRRAAHSQAAAALTPASRVGMLRAPRSLWRSR

**>NM_001370259.2(MEN1)**
AAEAEEPWGEEAREGRRRGPRRESKPEEPPPPKKPALDKGLGTGQGAVSGPPRKPPGTVAGTARGPEGGSTAQVPAPTASPPPEGPVLTFQSEKMKGMKELLVATKINSSAIKLQLTAQSQVQMK
KQKVSTPSDYTLSFLKRQRKGL

**>NM_001370259.2(MEN1):c.1382_1389dup**
RPRRPRPRSRGARKPGKAGGGAHGGSPSQRSPRRPRSQHWTRAWAPARVQCQDPPGSLLGLSLAQPEALKVAARLRCQHPQHHHRRRVQCSLSRVRR

**>NM_004421.2(DVL1)**
HPAAPWPLGQGYPYQYPGPPPCFPPAYQDPGFSYGSGSTGSQQSEGSKSSGSTRSSRRAPGREKERRAAGAGGSGSESDHTAPSGVGSSWRERPAGQLSRGSSPRSQASATAPGLPPPHPTTKAY
TVVGGPPGGPPVRELAAVPPELTGSRQSFQKAMGNPCEFFVDIM

**>NM_004421.2(DVL1):c.1505_1517del**
PGLWVRATPTSTRDPHPASRLPTRTRALAMAAAAPGVSRVKGAKAVGPPGAAAGPRAVRRSVGRRELGAVAVNRITRHRVGWGAAGESVRPASSAVAAAHAVRPRLPPRGSPRPTPRPRPIQWWG
GHPGDPLSGSWLPSPRN

**Extended Data Fig. 8 | Sequences of candidate proteins.** Sequences of the thirteen wild type and mutant HMGB3, FOXC1, FOXF1, MYOD1, RAX, RUNX1, PHOX2B, CALR, FOXL2, SOX2, SQSTM1, MEN1 and DVL1 proteins. In the wild type variants, only the sequences replaced by the frameshift variants are shown underlined. The sequences created by the frameshift variants are colored red and are underlined.

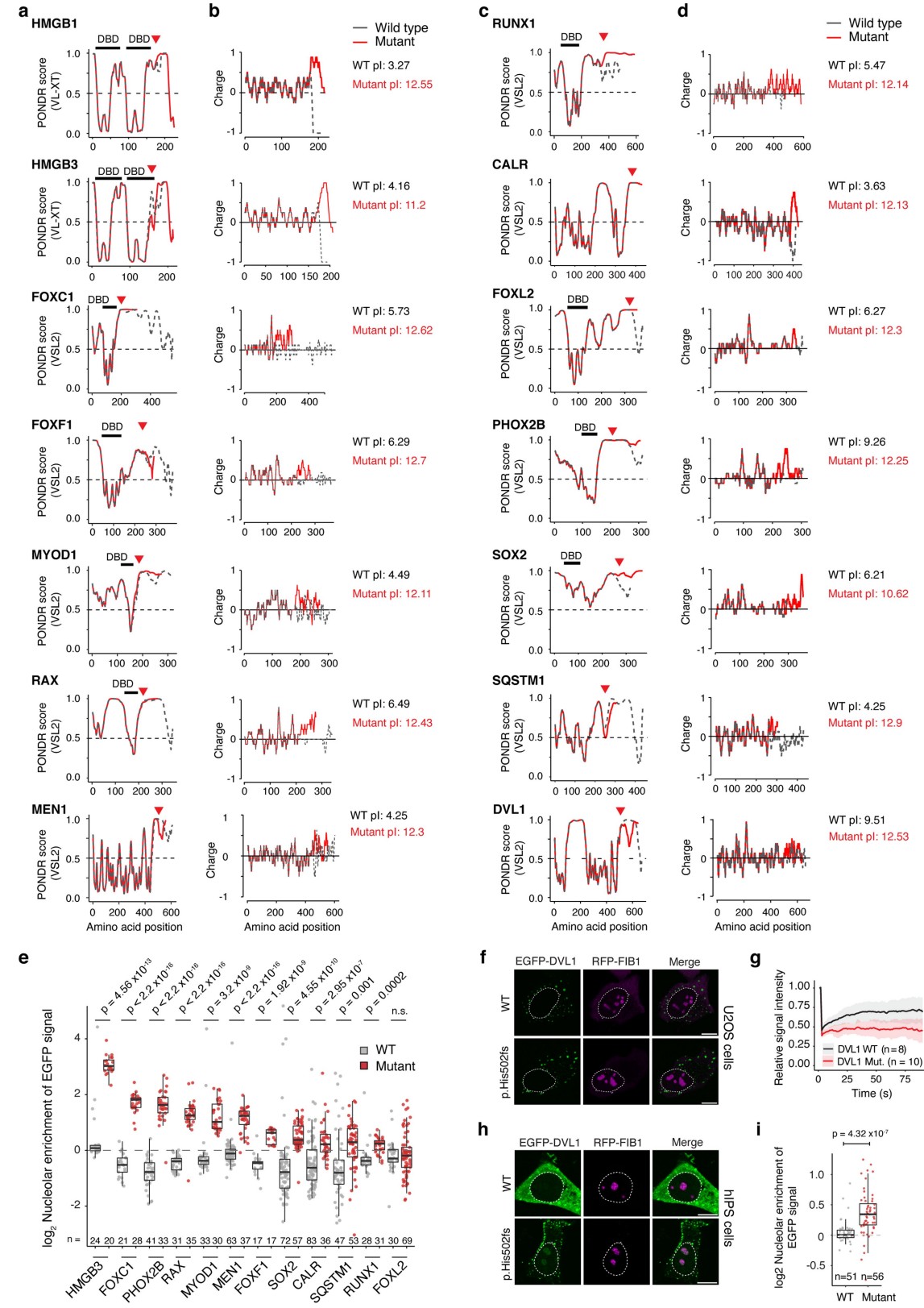

**Extended Data Fig. 9** | See next page for caption.

**Extended Data Fig. 9 | Disorder and charge analyses of proteins created by frameshifts in candidate proteins.** (**a**, **c**) Disorder analysis of HMGB1, HMGB3, FOXC1, FOXF1, MYOD1, RAX, RUNX1, CALR, FOXL2, PHOX2B, SOX2 and SQSTM1 wild type and frameshift mutant sequences using the PONDR algorithm. The PONDR scores for the wild type sequences are plotted with grey dashed line, the PONDR scores for the mutant sequences are plotted in red. The positions of the DNA binding domains (DBD) are highlighted with black bars and frameshift position is highlighted with red arrow. (**b**, **d**) Charge plots of wild type and mutant sequences. Note the increased positive charge in C-terminus of frameshift variants. Isoelectric points (pI) for the protein sequence following the frameshift position in wild type and mutant sequences are shown beside the charge plots. (**e**) Quantification of nucleolar enrichment of the indicated proteins in the FIB1-RFP co-expression experiments. Median is shown as a line within the boxplot, which spans from 25th to 75th percentiles. Whiskers depict a 1.5x interquartile range. *** $P < 10^{-3}$, **** $P < 10^{-4}$ from two-tailed Welch's t-test, n = number of nuclei examined per condition. (**f**) Representative images of U2OS cells co-expressing RFP-Fibrillarin and EGFP-tagged DVL1 proteins. Scale bar = 10 μm. (**g**) Relative fluorescence intensity of bleached EGFP-tagged DVL1 in U2OS cells before and after photobleaching with identical laser settings. Line: mean, lighter shade: ± SD. (**h**) Representative images of hiPSCs co-expressing RFP-Fibrillarin and EGFP-tagged DVL1 proteins. Note the nucleolar signal in the cells expressing Mutant EGFP-DVL1. Scale bar = 10 μm. (**i**) Quantification of nucleolar enrichment of DVL1 in the FIB1-RFP co-expression experiments in hiPSCs. Median is shown as a line within the boxplot, which spans from 25th to 75th percentiles. Whiskers depict a 1.5x interquartile range, **** $P < 10^{-4}$, two-tailed Welch's t-test, n = number of nuclei examined per condition.

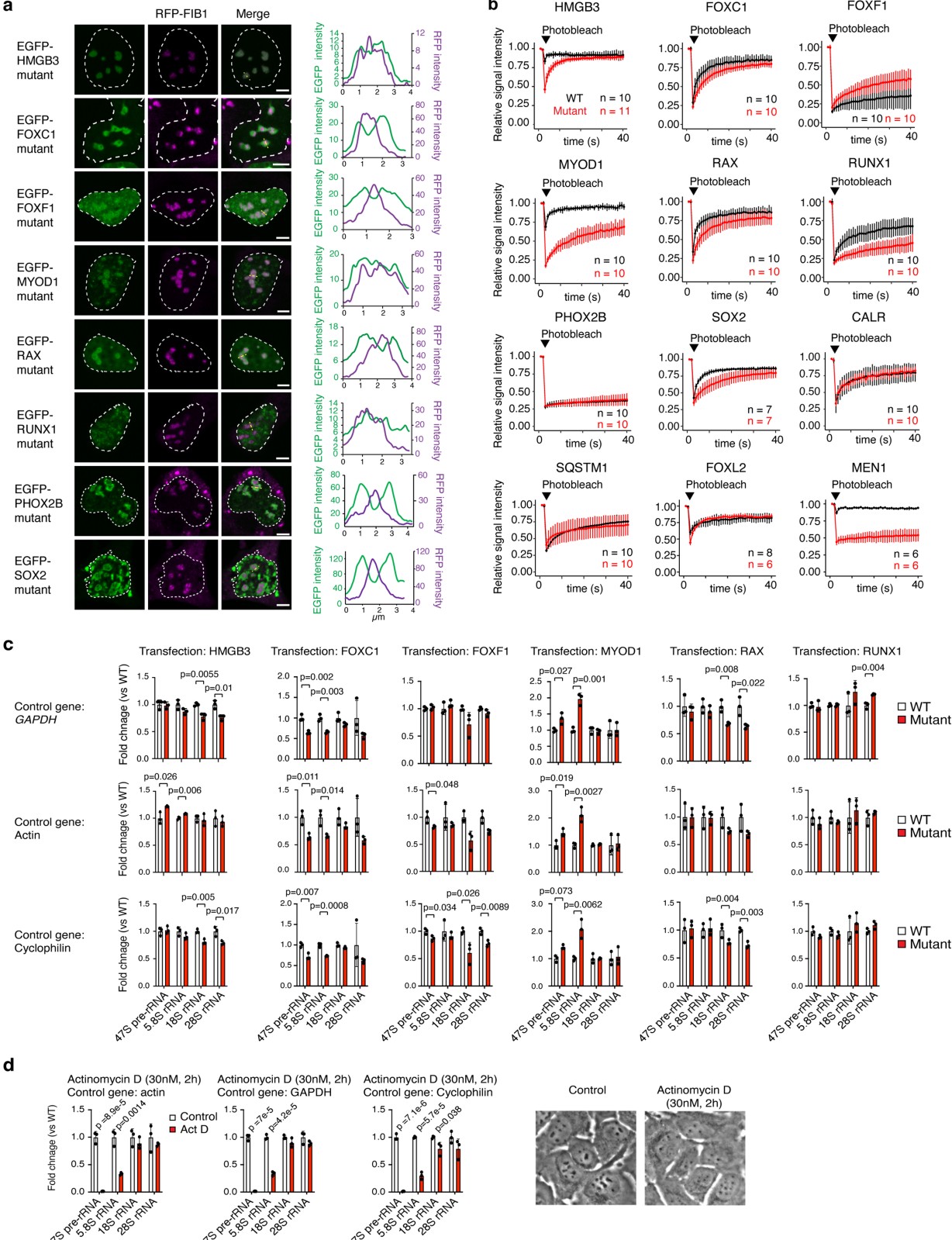

**Extended Data Fig. 10** | See next page for caption.

**Extended Data Fig. 10 | Nucleolar mispartitioning and dysfunction in cells expressing mutant proteins with disease-associated frameshifts.** (**a**) Cavitation of the nuclear inclusion formed by frameshift mutant FOXC1, FOXF1, RAX, HMGB3, PHOX2B and SOX2. Representative images of U2OS cells co-expressing RFP-Fibrillarin and EGFP-tagged mutant proteins are shown. Scale bar = 5 µm. Right: fluorescence intensity profiles of EGFP (green) and RFP (purple) quantified from the region highlighted with yellow a dashed line on images on the left. (**b**) Relative fluorescence intensity of bleached EGFP-tagged proteins before and after photobleaching with identical laser settings. Data displayed as mean ± SD. (**c**) qRT-PCR analysis of rRNA species in U2OS cells expressing the indicated wild type and mutant proteins. rRNA levels are normalized against a control RNAPII transcript (*GAPDH, actin, or Cyclophilin*), and the values are normalized against the rRNA/*control transcript* level measured in the cells expressing wild type proteins. Data displayed as mean +/− SD. *P*-values are from two-tailed Student's t-tests, n = 3 biologically independent experiments. (**d**) Actinomycin D control experiments for measuring rRNA levels. U2OS cells were treated with Actinomycin D (30 nM for 2 h), and rRNA levels were quantified using qRT-PCR. rRNA levels are normalized against a control RNAPII transcript (*actin, GAPDH* or *Cyclophilin*), and the values are normalized against the rRNA/*control transcript* level measured in the control (i.e. untreated) cells. Data displayed as mean +/− SD. *P*-values are from two-tailed Welch's t-test, n = 3 biologically independent experiments. On the right, light microscopy images of control and Actinomycin D-treated cells are shown. Note the dimming of the dark spots in the nucleus (corresponding to nucleoli).

# Reporting Summary

## Statistics

For all statistical analyses, confirm that the following items are present in the figure legend, table legend, main text, or Methods section.

| n/a | Confirmed | |
|---|---|---|
| ☐ | ☒ | The exact sample size (*n*) for each experimental group/condition, given as a discrete number and unit of measurement |
| ☒ | ☐ | A statement on whether measurements were taken from distinct samples or whether the same sample was measured repeatedly |
| ☐ | ☒ | The statistical test(s) used AND whether they are one- or two-sided<br>*Only common tests should be described solely by name; describe more complex techniques in the Methods section.* |
| ☒ | ☐ | A description of all covariates tested |
| ☒ | ☐ | A description of any assumptions or corrections, such as tests of normality and adjustment for multiple comparisons |
| ☐ | ☒ | A full description of the statistical parameters including central tendency (e.g. means) or other basic estimates (e.g. regression coefficient) AND variation (e.g. standard deviation) or associated estimates of uncertainty (e.g. confidence intervals) |
| ☐ | ☒ | For null hypothesis testing, the test statistic (e.g. *F*, *t*, *r*) with confidence intervals, effect sizes, degrees of freedom and *P* value noted<br>*Give P values as exact values whenever suitable.* |
| ☒ | ☐ | For Bayesian analysis, information on the choice of priors and Markov chain Monte Carlo settings |
| ☒ | ☐ | For hierarchical and complex designs, identification of the appropriate level for tests and full reporting of outcomes |
| ☐ | ☒ | Estimates of effect sizes (e.g. Cohen's *d*, Pearson's *r*), indicating how they were calculated |

*Our web collection on statistics for biologists contains articles on many of the points above.*

## Software and code

Policy information about availability of computer code

| | |
|---|---|
| Data collection | Data from fluorescence microscopy was acquired with Zen Black v3.2 (Zeiss). |
| Data analysis | Facial phenotyping and composite mask creation was performed using Face2Gene v.20.1.4 (face2gene.com). Microscopy data was analyzed using ZenBlue 3.2, except v 3.4 was used in puromycilation experiment, as described in the methods. Protein structure predictions were performed using in-house implementation of AlphaFold 2.0.0. Visualization with ChimeraX 1.5.<br><br>Following tools were used in building variant catalogue: Metapredict v1.51, Ensembldb v2.22.0, BEDtools v2.30.0, VEP v104, CADD v1.6, pVACtools v3.1.0, localCIDER 0.1.18, biopython v1.79, NoD v1.0.0<br><br>Custom code is available at GitHub:<br>https://github.com/hniszlab/HMGB1_2022<br>https://github.com/alexpmagalhaes/IDR-variant-catalog |

For manuscripts utilizing custom algorithms or software that are central to the research but not yet described in published literature, software must be made available to editors and reviewers. We strongly encourage code deposition in a community repository (e.g. GitHub). See the Nature Portfolio guidelines for submitting code & software for further information.

## Data

Policy information about availability of data

All manuscripts must include a data availability statement. This statement should provide the following information, where applicable:

- Accession codes, unique identifiers, or web links for publicly available datasets
- A description of any restrictions on data availability
- For clinical datasets or third party data, please ensure that the statement adheres to our policy

Patient consent did not cover the release of personal sequence information other than the causative pathogenic variants. Therefore, the pathogenic variants are disclosed in this paper, but whole genome and exome sequencing data cannot be made publicly available for reasons of data protection and patient privacy. Statistical source data are made available with this paper. Primer sequences are listed in Supplementary Table 6. Plasmids are available at Addgene. Circular Dichroism (CD) spectra were deposited at the PCDDB under the accession IDs: CD0006401000, CD0006401001, CD0006404000, CD0006404001.

Following databases were used in building the variant catalogue:
GENCODE GRCh38.p13 Release 41 https://www.gencodegenes.org/human/
MobiDB 4.1.0 https://mobidb.bio.unipd.it/
Ensembldb v104 https://bioconductor.org/packages/release/bioc/html/ensembldb.html
Clinvar 1.64 https://www.ncbi.nlm.nih.gov/clinvar/
COSMIC v95 https://cancer.sanger.ac.uk/cosmic
dbDNP from May 26, 2020 https://www.ncbi.nlm.nih.gov/snp/
1000 genomes from 2021-11-20 https://www.internationalgenome.org/data

# Field-specific reporting

Please select the one below that is the best fit for your research. If you are not sure, read the appropriate sections before making your selection.

☒ Life sciences  ☐ Behavioural & social sciences  ☐ Ecological, evolutionary & environmental sciences

For a reference copy of the document with all sections, see nature.com/documents/nr-reporting-summary-flat.pdf

# Life sciences study design

All studies must disclose on these points even when the disclosure is negative.

| | |
|---|---|
| Sample size | Required family number for new disease gene establishment is 2, number of unrelated families included in our studies is 5.<br><br>No statistical methods were used to predetermine sample sizes. Sample sizes are indicated in figures, legends or in the methods. For droplet assays, 5 − 10 independent fields of view were imaged per condition for each experiment based on current practices in the field (Sabari et al. 2018, Boija et al. 2018) and performed independently total 3 times. |
| Data exclusions | No data were excluded |
| Replication | Genetic test results were confirmed by an independent method (WGS by bidirectional Sanger sequencing; chromosomal microarray analysis by RT-qPCR). Genetic tests were also performed on unaffected controls.<br><br>Results from biochemistry and cellular experiments were replicated across multiple experiments with similar results. Numbers of replicates are indicated in figures, figure legends or in the methods. |
| Randomization | N.a. for participants. In cell culture, wells were randomly assigned for transfections and control or treatment groups. For microscopy, image fields were randomly selected while avoiding cells with highest expression to avoid saturation of EGFP channel. |
| Blinding | Blinding of patient phenotypical data is impossible, as this is required for genome data interpretation and blinding was not performed. Investigators were not blinded during data acquisition from biochemistry and cellular experiments. Differences between wild type and mutant samples were so apparent in microscopy experiments that blinding would not have been feasible. Analytical pipelines for experiments were uniform across samples, allowing unbiased analysis of data. |

# Reporting for specific materials, systems and methods

We require information from authors about some types of materials, experimental systems and methods used in many studies. Here, indicate whether each material, system or method listed is relevant to your study. If you are not sure if a list item applies to your research, read the appropriate section before selecting a response.

## Materials & experimental systems

| n/a | Involved in the study |
|---|---|
| ☐ | ☒ Antibodies |
| ☐ | ☒ Eukaryotic cell lines |
| ☒ | ☐ Palaeontology and archaeology |
| ☒ | ☐ Animals and other organisms |
| ☐ | ☒ Human research participants |
| ☒ | ☐ Clinical data |
| ☒ | ☐ Dual use research of concern |

## Methods

| n/a | Involved in the study |
|---|---|
| ☒ | ☐ ChIP-seq |
| ☐ | ☒ Flow cytometry |
| ☒ | ☐ MRI-based neuroimaging |

# Antibodies

| | |
|---|---|
| Antibodies used | HP1a (1:500, Cell signaling Cat# 2616S) MED1 (1:500, Abcam Cat# ab64965) RNAPII (1:500, Abcam, Cat# ab26721) NPM1 (1:250, Thermo Fisher Scientific, Cat# 32-5200) FIB1 (1:100, Santa Cruz Cat# sc-374022) SC-35 (1:200, Sigma-Aldrich Cat# S4045), HMGB1 (1:1000 Sigma-Aldrich Cat# H9664), EGFP (1:1000, Invitrogen Cat# A-11122), anti-Puromycin (1:1000, Sigma Aldrich MABE343), anti-GFP (1:2000, Abcam ab13970), Alexa Fluor 647 donkey anti-mouse (1:1000, Jackson Immuno Research, Cat# 715-605-150), Alexa Fluor 647 donkey anti-rabbit (1:1000, Jackson Immuno Research, Cat# 711-605-152), HRP-Donkey anti-Rabbit IgG (1:2000, Jackson Immuno Research, Cat# 711-035-152), 488-anti-chicken (1:250, Jackson Immuno Research, Cat# 703-545-155), 647-anti-mouse (1:250, Jackson Immuno Research, Cat #715-605-151) |
| Validation | All antibodies are verified by manufacturer, described to function in human cells in intended applications (immunofluorescence or western blot) and used in numerous publications. Lists for publications are available on manufacturer's websites:<br><br>HP1a: https://www.cellsignal.com/products/primary-antibodies/hp1a-antibody/2616<br>MED1: https://www.abcam.com/trap220med1-antibody-ab64965.html<br>RNAPII: https://www.abcam.com/rna-polymerase-ii-ctd-repeat-ysptsps-antibody-chip-grade-ab26721.html<br>NPM1: https://www.thermofisher.com/antibody/product/NPM1-Antibody-clone-FC-61991-Monoclonal/32-5200<br>FIB1: https://www.scbt.com/p/fibrillarin-antibody-g-8<br>SC-35: https://www.sigmaaldrich.com/DE/en/product/sigma/s4045<br>HMGB1: https://www.sigmaaldrich.com/DE/en/product/sigma/h9664<br>EGFP: https://www.thermofisher.com/antibody/product/GFP-Antibody-Polyclonal/A-11122<br>Puromycin: https://www.sigmaaldrich.com/DE/en/product/mm/mabe343<br>GFP: https://www.abcam.com/gfp-antibody-ab13970.html<br><br>Statement for knock-out verification for NPM1 antibody is provided on manufacturer's website: "This Antibody was verified by Knockdown to ensure that the antibody binds to the antigen stated.":<br>https://www.thermofisher.com/antibody/product/NPM1-Antibody-clone-FC-61991-Monoclonal/32-5200<br><br>HMGB1 antibody is provided with validation statement (Enhanced validation, independent):<br>https://www.sigmaaldrich.com/DE/en/product/sigma/h9664<br><br>https://www.sigmaaldrich.com/DE/en/technical-documents/technical-article/protein-biology/immunohistochemistry/antibody-enhanced-validation<br><br>EGFP antibody is provided with Advanced verification statement: "This Antibody was verified by Relative expression to ensure that the antibody binds to the antigen stated."<br>https://www.thermofisher.com/antibody/product/GFP-Antibody-Polyclonal/A-11122<br><br>No in-house validations for antibodies were performed. |

# Eukaryotic cell lines

Policy information about cell lines

| | |
|---|---|
| Cell line source(s) | HEK293T (ATCC CRL-3216), MCF7 (ATCC HTB-22), HCT116 (ATCC CCL-247), U2OS-2-6-3 cell line was received from Richard Young lab (Zamudio et al. 2019), U2OS cells from ATCC (HTB-96) were used for generating Doxycycline-inducible HMGB1-expressing cell lines. |
| Authentication | Identity of cell lines has been validated using morphological characteristics, but they were not authenticated. |
| Mycoplasma contamination | All cell lines were tested negative for mycoplasma using PCR Mycoplasma testkit II (A8994, AppliChem). |
| Commonly misidentified lines (See ICLAC register) | None used. |

# Human research participants

Policy information about <u>studies involving human research participants</u>

| | |
|---|---|
| Population characteristics | 5 participants have (de novo) heterozygous frameshifts in HMGB1's acidic tail, 1 individual has a de novo heterozygous deletion of HMGB1. Individuals originate from different populations (Russian, Iranian, German, Hong-Kong-Chinese, Venezuelan) and are unrelated. 3 are male 3 are female. Ages reach from 21 weeks of gestation to 29 years. 5 individuals are diagnosed with BPTAS, one is diagnosed with neurodevelopmental delay. |
| Recruitment | Individuals were recruited during routine patient care in 5 genetics departments (Berlin, Kiel, Nuremberg, Schwerin, Hong Kong). Potentially affected fetuses could not be systematically screened, thus the frequency and severity spectrum of BPTAS in unborn individuals remains unknown. |
| Ethics oversight | The study was approved by the ethical review board of the Charité - Universitätsmedizin Berlin (EA2/087/15). |

Note that full information on the approval of the study protocol must also be provided in the manuscript.

# Flow Cytometry

## Plots

Confirm that:

☒ The axis labels state the marker and fluorochrome used (e.g. CD4-FITC).

☒ The axis scales are clearly visible. Include numbers along axes only for bottom left plot of group (a 'group' is an analysis of identical markers).

☒ All plots are contour plots with outliers or pseudocolor plots.

☒ A numerical value for number of cells or percentage (with statistics) is provided.

## Methodology

| | |
|---|---|
| Sample preparation | Cells were collected for FACS from 6-well plates with TrypLE (12604013, Gibco), washed once with cell culture medium, once with PBS, then resuspended to FACS buffer (2 % FBS, 0.5 mM EDTA in PBS) and transferred to FACS tube (#352235, Falcon) through 35 μm nylon mesh. |
| Instrument | FACS Aria II Flow Cytometer (BD Biosciences) |
| Software | BD FACS Diva v.6.1.3 |
| Cell population abundance | The percentage of EGFP positive population in transfected cells ranged from 20 to 30 % between replicate experiments. |
| Gating strategy | To select appropriate gating for EGFP positive cells, untransfected cells were used as a negative control. |

☒ Tick this box to confirm that a figure exemplifying the gating strategy is provided in the Supplementary Information.

