## [Peer Review File · Nature]

Manuscript Title: Aberrant phase separation and nucleolar dysfunction in human genetic disease

Reviewer Comments & Author Rebuttals

Reviewer Reports on the Initial Version:

Referees' comments:

Referee #1 (Remarks to the Author):

Mensah et al report a new genetic mutation in the protein HMGB1 that is associated with pathogenicity of the rare disease BPTAS: a frame shift associated with a charge reversal in the C-terminal IDR, from an acidic to a basic, arginine-rich tail. The authors show that the mutant exhibits a lower threshold for phase separation compared to WT and somewhat different partitioning preference when tested in in vitro model systems of condensates. The mutant aberrantly partitions into the nucleolus, perturbs ribosome biogenesis and cell viability. The authors postulate that the aberrant nucleolar localization is driven by charge-charge interactions mediated by the newly acquired arginine-rich disordered tail and that this mechanism might be shared by other genetic translocations. They proceed to search in GWAS data bases and identify frame shifts in other genes that result in Arg-rich C-terminal IDRs. The authors perform an extensive bioinformatics analysis of the identified frameshift mutants and select a set of 6 genes to perform proof of concept cell biology testing.

The concept that positively charged proteins, and in particular proteins rich in arginines partition into the nucleolus is not novel. However, the concept that nucleolar sequestration of multiple genes associated with different pathologies and consequential nuclear functional disruption represents a common hub in the dysregulation is.

The manuscript nicely blends a broad range of techniques, ranging from clinical studies, to bioinformatics, genetics, in vitro biophysics and cell biology to investigate the problem from complementary angles. In general, the manuscript is well written.

However, some of the claims are overreaching, in particular that referring to the generality of the mechanism of nucleolar partitioning to 600+ Arg-rich frame shift mutations, based on an experimental sampling of 6 genes. I think that the manuscript will be valuable to the readers of Nature, once some of the concerns listed below have been addressed:

1. Abstract: "Here we show that disease-associated mutations in disordered regions frequently alter phase separation, cause mispartitioning into the nucleolus, and disrupt nucleolar function." This sentence is overreaching. The authors show for a small number of genes (<1%) that exhibit a frameshift resulting in an Arg-rich C-terminal tail partition in the nucleolus and identify thousands of genes with arguably similar amino-acid syntax. However, given the small number of experimental points, the conclusion of nucleolar partitioning and disruption of nucleolar function are only circumstantial. I suggest softening the claim accordingly.

2. For the in vitro experiments, it is imperative that the recombinant protein be clean of nucleic acids and other impurities, as well as of soluble oligomers, all of which might affect the observed phase behavior and experimental reproducibility. The fact that the authors used a single chromatography step for the purification of all their proteins raises some concerns regarding the level of purity and monodispersity of their material. It would be useful for the authors to show the recombinant protein gels demonstrating the level of purity and discuss the quality control criteria that were used for the characterization of the proteins. In particular, the observation of multi-phase separation in the presence of mutant but not WT HMGB1 could be a result of condensate network rearrangements due to preferential (1) homotypic mutHMGB1 OR (2) heterotypic mutHMGB1 and residual nucleic acids in the recombinant protein prep. The polycationic mutHMGB1 IDR is very likely to be interacting non-specifically to RNA and DNA and even phase separate with nucleic acids in a heterotypic fashion. I suggest the authors perform a control experiment, measuring the Csat of WT vs mutant in the presence of nucleic acids.

3. Pg. 5: "The droplets were spherical in size, settled on the surface (Fig. 2c) and occasionally underwent fusion (not shown), which are hallmarks of phase separation 34". I suggest including a SI video showing droplet fusion instead of the "(not shown)" specification.

4. Pg. 8: Please elaborate a bit more on the justification for why these particular 6 genes were selected for proof-of-concept studies?

5. In addition to the fraction of positive residues and Arg residues in particular, it would be interesting to look at the distribution of these residues (e.g., short Arg clusters/stickers connected by longer linkers vs. long, Arg-dense stickers). For example, C9Orf72 di-peptide repeats, which have been shown to partition into and disrupt nucleolar function exhibit a long and very dense Arg cluster, compared to endogenous nucleolar proteins, where the Arg clusters are shorter and less dense, broken up via linkers. Are the non-Arg linkers enriched in aromatic residues, which will make them stickers of a different type? Do the pathological variants fall into the category of IDRs with densely packed stickers?

6. Pg. 9: "Arginine-rich sequences were previously shown to facilitate nucleolar localization, but native proteins with arginine-rich stretches form monophasic phase-separated droplets with NPM1 in vitro, and do not affect the liquid-like properties of nucleolar condensates 40." Riback et al, Nature 2020 shows that changes in compositional blend of a three component in vitro NPM1 droplet do change the condensate thermodynamics; this is also true for nucleoli in living cells, when the relative levels of nucleolar proteins and rRNA change. While the paper did not directly measure changes in material properties, it is expected that these change with the dense phase thermodynamic stability. Please revise this sentence accordingly.

Referee #2 (Remarks to the Author):

The authors start from the finding that 5 different individual with BPTAS, an ultra-rare syndrome, share frameshift mutations in HMGB1 that turn the acidic tail into a long basic tail. They then show that the mutated HMGB1 is much more prone to phase separation in vitro, either alone or with

other proteins containing Intrinsically Disordered Regions. When expressed in cells, mutHMGB1 phase separates around structures containing FIB1 or NPM1, i.e. nucleoli. This impairs rDNA production and cell survival. They finally generalize their finding to a number of other proteins that, when mutated by frameshift, contain long basic stretches.

The ms is interesting to read. However, it feels like the joining of two different reports: one on the genetic pathogenesis of BPTAS, and the second on the molecular consequences of proteins harbouring long stretches of basic residues. Both are interesting, but neither is completely convincing.

The BPTAS story is fascinating. However, the authors have not found a convincing motivation of why the mutated HMGB1 should cause BPTAS. Based on the second part of the ms, the embryos should die in utero, not just have milder consequences compatible with birth and postnatal life. Perhaps the authors should have created one or mouse models with one or two of the mutations found. Another observation puzzles me: two of the de novo mutations in two unrelated individuals are exactly the same. This is extremely unlikely and statistically improbable. Why is this?

The story on the molecular consequences of having an unstructured acid tail turn into a basic stretch are more well developed. Here, my queries are more technical.

Q1: The HMGB boxes are domains, i.e. they are STRUCTURED. IDRs are unstructured by definition. But the authors' choice of IDR boundaries in HMGB1 are inconsistent with this: they include a part of Box B. This affects all the constructs they make. All such constructs should be redone.

Q2: it is well known that fixing HMGB1 with formaldehyde creates artifacts. See PMID: 12925773. All pictures shown (and statistical analysis of their features) should be from living cells.

Q3: the HMGB1 frameshift mutations probably do not create IDRs. The resulting acidic stretch is most likely an alpha helix, as predicted by AlphaFold, Fig 2b. The authors should prove experimentally whether it is an IDR or a structured alpha helix. If it is an alpha helix, then much of the subsequent train of thought does not make much sense. All consequences would not follow from the fact that the IDR is basic, but simply from the fact that an excess of a strongly basic protein would create a blob with RNA, and the highest concentration of RNA is in the nucleolus. The only way to show that nucleolar localization means something more biologically relevant is to overexpress intranuclearly 2 or 3 highly basic proteins of any type and show that they DO NOT precipitate onto nucleoli.

Incidentally, the mutHMGB1 does not behave a liquid droplet together with the rest of the nucleolus, but rather makes a solid shell around it, see photobleaching in Fig 3g. It looks more like a precipitate than an IDR-driven phase separation.

Q4: the genomic/proteomic generalization is nice, but why are the authors focusing on C-terminal tails? Shouldn't the same thing happen also with internal stretches of highly basic residues? Or N-terminal tails? And again, what is the role of basic content vs alpha helical structure in the stretches arising from frameshifts?

Referee #3 (Remarks to the Author):

This paper presents evidence that pathogenic genetic variants that alter the physico-chemical

properties of carboxyterminal domains of certain proteins that are "intrinsically disordered" in their structure, resulting in abnormalities of phase separation, most especially affecting nucleolar structures. The work begins with the evaluation of frameshift variants in the 3' end of HMGB1 in four unrelated families with a BPTAS, a rare malformation syndrome that to this point has an unknown genetic etiology. Using tagged proteins they demonstrate using both in vitro re-constitution assays as well as in vitro cellular transfection experiments that the presence of frameshifting variants that convert the carboxy-terminus of this protein from being acidic to basic, an effect that is not replicated by other non-pathogenic variants. In particular partitioning of these frameshifted proteins within nucleolar structures is altered, rRNA transcription is affected and there is reduced cell viability. Extending from these observations the authors identified a cohort of reported pathogenic variants from large compendia of such variants that also introduce frameshifting variants into a large number of proteins containing c terminal IDRs. They select six and show that most of these variants also alter its spatial partitioning in the nucleolus as well as rRNA biogenesis.

This paper is of substantial interest and I read it with fascination and enjoyment. Congratulations to the authors. Molecular pathologists, cell biologists and human geneticists including those interested in germline disorders as well as somatically-acquired conditions such as cancer will find it of interest. It represents the meeting of two topical areas of research - the definition of the functional basis for genetic diseases with the newer and expanding field of the definition and characterisation of phase separation of compartments within cellular organelles.

The paper is original and the findings novel. The data is displayed clearly and the statistical analyses are appropriate with minor caveats (see below).

Suggestions for improvement.

1. The Genetics of BPTAS.

It is convincing that frameshift variants in HMGB1 cause BPTAS, particularly the genotype-phenotype correlation between variants elsewhere in the gene being associated with an entirely different phenotype. As the "lead" condition for this body of work the authors define frameshifting variants in 4 families with this condition using genome sequencing. Some additional data could be presented to round out the analysis for this aspect of the work:

- i. What other variants were identified in the genome of the lead family and what criteria were used to discount these variants in favour of the HMGB1 frameshifts? An appendix should be presented to explicitly lay out the systematic approach taken here. This suggestion is not a serious criticism but is presented for completeness's sake.
- ii. The ACMG criteria for pathogenicity should be parenthetically presented rather than merely a narrative assertion that pathogenicity can be assumed. The case seems clear but the thinking should be explicit.
- iii. A recurrent variant is identified and therefore a haplotype analysis should be presented to determine if these alleles are ancestrally discrete or the same.
- iv. If the 3' frameshifting variants are causative of BPTAS and the mechanism is discrete from the allelic neurodevelopmental disorders caused by loss-of-function alleles, it would be comforting to see no effect on rRNA expression (or other nucleolar functions) in knockout or knockdown cells (that mimic the alleles that lead to the neurodevelopmental disorder). If this is not observed and the

neurodevelopmental conditions do exhibit nucleolar functional compromise, why is there complete phenotypic incongruence between these two groups of conditions?

v. An alternative hypothesis that could account for the initial observations reported here for BPTAS is that the frameshifts produce a protein that has a carboxy terminus that promotes its aggregation entrapping it within nucleoli. Arguing against this is the stark phenotypic incongruence between the two disorders presented here, implying that proteins with the frameshifted tails retain important function competency. If the authors managed to demonstrate some functional competency for these mutant frameshifted proteins, then this would argue against the proposition that these mutations are having their effect by a mere pro-aggregative mechanism because this latter mechanism would more closely overlap (at least functionally) with the haploinsufficient alleles.

vi. Extending this idea of assessment of other HMGB1 protein functions in cells expressing 3' frameshifts, the intactness of established functions of the HMGB1 protein (particularly nuclear ones like base excision repair, chromatin localisation) could be assessed. Have they been shown to be deficient in the neurodevelopmental phenotype and not so in this new discrete condition?

2. Assessment of nucleolar function.

i. The prime measure of disruption of nucleolar function was measurement of rRNA production using RT-PCR with the significance level arrived at being $P < 0.05$ for the HMGB1 mutants. As a sole measure of alteration of nucleolar function, this is modest and needs bolstering with alternative measures.

ii. The quantitative deficit in rRNA production is performed in transfection based systems. Although the point is made that "over-expression" of these alleles is modest and congruent with the effect that would be expected in cells heterozygous for these variants, this is taken on faith to some degree. I would suggest that gene-edited cells would provide a better measure of reduction in rRNA expression (or other measures of nucleolar function).

iii. In Figure 3m how do the authors explain that overexpression of simply the mutant IDR alone causes a dramatic reduction in cell viability? This would seem to suggest a toxic effect of the mutant over-expressed IDR as opposed to a misdirection of mutant HMGB1 proteins away from their appropriate nuclear phase compartment.

iv. It would be nice to see the nucleoli tracked through formation/dissolution over mitosis, what happens to the aggregates?, how do new nucleoli form? The FRAP experiments might suggest aggregate like qualities that might persist at nucleolar dissolution.

3. Preservation of non-nucleolar functions.

The competing hypothesis to the phase separation one (i.e. the frameshifted variants drive aberrant phase separation and the pathogenic mechanism is a combination of the formation of a toxic nucleolar aggregate forming in a crowded phase environment) is that HMGB1 still retains functionality and the deficit is primarily nucleolar. This thinking aligns BPTAS with other ribosomopathies. For these conditions (Diamond-Blackfan anemia, Treacher Collins syndrome, transcription and nuclear functional deficits have been observed. Can the authors align any of these with what is seen in BPTAS?

4. Identification of frameshifting variants causing other disorders.

This approach is at the centre of what this paper's title seeks to demonstrate. It therefore needs to be robustly demonstrated that the phenomenon demonstrated for HMGB1 is generalised across

other conditions.

- i. Cognate phenotypes - do the IDR variants in each gene identified define a different phenotype to the one conventionally associated with the studied genes? As for HMGB1, the hypothesis would suggest that this mechanism results in different phenotypes to those associated with haploinsufficiency. These data are not presented. in the wider group (let alone the 6 they studied more deeply).
- ii. How do the phenotypes (or functions) relate to established disorders of nucleolar function (e.g. craniofacial disorders like Treacher Collins syndrome, mandibulofacial dysostoses, Diamond Blackfan anaemia?)
- iii. Do the exemplar disorders (n=6) represent a cross section of the ontological functions assigned to the wider group? For the phase separation to be shown to be common, this group of 6 conditions needs to be shown to be representative (and not cherry-picked) for the 100+ examples the authors found.
- iv. For the six tested IDRs it is stated that the extent of mis-partitioning into the nucleolus strongly correlated with the length of the IDR sequence replaced by the frameshift. Does this also correlate with the assignment of pathogenicity that differentiates the 521 from the 104?
- v. It is stated (Fig 4(e,g)) that the charge plots of mutant vs WT proteins is converted from a net positive to net negative charge. A quantitative analysis should be applied to support this point (it is not so compelling for some of the proteins).
- vi. The functional effect of the 6 mutants studied on RNA biogenesis seem inconsistent and do not achieve a high level of statistical significance. Why should one rRNA species (different for each disease) be preferentially affected by these different mutations? An interpretation could be that these overexpression results are indicative of generalised nucleolar malaise and a more robust measure of nucleolar functions would need to be presented to be convincing evidence of the disease pathomechanism.
- vii. I would be interested to understand if there is phenotypic congruence/incongruence in those genes that have both a stop-gain and a fs for each relevant disease gene.

5. The conclusion that it is the alteration from acidic to basic charge that mediates the phase separation abnormality.

- i. The authors adopt mutagenesis to begin to address the critical element that drives the subcellular observations associated with frameshifts within the nucleus (Fig 3j). This is laudable but frustratingly they stop short of a body of experiments that offer consistent answers. In some instances it seems like the IDR is necessary for direction of the protein into the nucleolus, whereas substitution of arginine residues (or their deletion) results in granular immunostaining dissimilar to wildtype. The current data would indicate that (a) loss of the IDR excludes WT protein from the nucleus, (b) a function that is compensated for by reintroduction of the portion of the IDR proximal to the frameshift and (c) replacement of the acidic residues introduced by the frameshift seem to confer ambiguous results that are difficult to relate to the pattern seen in the full length mutant. This data needs some clarification.
- ii. It would add compelling evidence to disambiguate whether it is the loss of the acidic domain (e.g. like FIB), the gain of the basic domain (? not just Arg - will polylysine suffice?) or both, that mediate the characterised phase transition and rRNA biogenesis defects if similar mutagenesis experiments were performed for some of the other 6 exemplars. What is it about RUNX1 that its mutant does not locate to the FIB1 positive nucleolar compartment?

6. Final conclusions.

The concluding sentences of the abstract (rightly, in my view) summarises the findings of this work as suggestive of a mechanism for many different conditions with causative alleles positioned within C-terminal IDRs. This contrasts with the implied certainty within the title of this paper which should be changed with the current level of evidence offered in this manuscript.

Author Rebuttals to Initial Comments:

General response to reviewers

We thank the reviewers for their valuable comments and suggestions, which guided us to a substantially improved manuscript. The reviewers described the original submission as “fascinating, “of substantial interest”, and the findings as “novel.” The reviewers also noted that the manuscript would benefit from better joining the clinical and functional data, and improving genotype-phenotype correlation and functional characterization of the disease-associated frameshift variants. We have revised the manuscript to address these and additional minor concerns. In brief,

1. We have diagnosed an additional patient with BPTAS, and include clinical and genotype data further strengthening genotype-phenotype correlation.
2. We provide direct experimental evidence that the BPTAS-associated frameshift mutant HMGB1 protein disrupts nucleolar function.
3. We have performed extensive mutagenesis of the BPTAS-associated frameshift mutant HMGB1 protein, and identified the sequence features responsible for nucleolar mispartitioning and nucleolar arrest.
4. We characterized an additional five disease-associated frameshift variants in our catalog, and show that four out of five cause nucleolar mispartitioning – almost doubling the number of variants validated in our study, and further strengthening the generality of the findings.

Furthermore, we re-organized the text to better integrate the clinical and functional data, improved the biochemistry experiments, and reproduced critical data on transiently transfected cells by generating stable cell lines expressing wild type and mutant HMGB1 proteins.

Taken together, the results identify nucleolar mispartitioning and arrest as the cause of a rare complex syndrome in humans, and suggest a common mechanism of nucleolar condensate dysregulation for hundreds of variants with previously unknown function.

Referees' comments:

Referee #1 (Remarks to the Author):

Mensah et al report a new genetic mutation in the protein HMGB1 that is associated with pathogenicity of the rare disease BPTAS: a frame shift associated with a charge reversal in the C-terminal IDR, from an acidic to a basic, arginine-rich tail. The authors show that the mutant exhibits a lower threshold for phase separation compared to WT and somewhat different partitioning preference when tested in in vitro model systems of condensates. The mutant aberrantly partitions into the nucleolus, perturbs ribosome biogenesis and cell viability. The authors postulate that the aberrant nucleolar localization is driven by charge-charge interactions mediated by the newly acquired arginine-rich disordered tail and that this mechanism might be shared by other genetic translocations. They proceed to search in GWAS data bases and identify frame shifts in other genes that result in Arg-rich C-terminal IDRs. The authors perform an extensive bioinformatics analysis of the identified frameshift mutants and select a set of 6 genes to perform proof of concept cell biology testing.

The concept that positively charged proteins, and in particular proteins rich in arginines partition into the nucleolus is not novel. However, the concept that nucleolar sequestration of multiple genes associated with different pathologies and consequential nuclear functional disruption represents a common hub in the dysregulation is.

The manuscript nicely blends a broad range of techniques, ranging from clinical studies, to bioinformatics, genetics, in vitro biophysics and cell biology to investigate the problem from complementary angles. In general, the manuscript is well written. However, some of the claims are overreaching, in particular that referring to the generality of the mechanism of nucleolar partitioning to 600+ Arg-rich frame shift mutations, based on an experimental sampling of 6 genes. I think that the manuscript will be valuable to the readers of Nature, once some of the concerns listed below have been addressed:

We thank the reviewer for the useful comments and suggestions. We have doubled the number of functionally tested genes, and addressed the concerns as described below.

1. Abstract: “Here we show that disease-associated mutations in disordered regions frequently alter phase separation, cause mispartitioning into the nucleolus, and disrupt nucleolar function.” This sentence is overreaching. The authors show for a small number of genes (<1%) that exhibit a frameshift resulting in an Arg-rich C-terminal tail partition in the nucleolus and identify thousands of genes with arguably similar amino-acid syntax. However, given the small number of experimental points, the conclusion of nucleolar partitioning and disruption of nucleolar function are only circumstantial. I suggest softening the claim accordingly.

We characterized an additional five disease-associated frameshift variants in our catalog, and show that four out of five cause nucleolar mispartitioning – almost doubling the number of variants validated in our study, and further strengthening the generality of the findings. Furthermore, as suggested, we softened said claim in the abstract, which now reads: “Here we

show that a subset of disease-associated mutations in disordered regions alter phase separation, cause mispartitioning into the nucleolus, and disrupt nucleolar function.”

2. For the in vitro experiments, it is imperative that the recombinant protein be clean of nucleic acids and other impurities, as well as of soluble oligomers, all of which might affect the observed phase behavior and experimental reproducibility. The fact that the authors used a single chromatography step for the purification of all their proteins raises some concerns regarding the level of purity and monodispersity of their material. It would be useful for the authors to show the recombinant protein gels demonstrating the level of purity and discuss the quality control criteria that were used for the characterization of the proteins. In particular, the observation of multi-phase separation in the presence of mutant but not WT HMGB1 could be a result of condensate network rearrangements due to preferential (1) homotypic mutHMGB1 OR (2) heterotypic mutHMGB1 and residual nucleic acids in the recombinant protein prep. The polycationic mutHMGB1 IDR is very likely to be interacting non-specifically to RNA and DNA and even phase separate with nucleic acids in a heterotypic fashion. I suggest the authors perform a control experiment, measuring the C_{sat} of WT vs mutant in the presence of nucleic acids.

We have substantially expanded the biochemical characterization of the wild type and mutant HMGB1 proteins, improved the purity of the preparations, reproduced the key data with synthetic peptides, and performed in vitro experiments in the presence of nucleic acids. The new data reproduced all key findings.

We have purified wild type and mutant HMGB1 proteins using multiple purification steps, including affinity chromatography followed by size exclusion chromatography. We include the chromatography elution profile, PAGE gels, and Western blots, which show high purity of the proteins (Extended Data Fig. 8a-c). All key findings in the original submission were reproduced using the purified proteins: we found that the mutant HGMB1 protein has lower C_{sat} than the wild type (Fig. 2c-d), forms droplets that with negligible internal dynamics (Fig. 2e), and shows enhanced partitioning into NPM1 condensates in vitro compared to wild type (Fig. 2f-h).

To rule out the potential confounding effects of residual nucleic acids in the protein preparations, we reproduced the key experiments with synthetic wild type and mutant HMGB1 IDR peptides. The results were consistent with the results obtained with the purified proteins (new Fig. 2i, new Extended Data Fig. 8d-j).

Last we performed the suggested control experiments measuring C_{sat} of proteins in the presence of RNA. As expected, we found that RNA enhanced in vitro condensate formation of the mutant HMGB1 synthetic IDR peptide (Extended Data Fig. 8d-g). These results further strengthen the mechanistic basis of mispartitioning of the mutant protein into nucleoli in cells, which are the most RNA-rich nuclear compartment.

3. Pg. 5: “The droplets were spherical in size, settled on the surface (Fig. 2c) and occasionally underwent fusion (not shown), which are hallmarks of phase separation 34”. I suggest including a SI video showing droplet fusion instead of the “(not shown)” specification.

We include a video of a droplet fusion event as Supplementary video 1 in the revised manuscript.

4. Pg. 8: Please elaborate a bit more on the justification for why these particular 6 genes were selected for proof-of-concept studies?

We found that transcription factor genes in general are highly enriched for frameshift mutations that create arginine-rich basic protein tails (Extended Data Fig. 13), therefore we selected transcription factor genes for validation. We better explain this now in the text. To strengthen the generality of our findings, we also include validation experiments of two cytoplasmic proteins in addition to the nine transcription factors (Fig. 4e-f, Extended Data Fig. 16-20).

5. In addition to the fraction of positive residues and Arg residues in particular, it would be interesting to look at the distribution of these residues (e.g., short Arg clusters/stickers connected by longer linkers vs. long, Arg-dense stickers). For example, C9Orf72 di-peptide repeats, which have been shown to partition into and disrupt nucleolar function exhibit a long and very dense Arg cluster, compared to endogenous nucleolar proteins, where the Arg clusters are shorter and less dense, broken up via linkers. Are the non-Arg linkers enriched in aromatic residues, which will make them stickers of a different type? Do the pathological variants fall into the category of IDRs with densely packed stickers?

We thank the reviewer for this insightful comment, which led to new insights into the sequence features responsible for the phenotypic effects of the mutant HMGB1 protein. We found that in the mutant HMGB1 protein, the arginines are tightly stacked, but the C-terminus of the protein indeed contains a hydrophobic patch that is enriched in hydrophobic residues, including aromatic residues. We show that the stacked arginines drive nucleolar mispartitioning, and the hydrophobic patch is responsible for nucleolar arrest and consequent cytotoxicity (new Fig. 3e-g, 3j).

6. Pg. 9: “Arginine-rich sequences were previously shown to facilitate nucleolar localization, but native proteins with arginine-rich stretches form monophasic phase-separated droplets with NPM1 in vitro, and do not affect the liquid-like properties of nucleolar condensates 40.” Riback et al, Nature 2020 shows that changes in compositional blend of a three component in vitro NPM1 droplet do change the condensate thermodynamics; this is also true for nucleoli in living cells, when the relative levels of nucleolar proteins and rRNA change. While the paper did not directly measure changes in material properties, it is expected that these change with the dense phase thermodynamic stability. Please revise this sentence accordingly.

We include FRAP data on the material properties of frameshift mutant proteins that mispartition into nucleoli (new Extended Data Fig. 20b). As discussed above, the HMGB1 mutant protein that lacks a C-terminal hydrophobic patch did partition into the nucleolus, but did not arrest nucleolar dynamics and did not kill the cells (new Fig. 3e-g, 3j). We also include FRAP experiments of the additional mutant proteins (new Extended Data Fig. 20b). We revised the text accordingly.

Referee #2 (Remarks to the Author):

The authors start from the finding that 5 different individuals with BPTAS, an ultra-rare syndrome, share frameshift mutations in HMGB1 that turn the acidic tail into a long basic tail. They then show that the mutated HMGB1 is much more prone to phase separation in vitro, either alone or with other proteins containing Intrinsically Disordered Regions. When expressed in cells, mutHMGB1 phase separates around structures containing FIB1 or NPM1, i.e. nucleoli. This impairs rDNA production and cell survival. They finally generalize their finding to a number of other proteins that, when mutated by frameshift, contain long basic stretches.

The ms is interesting to read. However, it feels like the joining of two different reports: one on the genetic pathogenesis of BPTAS, and the second on the molecular consequences of proteins harbouring long stretches of basic residues. Both are interesting, but neither is completely convincing.

The BPTAS story is fascinating. However, the authors have not found a convincing motivation of why the mutated HMGB1 should cause BPTAS. Based on the second part of the ms, the embryos should die in utero, not just have milder consequences compatible with birth and postnatal life. Perhaps the authors should have created one or mouse models with one or two of the mutations found.

Another observation puzzles me: two of the de novo mutations in two unrelated individuals are exactly the same. This is extremely unlikely and statistically improbable. Why is this?

The story on the molecular consequences of having an unstructured acid tail turn into a basic stretch are more well developed. Here, my queries are more technical.

We thank the reviewer for the valuable comments, and sharing our fascination with the underlying biology. We have substantially improved the depth and description of the clinical findings and molecular findings, and re-organized the text to better integrate the clinical and molecular data.

We significantly improved the depth and description of the clinical findings, which together provide a compelling case for the pathogenicity of the HMGB1 frameshift mutation. We diagnosed an additional patient with BPTAS, and include key information of variants identified in the whole genomes of two individuals and whole exome of one individual (Extended Data Fig. 6a). These data, and the previously included genotyping data further contribute to linking the variants to BPTAS. Furthermore, we explicitly address the criteria of the American College of Medical Genetics (ACMG) used for evaluation of variant pathogenicity in the revised manuscript text. Applying the ACMG criteria we conclude that the de novo variants are classified as pathogenic. We indeed looked into generating a mouse model, but this is not trivial as the mouse HMGB1 sequence contains a different 3'UTR, therefore, the same frameshift mutation results in a protein sequence in mice that is substantially different from the sequence created by the same frameshift in the human sequence. Last, the mutant HMGB1 is indeed cytotoxic in our experiments, but we argue that this is not necessarily incompatible with life. We hypothesize that cells encoding the frameshift mutant HMGB1 attempt to suppress the RNA produced from the

mutant allele. This hypothesis is consistent with findings that i) HMGB1 knockout mice are not embryonically lethal, but the mice are born alive and die within 24 hours due to hypoglycemia (Calogero et al., 1999); ii) BPTAS patients do tend to have early onset phenotypes (Supplementary Table 2); iii) mutant HMGB1-expressing cells are capable of undergoing mitosis (new Supplementary video 2).

The identification of the same de novo variant in several unrelated individuals with an ultra-rare congenital malformation syndrome is considered as evidence for the pathogenicity of such variant. In this new version of the manuscript four unrelated affected individuals with the same HMGB1 variant were identified. It is virtually impossible for the same de novo variant to occur in these four unrelated individuals with the same and ultra-rare phenotype merely by chance. Using the recurrence (ACMG criterion PS4) of de novo (ACMG criterion PS2) variants to prove the association of a phenotype and a certain variant is a well-established principle in human genetics. Such recurrent pathogenic de novo variants occur in many genetic disorders, e.g. achondroplasia (specific pathogenic variants in the *FGFR3* gene), Crouzon syndrome (specific pathogenic variants in the *FGFR2* gene), ceroid lipofuscinosis, Kufs type (specific variants in the *DNAJC5* gene) and many more. Therefore, while it may appear counterintuitive, the same de novo variant identified in four BPTAS individuals in fact supports that notion that the variant is pathogenic.

Q1: The HMGB boxes are domains, i.e. they are STRUCTURED. IDRs are unstructured by definition. But the authors' choice of IDR boundaries in HMGB1 are inconsistent with this: they include a part of Box B. This affects all the constructs they make. All such constructs should be redone.

We include new data on a smaller IDR protein that is devoid of any sequence overlapping the HMG boxes. The small IDR recapitulated previous findings: it accumulated in the nucleolus, and arrested nucleolar dynamics (new Extended Data Fig. 9c.)

We note that IDRs can and do tend to have *propensity* to form heterogeneous secondary structures, but they do not assume one stable conformation. Also, the definition of domains in protein sequences are confounded by limitation of the experimental approaches to detect them. Therefore, IDR- and structured domain annotations are imperfect, and there can be major inconsistencies among the dozens of tools that predict them. The current standard AlphaFold2, and the previous standard PONDR for example, both predict the C-terminal portion of the second HMG box to be largely unstructured, which guided our original selection of sequences for functional analyses.

Q2: it is well known that fixing HMGB1 with formaldehyde creates artifacts. See PMID: 12925773. All pictures shown (and statistical analysis of their features) should be from living cells.

We thank the reviewer for pointing this out. We show pictures and provide key analyses from living cells (Fig. 3a, 3c, 3f, Extended Data Fig. 9a-d, 9k, 10e-f). We kept one experiment with

fixed cells in the Extended Data Fig. 9h-j, where we show that heterochromatin foci, RNPAII puncta, BRD4 puncta and SC35 speckles do not overlap HMGB1 inclusions, and now include a control experiment showing that the results are not affected by the crosslinking (new Extended Data Fig. 9f-g).

Q3: the HMGB1 frameshift mutations probably do not create IDRs. The resulting acidic stretch is most likely an alpha helix, as predicted by AlphaFold, Fig 2b. The authors should prove experimentally whether it is an IDR or a structured alpha helix. If it is an alpha helix, then much of the subsequent train of thought does not make much sense. All consequences would not follow from the fact that the IDR is basic, but simply from the fact that an excess of a strongly basic protein would create a blob with RNA, and the highest concentration of RNA is in the nucleolus. The only way to show that nucleolar localization means something more biologically relevant is to overexpress intranuclearly 2 or 3 highly basic proteins of any type and show that they DO NOT precipitate onto nucleoli. Incidentally, the mutHMGB1 does not behave a liquid droplet together with the rest of the nucleolus, but rather makes a solid shell around it, see photobleaching in Fig. 3g. It looks more like a precipitate than an IDR-driven phase separation.

The frameshift in HMGB1 seems to create an IDR that has the *propensity* to form an alpha helix. We again stress that IDRs can and do tend to assume heterogeneous secondary structures that are transient and unstable, rather than being inherently devoid of any structure.

We performed circular dichroism experiments on synthetic peptides corresponding to the C-terminus of wild type and mutant HMGB1 in the revised manuscript. We found that the mutant peptide had some propensity to form a helix in solution in the presence of low amounts of TFE (which is used to dehydrate proteins which stabilizes their secondary structure) (new Extended Data Fig. 7c-e). The data suggest helicity in <10% of the time/molecule population, consistent with the AlphaFold2 prediction of some helical propensity. The sequence is clearly not a stable helix (new Extended Data Fig. 7d-e).

We also performed additional dissection of the frameshift sequence, which led to new insights into the sequence features responsible for the phenotypic effects of the mutant HMGB1 protein. We found that the mutant HMGB1 contains an arginine-rich portion, but the C-terminus of the protein contains a hydrophobic patch that is enriched in hydrophobic residues (Fig. 3e). We show that arginine enrichment drives mispartitioning into a nucleolus due to a phase-separation mechanism, and the hydrophobic patch is responsible for the nucleolar arrest and consequent cytotoxicity (new Fig. 3e-g, 3j). Consistent with the suggested experiments, expression of a mutant protein that contains the arginine-enriched portion but lacks the hydrophobic patch (predicted to drive helical propensity) mispartitioned into the nucleolus, but did not arrest the nucleolar dynamics and did not kill the cells (“Patchless” in new Fig. 3e-g, 3j).

Q4: the genomic/proteomic generalization is nice, but why are the authors focusing on C-terminal tails? Shouldn't the same thing happen also with internal stretches of highly basic residues? Or N-terminal tails? And again, what is the role of basic content vs alpha helical structure in the stretches arising from frameshifts?

In general, frameshifts (or any other mutation) that produce a STOP codon that occurs before the last 50 bp of a gene's penultimate exon typically triggers nonsense mediated decay (NMD) of the affected mRNA (Kurosaki and Maquat, 2016; Lindeboom et al., 2016; Litchfield et al., 2020). As a consequence, frameshifts that occur in internal protein stretches or N-terminal tails could only lead to a translated protein if the frameshift does not create a STOP codon before the last 50bp of the gene's penultimate exon. To make this clearer, we now include the information on the likelihood of the frameshift variant sequences to not be targeted by NMD (Supplementary table 5). In brief, we found that ~90% of the variants in our catalog are predicted to escape NMD. Last, consistent with the new insights described at Q3, we expanded the characterization of the mutations in the catalog to include the number and position of hydrophobic patches.

Referee #3 (Remarks to the Author):

This paper presents evidence that pathogenic genetic variants that alter the physico-chemical properties of carboxyterminal domains of certain proteins that are "intrinsically disordered" in their structure, resulting in abnormalities of phase separation, most especially affecting nucleolar structures. The work begins with the evaluation of frameshift variants in the 3' end of HMGB1 in four unrelated families with a BPTAS, a rare malformation syndrome that to this point has an unknown genetic etiology. Using tagged proteins they demonstrate using both in vitro reconstitution assays as well in vitro cellular transfection experiments that the presence of frameshifting variants that convert the carboxy-terminus of this protein from being acidic to basic, an effect that is not replicated by other non-pathogenic variants. In particular partitioning of these frameshifted proteins within nucleolar structures is altered, rRNA transcription is affected and there is reduced cell viability. Extending from these observations the authors identified a cohort of reported pathogenic variants from large compendia of such variants that also introduce frameshifting variants into a large number of proteins containing c terminal IDRs. They select six and show that most of these variants also alter its spatial partitioning in the nucleolus as well as rRNA biogenesis.

This paper is of substantial interest and I read it with fascination and enjoyment. Congratulations to the authors. Molecular pathologists, cell biologists and human geneticists including those interested in germline disorders as well as somatically-acquired conditions such as cancer will find it of interest. It represents the meeting of two topical areas of research - the definition of the functional basis for genetic diseases with the newer and expanding field of the definition and characterisation of phase separation of compartments within cellular organelles.

The paper is original and the findings novel. The data is displayed clearly and the statistical analyses are appropriate with minor caveats (see below).

We are grateful for sharing our fascination and enthusiasm, and appreciate the useful comments and suggestions. We have substantially improved the clinical and molecular analyses as described below.

Suggestions for improvement.

1. The Genetics of BPTAS.

It is convincing that frameshift variants in HMGB1 cause BPTAS, particularly the genotype-phenotype correlation between variants elsewhere in the gene being associated with an entirely different phenotype. As the "lead" condition for this body of work the authors define frameshifting variants in 4 families with this condition using genome sequencing. Some additional data could be presented to round out the analysis for this aspect of the work:

i. What other variants were identified in the genome of the lead family and what criteria were used to discount these variants in favour of the HMGB1 frameshifts? An appendix should be presented to explicitly lay out the systematic approach taken here. This suggestion is not a serious criticism but is presented for completeness's sake.

We diagnosed an additional patient with BPTAS and now include clinical information on five individuals. We include key information of variants identified, the filtering strategy, and numbers in the whole genomes of two individuals and whole exome of one individual (new Extended Data Fig. 6a).

ii. The ACMG criteria for pathogenicity should be parenthetically presented rather than merely a narrative assertion that pathogenicity can be assumed. The case seems clear but the thinking should be explicit.

We have revised the paragraph in question explicitly adding the ACMG criteria and summarizing the evidence level according to the ACMG's terminology and guidelines. (Paragraph starting with the sentence: "The clinical genetic information of the five BPTAS individuals and functional evidence were used to classify the HMGB1 frameshift variants in accordance with the ACMG criteria as pathogenic.")

iii. A recurrent variant is identified and therefore a haplotype analysis should be presented to determine if these alleles are ancestrally discrete or the same.

Haplotype analyses searching for shared ancestry could be interesting. However, the variants we identified occurred *de novo*, and the five affected individuals have various ethnic backgrounds (German, Iranian, Kirgisian, Hong Kong Chinese), which rules out an ancestrally shared HMGB1 haplotype.

iv. If the 3' frameshifting variants are causative of BPTAS and the mechanism is discrete from the allelic neurodevelopmental disorders caused by loss-of-function alleles, it would be comforting to see no effect on rRNA expression (or other nucleolar functions) in knockout or knockdown cells (that mimic the alleles that lead to the neurodevelopmental disorder). If this is not observed and the neurodevelopmental conditions do exhibit nucleolar functional compromise, why is there complete phenotypic incongruence between these two groups of conditions?

HMGB1 loss of function is generally not associated with nucleolar dysfunction (Kang et al., 2014). Recent papers have reported on HMGB1 knockdown, and these papers did not describe effects associated with nucleolar dysfunction. For example, Sofiadis et al found that HMGB1 knockdown in multiple cell types affected chromatin looping, senescence and DNA repair (Sofiadis et al., 2021).

We note that BPTAS-phenotypes caused by the HMGB1 frameshift mutation are not entirely incongruent with the neurodevelopmental syndromes associated with HMGB1 loss of function. For example, 11/12 BPTAS patients present with microcephaly, and for six patients with available clinical data, six out of six presented with developmental delay of neurological features, similar to the patient described in Uguen et al. These data are included in Supplementary Table 1.

v. An alternative hypothesis that could account for the initial observations reported here for BPTAS is that the frameshifts produce a protein that has a carboxy terminus that promotes its

aggregation entrapping it within nucleoli. Arguing against this is the stark phenotypic incongruence between the two disorders presented here, implying that proteins with the frameshifted tails retain important function competency. If the authors managed to demonstrate some functional competency for these mutant frameshifted proteins, then this would argue against the proposition that these mutations are having their effect by a mere pro-aggregative mechanism because this latter mechanism would more closely overlap (at least functionally) with the haploinsufficient alleles.

Our model is that the frameshift mutation leads to nucleolar mispartitioning and nucleolar arrest which is ultimately cytotoxic. As discussed at points 'iv' and 'vi', we agree that the partial incongruence of the BPTAS phenotypes and the neurodevelopmental syndromes associated with HMGB1 loss of function provide compelling evidence that the frameshift mutation causing BPTAS is not loss of function. At the same time, the partial congruence of phenotypes does suggest that some of the phenotypes described in BPTAS patients are in part caused by a loss of function effect (the patients are heterozygous for the wild type allele). We describe these findings in more detail in the Supplementary Notes.

vi. Extending this idea of assessment of other HMGB1 protein functions in cells expressing 3' frameshifts, the intactness of established functions of the HMGB1 protein (particularly nuclear ones like base excision repair, chromatin localisation) could be assessed. Have they been shown to be deficient in the neurodevelopmental phenotype and not so in this new discrete condition?

The neurodevelopmental phenotype was only very recently linked to *HMGB1* (Uguen et al., 2021) and no functional data exist yet about the underlying pathogenic variants. However, type and position of the variants (microdeletions of the entire locus, early frameshifts) characterize them as loss of function variants contrasting the apparent neomorphic effects of the BPTAS causing frameshift variants presented and analyzed in our study. To address the reviewer's concerns, we now explain and clarify this in the genotype-phenotype correlation section in the Supplementary Notes.

2. Assessment of nucleolar function.

i. The prime measure of disruption of nucleolar function was measurement of rRNA production using RT-PCR with the significance level arrived at being $P < 0.05$ for the HMGB1 mutants. As a sole measure of alteration of nucleolar function, this is modest and needs bolstering with alternative measures.

In the revised manuscript we provide direct experimental evidence that the BPTAS-associated frameshift mutant HMGB1 protein disrupts nucleolar function. As a readout for ribosome biogenesis, we measured nascent translation using puromycin staining in U2OS cells expressing wild type and mutant HMGB1 proteins. Cells expressing mutant HMGB1 consistently displayed lower level of puromycylation of nascent proteins compared to non-transfected cells or compared to cells transfected with wild type EGFP-HMGB1 (new Fig. 3h-I, new Extended Data Fig. 10b-c).

ii. The quantitative deficit in rRNA production is performed in transfection based systems. Although the point is made that "over-expression" of these alleles is modest and congruent with the effect that would be expected in cells heterozygous for these variants, this is taken on faith to some degree. I would suggest that gene-edited cells would provide a better measure of reduction in rRNA expression (or other measures of nucleolar function).

In the revised manuscript, we reproduced key findings of the transient transfection experiments by generating and testing cell lines that express stably integrated transgenes. In brief, we used a PiggyBac transposon system to generate cell lines that express wild type and mutant HMGB1 proteins under a Doxycycline-inducible promoter. In the stable cell line system, mutant HMGB1 formed nuclear inclusions, caused nuclear arrest and reduced viability, consistent with the results using transiently transfected cells (new Extended Data Fig. 10e-j).

iii. In Figure 3m how do the authors explain that overexpression of simply the mutant IDR alone causes a dramatic reduction in cell viability? This would seem to suggest a toxic effect of the mutant over-expressed IDR as opposed to a misdirection of mutant HMGB1 proteins away from their appropriate nuclear phase compartment.

Our model is that the frameshift mutation leads to nucleolar mispartitioning and nucleolar arrest which is ultimately cytotoxic. In the revised manuscript we identified the sequence features within the frameshifted IDR responsible for these phenotypes. We show that the mutant HMGB1 contains an arginine-rich portion, but the C-terminus of the protein contains a hydrophobic patch that is enriched in hydrophobic residues (Fig. 3e). We show that arginine enrichment drives mispartitioning into a nucleolus due to a phase-separation mechanism, and the hydrophobic patch is responsible for the nucleolar arrest and consequent cytotoxicity (Fig. 3e-g, 3j). Consistent with the suggested experiments, expression of a mutant protein that contains the arginine-enriched portion but lacks the hydrophobic patch (predicted to drive helical propensity) mispartitioned into the nucleolus, but did not arrest the nucleolar dynamics and did not kill the cells ("Patchless" in new Fig. 3e-g, 3j).

iv. It would be nice to see the nucleoli tracked through formation/dissolution over mitosis, what happens to the aggregates?, how do new nucleoli form? The FRAP experiments might suggest aggregate like qualities that might persist at nucleolar dissolution.

Surprisingly, the nucleoli accumulating mutant HMGB1 protein dissolve and reform over mitosis. We include these data as new Supplementary video 2.

3. Preservation of non-nucleolar functions.

The competing hypothesis to the phase separation one (i.e. the frameshifted variants drive aberrant phase separation and the pathogenic mechanism is a combination of the formation of a toxic nucleolar aggregate forming in a crowded phase environment) is that HMGB1 still retains functionality and the deficit is primarily nucleolar. This thinking aligns BPTAS with other ribosomopathies. For these conditions (Diamond-Blackfan anemia, Treacher Collins syndrome, transcription and nuclear functional deficits have been observed. Can the authors align any of these with what is seen in BPTAS?

There appears to be some but little overlap between the symptoms of BPTAS, and the symptoms of ribosomopathies like Diamond-Blackfan anemia or Treacher Collins syndrome. For example, approximately half of individuals with Diamond-Blackfan anemia display microcephaly. BPTAS and Treacher-Collins syndrome share microtia, but no hematological or malignant findings are described in BPTAS. Also, as described above at point ‘1/iv’, BPTAS-phenotypes may in part be caused by the nucleolar dysfunction and in part heterozygous loss of function, and we hypothesize that cells encoding the frameshift mutant HMGB1 likely attempt to suppress the RNA produced from the mutant allele. We added all these information as a paragraph in the Supplementary Notes.

4. Identification of frameshifting variants causing other disorders.

This approach is at the centre of what this paper's title seeks to demonstrate. It therefore needs to be robustly demonstrated that the phenomenon demonstrated for HMGB1 is generalised across other conditions.

We have expanded the computational analyses, and doubled the number of functionally tested variants. The results provide further support that arginine-rich sequences created by frameshift variants mispartition into the nucleolus and may cause nucleolar dysfunction.

i. Cognate phenotypes - do the IDR variants in each gene identified define a different phenotype to the one conventionally associated with the studied genes? As for HMGB1, the hypothesis would suggest that this mechanism results in different phenotypes to those associated with haploinsufficiency. These data are not presented in the wider group (let alone the 6 they studied more deeply).

We thank the reviewer for these important questions and ideas. First, we stress that virtually all arginine-rich frameshift variants we identify are extremely rare, which makes meaningful genotype-phenotype correlation analyses unfeasible. For example, among the 11 variants we selected from the catalog for functional testing, four affect genes that contain only those pathogenic variants. Nevertheless, evidence for a genotype-phenotype correlation was discernible from clinical data for FOXC1. We describe the relevant literature evidence on genotype-phenotype correlation for each of the 11 genes and variants in the revised Supplementary Notes.

Furthermore, we found that genes that contain pathogenic frameshift variants, or pathogenic frameshift variants that create arginine-rich sequences are on average associated with ~4 diseases, whereas the frameshift variants and frameshift variants that create arginine-rich sequences are associated on average with one disease only. We find these data given the limited clinical and human genetic

information suggestive, but not compelling enough to include in the manuscript. We include these data here for the reviewer.

ii. How do the phenotypes (or functions) relate to established disorders of nucleolar function (e.g. craniofacial disorders like Treacher Collins syndrome, mandibulofacial dysostoses, Diamond Blackfan anaemia?)

As described above, there appear to be some but little overlap between the symptoms of BPTAS, and the symptoms of ribosomopathies like Diamond-Blackfan anemia or Treacher Collins syndrome. We added all this information as a paragraph in the Supplementary Notes. To the extent of our analyses, the rare diseases associated with arginine-rich frameshifts appear to share little to no overlap in symptoms with established disorders of nucleolar function. We believe that the lack of overlap, and disease-specific phenotypes are a consequence of different tissues being predominantly affected. The genes in which we identify arginine-rich frameshifts tend to have tissue-restricted expression profiles (most are lineage-specific transcription factors), suggesting that the tissues in which those genes are transcribed will be predominantly affected. This is different from e.g. Diamond Blackfan anemia which is caused by mutations in ribosomal proteins. We added this information to the revised Supplementary Notes.

iii. Do the exemplar disorders (n=6) represent a cross section of the ontological functions assigned to the wider group? For the phase separation to be shown to be common, this group of 6 conditions needs to be shown to be representative (and not cherry-picked) for the 100+ examples the authors found.

In the revised version we tested a further five examples, making the total number variants tested eleven. We found in the mutation catalog that frameshifts that create arginine-rich sequences are highly enriched in transcription factor genes (Extended Data Fig. 13). Therefore, we prioritized variants in transcription factor genes for functional testing. We selected nine transcription factor genes from the catalog, making the number of TFs tested in total ten (including HMGB1). To further improve the generality of the findings, we also included a perinuclear and a cytoplasmic protein in the functional testing (CALR, SQSTM1). All this information is included in the revised text and in Figure 4.

iv. For the six tested IDRs it is stated that the extent of mis-partitioning into the nucleolus strongly correlated with the length of the IDR sequence replaced by the frameshift. Does this also correlate with the assignment of pathogenicity that differentiates the 521 from the 104?

We thank the reviewer for suggesting this analysis. There is indeed a weak but significant correlation between length and pathogenicity. The difference in the length between the arginine-rich frameshift variants and the length of the sequence removed by the variant is smaller for pathogenic frameshifts. This new analysis is included as new Extended Data Fig. 14g.

v. It is stated (Fig 4(e,g)) that the charge plots of mutant vs WT proteins is converted from a net positive to net negative charge. A quantitative analysis should be applied to support this point (it is not so compelling for some of the proteins).

We include quantitative analysis of the isoelectric points of the sequences created by the frameshifts, and the sequences replaced by the frameshifts in new Extended Data Fig. 17. For 9/11 proteins tested (6/6 in the original submission), the frameshift replaces a net acidic sequence with a net basic sequence. For the other two, the charge is substantially increased by the frameshift. Since the new functional experiments revealed that arginine enrichment drives nucleolar mispartitioning (new Fig. 3e-f), we removed the sentence on charge from the text.

vi. The functional effect of the 6 mutants studied on RNA biogenesis seem inconsistent and do not achieve a high level of statistical significance. Why should one rRNA species (different for each disease) be preferentially affected by these different mutations? An interpretation could be that these overexpression results are indicative of generalised nucleolar malaise and a more robust measure of nucleolar functions would need to be presented to be convincing evidence of the disease pathomechanism.

As described in detail at the response to comment 3, we now provide direct experimental evidence that the BPTAS-associated frameshift mutant HMGB1 protein disrupts nucleolar function. As a readout for ribosome biogenesis, we measured nascent translation using puromycin staining in U2OS cells expressing wild type and mutant HMGB1 proteins. We analyzed over 30,000 (!) total cells, and found that cells expressing mutant HMGB1 consistently displayed lower level of puromycin staining of nascent proteins compared to non-transfected cells or compared to cells transfected with wild type EGFP-HMGB1 (new Fig. 3h-i, Extended Data Fig. 10b-c). These results suggest that the relatively small differences in rRNA levels measured by qRT-PCR do translate to measurable functional differences.

We also note that while statistical significance of the qRT-PCR data in Figure 4g indeed appear modest, we corroborated the measured differences using three different internal controls (Extended Data Fig. 21a-c).

Furthermore, for the HMGB1 variant we do reproduce key functional data of the HMGB1 variant using cell lines that express stably integrated transgenes, (without overexpressing transiently transfected constructs). In brief, we used a PiggyBac transposon system to generate cell lines that express with type and mutant HMGB1 proteins under a Doxycycline-inducible promoter. In the stable cell line system, mutant HMGB1 formed nuclear inclusion, caused nuclear arrest and reduced viability, consistent with the results using transiently transfected cells (new Extended Data Fig. 10e-j).

Last, the reviewer makes an important point that the impact of the various mutant proteins on the nucleolus could be a generalized nucleolar malaise. This idea is compatible and entirely consistent with our model! The new data on HMGB1 mutant revealed that the cytotoxic effect on the cells is correlated with nucleolar arrest (Fig. 3e-j). We measured nucleolar dynamics using FRAP in cells expressing eleven additional mutant proteins and found evidence for nucleolar arrest for several (but not all of them). Dissecting the precise sequence features that cause nucleolar arrest will be an important goal for follow up work.

vii. I would be interested to understand if there is phenotypic congruence/incongruence in those genes that have both a stop-gain and a fs for each relevant disease gene.

Again, this is an insightful question, where the amount of available clinical data is limiting. In our variant catalog that contains >5,000 genes, >7,000 frameshifts, and >10,000 truncating variants, there are only thirteen genes that contain a pathogenic arginine-rich frameshift, and a pathogenic truncating variant. For all thirteen, the disease associated with the two variants is the same, suggesting – on the surface – a phenotypic congruence. However, closer inspection of the data used to annotate the variants in Clinvar revealed that the thirteen variants have the lowest (one star) rating of the level of evidence for pathogenicity in Clinvar, and that the level of evidence was in fact that the variants occurred in a gene that contains another variant associated with the same disease. We note that for BPTAS, it is indeed our clinical investigation and functional studies that were necessary to reveal the phenotypic incongruence between the loss of function variants and arginine-rich frameshift variants. Our work will therefore likely facilitate revisiting the annotation and inspire functional studies of these and other variants.

5. The conclusion that it is the alteration from acidic to basic charge that mediates the phase separation abnormality.

i. The authors adopt mutagenesis to begin to address the critical element that drives the subcellular observations associated with frameshifts within the nucleus (Fig 3j). This is laudable but frustratingly they stop short of a body of experiments that offer consistent answers. In some instances, it seems like the IDR is necessary for direction of the protein into the nucleolus, whereas substitution of arginine residues (or their deletion) results in granular immunostaining dissimilar to wildtype. The current data would indicate that (a) loss of the IDR excludes WT protein from the nucleus, (b) a function that is compensated for by reintroduction of the portion of the IDR proximal to the frameshift and (c) replacement of the acidic residues introduced by the frameshift seem to confer ambiguous results that are difficult to relate to the pattern seen in the full length mutant. This data needs some clarification.

We thank the reviewer for the constructive criticism that guided us to new important new insights. We performed extensive mutagenesis of the BPTAS-associated frameshift mutant HMGB1 protein, and identified the sequence features responsible for nucleolar mispartitioning and nucleolar arrest.

As expected, deleting the whole IDR from wt HMGB1 leads to loss of nuclear localization, as the second NLS of HMGB1 is lost with this truncation (new Fig. 3e-f). Deleting only the sequence after the frameshift (del FS) retains most of the NLS and nuclear localization is maintained (new Fig. 3f). However, EGFP signal is not as homogeneous as in the case of the wild type, so the acidic C-terminal tail has an influence on HMGB1 distribution in the nucleus, but not to a same degree as the introduction of the mutant sequence. (Normalized SD of mutant full length is 5.5X larger than wild type full length, when the normalized SD of del FS has 1.4X larger than WT, displayed in new Extended Data Fig. 9k). In addition, FRAP experiments revealed that del FS variant recovers with similar rate as WT HMGB1, showing that impaired molecular exchange kinetics of mutant HMGB1 are not due to the loss of C-terminal acidic tail (new Fig. 3g).

ii. It would add compelling evidence to disambiguate whether it is the loss of the acidic domain (e.g. like FIB), the gain of the basic domain (? not just Arg - will polylysine suffice?) or both, that mediate the characterised phase transition and rRNA biogenesis defects if similar mutagenesis experiments were performed for some of the other 6 exemplars. What is it about RUNX1 that its mutant does not locate to the FIB1 positive nucleolar compartment?

As suggested by the reviewer, our arginine replacement mutants now include R > K variant, that fails to produce similar nuclear inclusions as the original full length mutant, mutant IDR, or K > R variant (new Fig. 3f, new Extended Data Fig. 9k), suggesting that the arginines are the most important contributor to the formation of nuclear inclusions. While R del, R > A and R > K variants localize differently than the full length mutant, they show comparably slow FRAP recovery (new Fig. 3g) indicative of a gel-like state.

The C-terminal end of the mutant consisting nearly exclusively of hydrophobic amino acids, which we call Hydrophobic Patch, appears responsible for the nucleolar arrest. “Patchless” variant still forms similar nuclear inclusions as full length HMGB1 (new Fig. 3f, new Extended Data Fig. 9k and new Extended Data Figure 10e-h) but this variant recovers within ~20s after photobleaching (new Fig. 3g, new Extended Data Fig. 10g) suggesting that nuclear inclusions formed by “Patchless” variant are liquid-like. Importantly, expression of “Patchless” mutant is not cytotoxic (new Fig. 3j, new Extended Data Fig. 10i-j). These results collectively revealed that the arginine enrichment in the mutant tail drives nucleolar mispartitioning, and the hydrophobic patch contributes to the nucleolar arrest.

When it comes to RUNX1, we do observe an increase in nucleolar localization of mutant RUNX1, since the wt RUNX1 is devoid from FIB1+ compartment and mutant RUNX1 can be found within (Fig. 3e, Extended Data Fig. 19a), although it is not enriched as strongly as some of the other tested frameshift variants. We have revised this in the text.

Last we note that we attempted to predict hydrophobic patches in frameshift sequences, and include these information in Supplementary Table 5. However, we note that the features of the hydrophobic patch in HMGB1 that contribute to the nucleolar arrest are unknown, so the prediction tools are rudimentary at best. We measured nucleolar dynamics using FRAP in cells expressing eleven additional mutant proteins and found evidence for nucleolar arrest for several (but not all of them). Dissecting the precise sequence features that cause nucleolar arrest will be an important goal for follow up work.

6. Final conclusions.

The concluding sentences of the abstract (rightly, in my view) summarises the findings of this work as suggestive of a mechanism for many different conditions with causative alleles positioned within C-terminal IDRs. This contrasts with the implied certainty within the title of this paper which should be changed with the current level of evidence offered in this manuscript.

We adjusted the title in the revised manuscript.

References

- Calogero, S., Grassi, F., Aguzzi, A., Voigtlander, T., Ferrier, P., Ferrari, S., and Bianchi, M.E. (1999). The lack of chromosomal protein Hmg1 does not disrupt cell growth but causes lethal hypoglycaemia in newborn mice. *Nat Genet* *22*, 276-280.
- Kang, R., Chen, R., Zhang, Q., Hou, W., Wu, S., Cao, L., Huang, J., Yu, Y., Fan, X.G., Yan, Z., *et al.* (2014). HMGB1 in health and disease. *Mol Aspects Med* *40*, 1-116.
- Kurosaki, T., and Maquat, L.E. (2016). Nonsense-mediated mRNA decay in humans at a glance. *J Cell Sci* *129*, 461-467.
- Lindeboom, R.G., Supek, F., and Lehner, B. (2016). The rules and impact of nonsense-mediated mRNA decay in human cancers. *Nat Genet* *48*, 1112-1118.
- Litchfield, K., Reading, J.L., Lim, E.L., Xu, H., Liu, P., Al-Bakir, M., Wong, Y.N.S., Rowan, A., Funt, S.A., Merghoub, T., *et al.* (2020). Escape from nonsense-mediated decay associates with anti-tumor immunogenicity. *Nat Commun* *11*, 3800.
- Sofiadis, K., Josipovic, N., Nikolic, M., Kargapolova, Y., Ubelmesser, N., Varamogianni-Mamatsi, V., Zirkel, A., Papadionysiou, I., Loughran, G., Keane, J., *et al.* (2021). HMGB1 coordinates SASP-related chromatin folding and RNA homeostasis on the path to senescence. *Mol Syst Biol* *17*, e9760.
- Uguen, K., Krysiak, K., Audebert-Bellanger, S., Redon, S., Benech, C., Viora-Dupont, E., Tran Mau-Them, F., Rondeau, S., Elsharkawi, I., Granadillo, J.L., *et al.* (2021). Heterozygous HMGB1 loss-of-function variants are associated with developmental delay and microcephaly. *Clin Genet* *100*, 386-395.

Reviewer Reports on the First Revision:

Referees' comments:

Referee #1 (Remarks to the Author):

The revised manuscript is significantly improved from the original submission; the storyline is more robust, with additional clinical evidence, more in depth mechanistic insights into the sequence features that drive aberrant nucleolar localization of mutant HMGB1, and enriched evidence for the generality of the frameshift/aberrant nucleolar localization pathological mechanism.

Specific comments/suggestions:

- Please check the age of patient #3 in SI Table 2 (line 7): 44328 (???)
- The addition of the SEC step in the purification protocol significantly improved the quality of the protein. I thank the authors for taking the extra steps.
 - o I was intrigued by the difference in the SEC elution profiles of the WT vs the mutant, which suggest that the mutant exists in solution as a soluble oligomer, in contrast with WT, which elutes as a smaller species. Perhaps this warrants a comment in the text.
 - o Piggybacking on Referee #2's comment regarding the potential contribution of the AlphaFold (and also CD-backed) alpha-helix conformation of the mutant "IDR" – it could be possible that the transient alpha-helix is stabilized by intermolecular interaction via coiled-coil (?) interactions; this scenario would explain the presence of soluble oligomers, the reduced FRAP-measured dynamics in nucleoli and in vitro condensates and the preference for homotypic phase separation under heterotypic conditions for the mutant HMGB1.
 - o Both the SEC and WB show clearly 2 species for both the WT and mutant, which suggest that both proteins tend to have part of their HMGB1 C-terminus truncated (anti-EGFP and anti-HMGB1 both probe the N-terminal half of the WT and mutant fusion proteins). It would be great if the authors could comment on the location of the truncation (this can be mapped via intact MS) and any potential alternative interpretations of the in vitro results in the light of the convoluted contribution of the full-length and truncation species.
- This comment is inconsequential for the decision on this manuscript, but rather a general recommendation for future submissions/resubmissions: as a reviewer, I find it helpful to see highlighted the sections that have been modified with respect to the original submission; this can be easily achieved by formatting the edited sections differently (e.g., different color, underline)

Additional comments on concerns by Reviewer #2:

I do not think the mouse study is necessary to support this story. It is my understanding that the main concern of Reviewer #2 is rooted in the assumption that disruption of nucleolar function equates incompatibility with life. I do not share that concern. The function and composition of the nucleolus is often altered in many diseases, in various ways; for example, in AML with NPM1c+ translocation the nucleolar scaffold NPM1 is delocalized from the nucleolus to the cytoplasm; in solid cancers the nucleolus is enlarged; in viral infections the nucleolar function is hijacked by the virus and in ALS C9orf72 the (PR)_n dipeptide repeats accumulate in the nucleolus and hinder ribosome biogenesis and dynamics of nucleolar components. These are just a few examples, in addition to those that the authors presented that demonstrate that nucleolar disruption need not

necessarily lead to death.

I do not think that a mouse model is necessary to support the data and conclusions in this manuscript. I also am not convinced that the results from a mouse model would more clearly make the connection between points i-ii and iii-v below. If the mouse model is created with a hybrid sequence that contains the mouse 3' UTR and the human coding sequence, I am concerned that it would cause challenges in interpretation and in the case of a negative result, it would not unequivocally answer the reviewer's question, as several alternative explanations could contribute to this result, including differences in the regulation of localization via interactions with the 3' UTR. Furthermore, the grammar of the nucleolar localization of the mutant HMGB1 and misfunction is in line with previous reports showing that localization of proteins to the nucleolus has a strong electrostatic aspect and that multivalency in arginine patches is one of the driving forces: <https://www.ncbi.nlm.nih.gov/pmc/articles/PMC4615656/>; <https://pubmed.ncbi.nlm.nih.gov/26836305/>

Referee #2 (Remarks to the Author):

The authors have replied to my questions, but overall I am not satisfied with their answers. See below.

The authors suggest a number of reasons why the mutant HMGB1 is not necessarily incompatible with life. The only one that looks reasonable to me is that the humans with the mutation are indeed alive. My question was rather: why are they alive if the mutant protein severely interferes with the nucleolus? Thus, it is the interference with the nucleolus that I question, not the fact that the patients are alive. And this is why I suggested a mouse model. Although it true that "... the mouse HMGB1 sequence contains a different 3'UTR, therefore, the same frameshift mutation results in a protein sequence in mice that is substantially different from the sequence created by the same frameshift in the human sequence", this does not making a mouse model any more difficult than most other mutations. One could simply add the human mutant sequence to the mouse protein, and leave the 3'UTR alone. Comparisons with the current mouse models is not warranted, since in those HMGB1 is completely deleted and thus is a LOF, and their human mutants HMGB1 is a GOF.

They also say "The identification of the same de novo variant in several unrelated individuals with an ultra-rare congenital malformation syndrome is considered as evidence for the pathogenicity of such variant." I agree, but, if anything, the finding of same de novo mutations several times tells us that that SPECIFIC variant is pathogenic, not that the transformation of the acidic tail into a basic tail, which could be obtained in many other ways, is pathogenic. This undermines my confidence that the real cause of BPTAS is the nucleolar disfunction.

I also question the logic of using the AlphaFold2 prediction of HMGB1 structure when its structure is indeed known. See e.g. doi:10.1016/j.jmb.2010.07.045

Most importantly, BoxB is NOT partially unfolded.

I also understand that the authors would like to have a reasonable prediction of what happens to the mutant protein, but Alphafold2 IS NOT useful for that. See a clear explanation in DOI:10.1101/2021.09.19.460937

Indeed, their CD experiments prove that the basic tail in the mutant has a VERY LOW propensity to being alpha-helical. A very questionable alpha-helical fingerprint appears only if TFE is added. TFE is an agent that INDUCES folding of peptides into alpha-helices, so I would say that the experiment shows that the mutant basic tail is NOT alpha-helical unless forced.

Supplementary Fig 8b,c shows two bands, but wt HMGB1 is monomeric, so is HMGB1-GFP and I would suppose mtHMGB1-GFP as well. If this is the status of the purified HMGB1s, they are not sufficiently pure for phase separation experiments.

The precipitation of RNA with a basic peptide (Supplementary Fig 8d-f) is absolutely expected, and means nothing relevant for the present paper.

The irregular shape of the mt-HMGB1-GFP condensates and the lack of FRAP recovery (Fig. 2) suggests to me that mtHMGB1-GFP does not form droplets, but precipitates. This is biophysically as different as oil and water. Fig 3 also suggests that mtHMGB1 precipitates on top of FIB1.

There is NO disruption in translation in cells expressing mtHMGB1; there is some, but rather minor, disruption caused by mt-GMGB1-GFP. This does not prove that mtHMG1 causes translation problems. The suggestion that mtHMGB1 causes nucleolar dysfunction is severely undermined by this experiment.

Overall, I am now fairly convinced that:

- i. the mutations described cause BPTAS
- ii. the mutations cause the conversion of an unstructured acidic tail into an unstructured basic tail
- iii. wt HMGB1 has a propensity to phase-separate
- iv. mtHMGB1 does not phase separate but forms precipitates in vitro and inclusions in vivo. These effects are only visible with overexpressed mtHMGB1 protein, and are rather minor. It is not even clear that there is a serious nucleolar dysfunction.
- v. basic tails appended to many proteins cause the same cellular phenotypes.

However, I find no relation between i,ii and iii-v. I also find that the suggestion of a grammar of nucleolar dysfunction is insufficiently supported by these data.

Referee #3 (Remarks to the Author):

This paper is substantially improved from its original version with the flow and contiguity much better. A significant improvement has been the mutagenesis experiments that identify different

functions that are affected by the Arg residues as distinct from a newly recognised hydrophobic patch. The results from transfection of the patchless construct is particularly illuminating. The human genetics on the BTPS phenotype is now very well presented and conclusive in its findings.

The cellular biology is easier to follow and the addition of extra data, especially the addition of the puromycin labelling data has addressed many of the concerns I had regarding the nature of the nucleolar dysfunction.

My only remaining suggestions relate to the bioinformatic analysis of the mutation databases.

1. The authors used the COSMIC database to assemble their lists of variants for analysis. This database catalogues somatic variants found in cancer some of which will be driver mutations but many will be passenger variants of no functional effect (necessarily). How did the authors decide which frameshifts and SNVs they found from this source were pathogenic or otherwise? I wonder if the COSMIC data should be excluded from this exercise.

2. The authors present a nice diagrammatic representation of the fraction of alleles located in C-terminal IDRs with different mutational mechanisms and variant sequences downstream of the variant and how many of these are assigned as pathogenic. The fraction of alleles that are assigned to be pathogenic is given as a percentage parenthetically, and there is an apparent increase in this fraction as the variant and derivative sequence is shown to have the characteristics of the HMGB1 frameshifts studied in detail here. The authors described this as "higher-than-average pathogenicity". This series of data requires some statistical analysis. Each category will have a prior probability assignable to it according to a null hypothesis and therefore a P value will be able to be calculated for each category. These should be presented. Since the authors present data suggesting the significance of the appearance of a hydrophobic patch is important, I would like to understand probability of such a patch appearing by chance is as well.

3. I still have difficulty in accepting the semiquantitative assertion that these data suggest "hundreds of disease-associated and common variants operate by this mechanism". It seems like a bit of a "shoot-from-the-hip" statement predicated on the functional evaluation of just 11 examples extracted from sets of data that have their imperfections and that no "common variants" are evaluated functionally. I think the significance of the data speak for themselves without the need for this speculative extrapolation.

My congratulations to the authors. This is fine work.

=====

Additional comments to referee 2's concerns:

1. Given the proposed nucleolar dysfunction is it plausible that patients with BPTAS are alive (casting into doubt the causative link between the observed variants and the BPTAS phenotype)?

Many human syndromes exist that are similar to BPTAS in that they exert nucleolar dysfunction as their prime pathogenic mechanism. Like BPTAS, some are due to heterozygosity for a pathogenic allele. Although in the initial draft of this paper the evidence of nucleolar dysfunction was (to me) disconcertingly subtle, the addition of puromycylation experiments have solidified this contention to

my satisfaction. I don't identify any incompatibility between the proposed pathogenic mechanism underlying BPTAS and its survivability, given these new data and the precedent set by other disorders with nucleolar dysfunction such as Treacher Collins syndrome (and the other conditions noted by the authors in their rebuttal).

2. Is a mouse model necessary to prove that heterozygosity for pathogenic variants in HMGB1 are causative of BPTAS?

I agree with Reviewer 2 that existing mouse models are not useful to answer this question. They propose a heterologous construct for such a mouse model where the C terminus of the protein is humanized with a relevant mutation introduced. This would be plausible.

However, in my view the study of such a mouse model will add little to the evidence presented in this paper relating to the question whether or not the two different variants found in 5 BPTAS patient are causative of the phenotype. This is for at least three reasons:

1. The demonstration of precisely the same variant arising as de novo events in 4 independent families in addition to a variant that lies very close to the first and predicts that same functional consequence, is compelling evidence for a causal relationship between these variants and the observed phenotype. The rarity of the phenotype (and its specificity) alongside these genetic observations further reinforces this confidence.
2. The absence of these variants in large population databases of healthy controls and furthermore the absence of variants that predict similar functional consequences bring orthogonal levels of evidence to this conclusion.
3. A mouse model enables the testing of a single variant on defined in singular background. Failure to replicate the same or a similar phenotype using this approach has plenty of precedence relating most likely to interspecies differences and certainly would not trump what I consider to be the unassailable human genetic evidence brought by the description of 2 alleles, all de novo, on 5 different "genetic backgrounds" delivering a highly concordant, extremely rare phenotype.

In conclusion I think that the evidence that the two alleles described in this paper cause BPTAS is unassailable. The recurrency of one of these alleles is consistent with a mechanism that may only be induced by a very restricted range of variants at the HMGB1 locus. In this respect I disagree with Reviewer 2 in that just because only 2 alleles have been described, this speaks against the claim for the proposed pathogenic mechanism because a multiplicity of alternatives must exist if the phase-change hypothesis is supported. A large range of alternatives have been tested by these authors using in vitro mutagenesis and survey of standing variation in humans at this locus using large databases. A reasonable conclusion is that a restricted range of mutational alternatives are only available to convert the tail of HMGB1 from acidic to basic (and whatever other qualities are required such as length). The reviewer suggests that this is possible in many other ways but the presented bioinformatic and experimental evidence argues that such alternatives are not apparent. A study of a single mouse allele would not advance this matter any further.

In summary, I do not share Reviewer 2's concerns and insistence that a mouse model is required for the claims in this paper to be supported.

Author Rebuttals to First Revision:

Referees' comments:

Referee #1 (Remarks to the Author):

The revised manuscript is significantly improved from the original submission; the storyline is more robust, with additional clinical evidence, more in depth mechanistic insights into the sequence features that drive aberrant nucleolar localization of mutant HMGB1, and enriched evidence for the generality of the frameshift/aberrant nucleolar localization pathological mechanism.

Specific comments/suggestions:

- Please check the age of patient #3 in SI Table 2 (line 7): 44328 (???)

We checked and fixed the numbers in SI Table 2.

- The addition of the SEC step in the purification protocol significantly improved the quality of the protein. I thank the authors for taking the extra steps.

We further verified the quality of the purified proteins with Mass Spectrometry, as detailed below.

- I was intrigued by the difference in the SEC elution profiles of the WT vs the mutant, which suggest that the mutant exists in solution as a soluble oligomer, in contrast with WT, which elutes as a smaller species. Perhaps this warrants a comment in the text.

We added a comment that the mutant protein may form soluble oligomers in the “Protein purification and peptide synthesis” paragraph in the Methods section.

- Piggybacking on Referee #2’s comment regarding the potential contribution of the AlphaFold (and also CD-backed) alpha-helix conformation of the mutant “IDR” – it could be possible that the transient alpha-helix is stabilized by intermolecular interaction via coiled-coil (?) interactions; this scenario would explain the presence of soluble oligomers, the reduced FRAP-measured dynamics in nucleoli and in vitro condensates and the preference for homotypic phase separation under heterotypic conditions for the mutant HMGB1.

We agree that the idea of the mutant protein having a propensity to form soluble oligomers in part through coiled-coil interactions of the C-terminal sequence is indeed plausible. We added a sentence acknowledging this possibility in the “Protein purification and peptide synthesis” paragraph in the Methods section. We note that DeepCoil indeed predicts increased probability for coiled coils within the mutant HMGB1 sequence, but other tools (DeepCoil2, Marcoil, Waggawagga) do not. Therefore, experimental confirmation of coiled-coil interaction will require future work.

- Both the SEC and WB show clearly 2 species for both the WT and mutant, which suggest that both proteins tend to have part of their HMGB1 C-terminus truncated (anti-EGFP and anti-

HMGB1 both probe the N-terminal half of the WT and mutant fusion proteins). It would be great if the authors could comment on the location of the truncation (this can be mapped via intact MS) and any potential alternative interpretations of the in vitro results in the light of the convoluted contribution of the full-length and truncation species.

We thank the Reviewers for pressing on the issue of protein purity, which is undoubtedly important. Overall, several lines of evidence suggest that the presence of two separate bands on the Western blots is associated with features of HMGB1 and the presence of a HisTag, and likely do not represent truncation of the protein product. We reviewed available information on purified recombinant HMGB1 in detail at the response to Reviewer 2 on the same issue.

We performed Mass Spectrometry analysis of the individual bands observed on the protein gel (Extended Data Fig. 4b). We found that both for the WT and Mutant fusion proteins, the peptide coverage, including the C-terminal part, is virtually identical for both the high molecular weight and lower bands (Reviewer Figure 1 below). We note that because of the sequence features, the last 51 amino acids (i.e. the post-frameshift sequence) are invisible in the Mass Spectrometry experiment for both the wild type and mutant proteins. However, even if the last 51 amino acids were truncated, it would not explain the apparent difference in the high and lower bands of ~10 kDa (Extended Data Fig. 4b-c). The N-terminal part of the protein is likely not truncated, as the HisTag used for purification is on the N-terminus. We added a sentence on the Mass Spectrometry confirmation in the “Protein purification and peptide synthesis” paragraph in the Methods section.

Reviewer Figure 1. Mass Spectrometry analysis of purified recombinant HMGB1 proteins.

The higher molecular weight and lower bands (see Extended Data Fig. 4b) of WT and Mutant HMGB1 were excised and analyzed separately. The colored blocks represent peptides detected in the experiment, and the depth of color is proportional to the number of times a peptide was detected. Note that the coverage map only includes position 90-430 of the fusion proteins, and lack the last 51 C-terminal amino-acids. Regardless, the coverage maps of the high and lower bands are virtually identical.

- This comment is inconsequential for the decision on this manuscript, but rather a general recommendation for future submissions/resubmissions: as a reviewer, I find it helpful to see highlighted the sections that have been modified with respect to the original submission; this can be easily achieved by formatting the edited sections differently (e.g., different color, underline)

Duly noted. We highlighted all changes in blue font in the newly revised manuscript document.

Additional comments on concerns by Reviewer #2:

I do not think the mouse study is necessary to support this story. It is my understanding that the main concern of Reviewer #2 is rooted in the assumption that disruption of nucleolar function equates incompatibility with life. I do not share that concern. The function and composition of the nucleolus is often altered in many diseases, in various ways; for example, in AML with NPM1c+ translocation the nucleolar scaffold NPM1 is delocalized from the nucleolus to the cytoplasm; in solid cancers the nucleolus is enlarged; in viral infections the nucleolar function is hijacked by the virus and in ALS C9orf72 the (PR)n dipeptide repeats accumulate in the nucleolus and hinder ribosome biogenesis and dynamics of nucleolar components. These are just a few examples, in addition to those that the authors presented that demonstrate that nucleolar disruption need not necessarily lead to death.

I do not think that a mouse model is necessary to support the data and conclusions in this manuscript. I also am not convinced that the results from a mouse model would more clearly make the connection between points i-ii and iii-v below. If the mouse model is created with a hybrid sequence that contains the mouse 3' UTR and the human coding sequence, I am concerned that it would cause challenges in interpretation and in the case of a negative result, it would not unequivocally answer the reviewer's question, as several alternative explanations could contribute to this result, including differences in the regulation of localization via interactions with the 3' UTR. Furthermore, the grammar of the nucleolar localization of the mutant HMGB1 and misfunction is in line with previous reports showing that localization of proteins to the nucleolus has a strong electrostatic aspect and that multivalency in arginine patches is one of the driving forces:

[https://www.ncbi.nlm.nih.gov/pmc/articles/PMC4615656/;](https://www.ncbi.nlm.nih.gov/pmc/articles/PMC4615656/)

<https://pubmed.ncbi.nlm.nih.gov/26836305/>

Referee #2 (Remarks to the Author):

The authors have replied to my questions, but overall I am not satisfied with their answers. See below.

The authors suggest a number of reasons why the mutant HMGB1 is not necessarily incompatible with life. The only one that looks reasonable to me is that the humans with the mutation are indeed alive. My question was rather: why are they alive if the the mutant protein severely interferes with the nucleolus? Thus, it is the interference with the nucleolus that I question, not the fact that the patients are alive. And this is why I suggested a mouse model. Although it true that "... the mouse HMGB1 sequence contains a different 3'UTR, therefore, the same frameshift mutation results in a protein sequence in mice that is substantially different from the sequence created by the same frameshift in the human sequence", this does not making a mouse model any more difficult than most other mutations. One could simply add the human mutant sequence to the mouse protein, and leave the 3'UTR alone. Comparisons with the current mouse models is not warranted, since in those HMGB1 is completely deleted and thus is a LOF, and their human mutants HMGB1 is a GOF.

Several human genetic diseases are associated with nucleolar dysfunction, therefore nucleolar dysfunction does not generally appear to be incompatible with life in humans. For example, Fanconi Anemia caused by mutations in the FANCI or FANCA genes is associated with nucleolar defects ^{1,2}, and mutations in DDX21 cause Treacher-Collins syndrome associated with nucleolar dysfunction including rRNA processing defects ^{3,4}. There is evidence for nucleolar dysfunction in other rare diseases e.g. Woodhouse-Sakati syndrome ^{5,6}. Some of these diseases have partially overlapping symptoms with BPTAS, which we described in detail in the Supplementary Notes as per a previous request by Reviewer 3.

The key question of the reviewer is “why are they alive if the mutant protein severely interferes with the nucleolus?” In the previous response letter, we speculated that the patients are heterozygous for the mutation and that the cells might adapt to the presence of one mutant *HMGB1* allele using transcriptional or post-transcriptional mechanisms to suppress production of mutant HMGB1. Such suppression mechanisms may help explain why cells (or BPTAS individuals) encoding the mutant *HMGB1* allele are viable. We have preliminary data consistent with this hypothesis. We established a stable lymphoblastoid cell line (LCL) from a BPTAS individual, and found that the total level of HMGB1 protein appears lower in the BPTAS LCL cells compared to LCL cells from a control individual (see Reviewer Figure 2 on the next page). We are happy to perform e.g. allele-specific qRT-PCR analysis to test whether the mutant allele produces lower levels of mRNA, if necessary. In general, we would prefer not to include data from the LCL cells at this stage, because BPTAS is not associated with dysfunction of lymphoid cells, therefore the cells may not represent a good model system for the phenotype (characterized by skeletal and developmental defects as described in Table S2).

We are working on knocking in the mutant human *HMGB1* gene into the *HMGB1* locus in mouse embryonic stem cells to generate a mouse model, using a strategy similar to what is proposed by the Reviewer. We argue that the mouse model is outside the scope of this study.

Reviewer Figure 2. Western blot analysis of HMGB1 in immortalized LCL cells derived from a (left) control individual and (right) BPTAS patient. GAPDH is shown as a loading control. Note that the antibody does not distinguish between wild type and mutant HMGB1 proteins.

They also say "The identification of the same de novo variant in several unrelated individuals with an ultra-rare congenital malformation syndrome is considered as evidence for the pathogenicity of such variant." I agree, but, if anything, the finding of same de novo mutations several times tells us that that SPECIFIC variant is pathogenic, not that the transformation of the acidic tail into a basic tail, which could be obtained in many other ways, is pathogenic. This undermines my confidence that the real cause of BPTAS is the nucleolar dysfunction.

We have characterized over 1,100 *HMGB1* variants reported in the gnomAD database. Not a single one of these variants, including 43 non-synonymous variants, is predicted to transform the acidic tail into a basic tail (new Extended Data Fig. 2f-g). There are three frameshift variants described in *HMGB1* in the literature ⁷. These frameshifts occur upstream of the position of the BPTAS-frameshifts, and for at least two of the three variants there is evidence that the RNA product is substrate for nonsense-mediated decay (NMD). These variants cause a neurodevelopmental phenotype similar to the phenotype caused by heterozygous deletion of the entire *HMGB1* locus [new Extended Data Fig. 2h-j, Table S2, and ⁷]. Overall, we argue that the genetics data is clear: no disease-associated or common variant described to date in *HMGB1* leads to the transformation of the acidic tail into a basic tail, except for the BPTAS variants we report. We explain this in more detail in the "Genotype-phenotype correlation of HMGB1 variants" section in the Supplementary Notes.

Finally, we would like to clarify that we describe not one, but two very similar frameshift variants in BPTAS individuals. The two variants produce basic tails that only differ in one amino acid (R/K in position 185), as shown in Fig. 1d, but are nevertheless different variants, further supporting that the transformation of the acidic tail into a basic tail causes BPTAS.

I also question the logic of using the AlphaFold2 prediction of HMGB1 structure when its structure is indeed known. See e.g. doi:10.1016/j.jmb.2010.07.045. Most importantly, BoxB is NOT partially unfolded. I also understand that the authors would like to have a reasonable prediction of what happens to the mutant protein, but AlphaFold2 IS NOT useful for that. See a clear explanation in DOI:10.1101/2021.09.19.460937. Indeed, their CD experiments prove that the basic tail in the mutant has a VERY LOW propensity to being alpha-helical. A very questionable alpha-helical fingerprint appears only if TFE is added. TFE is an agent that INDUCES folding of peptides into alpha-helices, so I would say that the experiment shows that the mutant basic tail is NOT alpha-helical unless forced.

The concerns of the Reviewer of using AlphaFold2 to predict structure are duly noted. As the Reviewer noted, we mostly wanted to get insights into the structure of the mutant protein using AlphaFold2, and we did include experimental CD data that are consistent the AlphaFold2

prediction. Furthermore, as noted by Reviewer 1, a transient alpha-helix of the mutant tail stabilized by intermolecular coiled-coil interactions could explain much of the observed behavior of the mutant protein. In general, we agree with the reviewer that the CD experiments show a very low propensity of the mutant to be alpha-helical, but the data do show propensity that is different than the propensity of the wild type protein (new Extended Data Fig. 3c-d). We changed the clause “...moderately elevated propensity of the C-terminal portion of the IDR to assume a helical conformation in the frameshift mutant HMGB1...” with “...a slight propensity of the C-terminal portion of the IDR to assume a helical conformation in the frameshift mutant HMGB1...” in the main text. (*Changes underlined*).

Supplementary Fig 8b,c shows two bands, but wt HMGB1 is monomeric, so is HMGB1-GFP and I would suppose mtHMGB1-GFP as well. If this is the status of the purified HMGB1s, they are not sufficiently pure for phase separation experiments.

We much appreciate the thorough inspection of the data by the Reviewer. We stress that both the wt-HMGB1-GFP and mut-HMGB1-GFP recombinant proteins elute in a single dominant peak after the size exclusion chromatography, suggesting high purity of the proteins (new Extended Data Fig. 4a). Running as a double band on a protein gel is documented for HMGB1. For example, commercially available, pure HMGB1 proteins run as double bands on a protein gel (see e.g. tinyurl.com/26bezsbn, tinyurl.com/4sn5mf82, tinyurl.com/2bf249nh). We note that the double bands occur when HMGB1 is purified with a HisTag, as commercially available recombinant HMGB1 protein purified with an Fc-tag runs as a single band (see e.g. tinyurl.com/6jr53vre). We argue that the HisTag does not substantially alter the biochemistry results, as we reproduced all key findings using synthetic HMGB1 IDR peptides that are highly pure and lack any affinity tag (new Extended Data Fig. 4d-i).

Moreover, as described in detail at the response to Reviewer 1, Mass Spectrometry analysis of the individual bands excised from the protein gel suggest that both the high and lower molecular weight products are likely full-length fusion proteins (see also Reviewer Figure 1, above).

The precipitation of RNA with a basic peptide (Supplementary Fig 8d-f) is absolutely expected, and means nothing relevant for the present paper.

We agree that precipitation of RNA with the basic peptide (new Extended Data Fig. 4d-f) is expected, given the extensive literature of basic nucleolar proteins. The cited data on the basic peptide serves as an independent experiment confirming the experiments on the purified, full-length proteins in Fig. 2c-d, showing that in solution, the mutant HMGB1 protein forms condensates at a substantially lower concentration than the wild type. Also, the synthetic peptide is >90% pure per the manufacturer's QC, which may also help address the purity issues raised by the Reviewer above.

The irregular shape of the mt-HMGB1-GFP condensates and the lack of FRAP recovery (Fig. 2) suggests to me that mtHMGB1-GFP does not form droplets, but precipitates. This is biophysically as different as oil and water. Fig 3 also suggests that mtHMGB1 precipitates on top of FIB1.

We acknowledge that the Reviewer has a valid point, the consequences of which appear minor, and can be resolved by clarifying terminology. We agree that the precise biophysical property of the mt-HMGB1-GFP condensates in Fig. 2 is unclear, and the lack of substantial FRAP recovery indicates that the condensates may be precipitates. We also note that around 10% of the fluorescence signal does recover within 2 minutes after the photobleaching of the mtHMGB1-GFP condensates, indicating that there is minimal, but measurable mobile fraction of molecules within the condensates (Fig. 2e). If necessary, we are happy to perform longer FRAP experiments, but believe that this is a minor issue. Our major finding is that the phase separation capacity of the wild type protein is altered by the mutation, which is supported by the biochemistry data. We have revised the wording, including referring to the mutant HMGB1 in vitro condensates as “condensates” in the main text, and adjusted the language in the summary paragraph accordingly.

There is NO disruption in translation in cells expressing mtHMGB1; there is some, but rather minor, disruption caused by mt-GMGB1-GFP. This does not prove that mtHMG1 causes translation problems. The suggestion that mtHMGB1 causes nucleolar dysfunction is severely undermined by this experiment.

We believe this comment is caused by misinterpretation of the data of the puromycilation experiment (Fig. 3h-i, new Extended Data Fig. 6b-c), which we are happy to clarify.

In these experiments, cell cultures are transfected either with wt-HMGB1-GFP or mut-HMGB1-GFP. After one day of incubation, the media is replaced with media containing puromycin to perform 15-minute pulse labeling of nascently translated peptide chains. Afterwards, cells were fixed and stained for puromycin. The GFP signal is also visualized and recorded. The GFP signal is then used to identify transfected (i.e. GFP+), and untransfected (i.e. GFP-) cells. The data revealed that untransfected and transfected cells in the culture transfected with the wt-HMGB1-GFP have similar levels of puromycin intensity. In contrast, in the cultures transfected with mut-HMGB1-GFP, the transfected (i.e. GFP+) cells have on average significantly lower level of puromycin signal compared to the untransfected (i.e. GFP-) cells in the same population. The (GFP-) cells are thus used as an internal control. The data were collected for over 30,000 cells in three independent transfection experiments. Overall, the data suggest that cells expressing mut-HMGB1-GFP have lower global nascent translation compared to untransfected cells in the same cell population. We note that the puromycilation has been used for decades to measure global nascent translation, which is an indirect readout of ribosome biogenesis and nucleolar function ⁸.

We believe that the confusion is caused by the Reviewer interpreting the “Mutant (GFP-)” bar in Fig. 3i as cells expressing the mutant HMGB1 without the GFP tag. In reality, these are cells that do NOT express mut-HMGB1-GFP in the cell population.

We changed the name of the bars in Fig. 3i to clearly identify data of transfected (GFP-) cells, and highlighted GFP+ and GFP- cells in the images in Fig 3h.

Overall, I am now fairly convinced that:
i. the mutations described cause BPTAS

- ii. the mutations cause the conversion of an unstructured acidic tail into an unstructured basic tail
- iii. wt HMGB1 has a propensity to phase-separate
- iv. mtHMGB1 does not phase separate but forms precipitates in vitro and inclusions in vivo. These effects are only visible with overexpressed mtHMGB1 protein, and are rather minor. It is not even clear that there is a serious nucleolar dysfunction.
- v. basic tails appended to many proteins cause the same cellular phenotypes.

However, I find no relation between i,ii and iii-v. I also find that the suggestion of a grammar of nucleolar dysfunction is insufficiently supported by these data.

We have provided responses to the concerns in detail above. In brief, no disease-associated or common variant described to date in *HMGB1* leads to the transformation of the acidic tail into a basic tail, except for the BPTAS variants we report here, suggesting a relationship between i,ii. Furthermore, we clarified the puromycylation experiments that provide key evidence for nucleolar dysfunction in cells expressing mutant HMGB1 protein, which we believe were previously misinterpreted by the Reviewer. Moreover, we note that nucleolar mispartitioning is evident in cells that express very low levels of the mutant protein (see e.g. new Extended Data Fig. 5d, or 5h) and is virtually never observed in live cells that overexpress ectopic wild type HMGB1 (see e.g. new Extended Data Fig 5a-b). Finally, we included and described additional data further strengthening the insights above.

Referee #3 (Remarks to the Author):

This paper is substantially improved from its original version with the flow and contiguity much better. A significant improvement has been the mutagenesis experiments that identify different functions that are affected by the Arg residues as distinct from a newly recognised hydrophobic patch. The results from transfection of the patchless construct is particularly illuminating. The human genetics on the BTPS phenotype is now very well presented and conclusive in its findings.

The cellular biology is easier to follow and the addition of extra data, especially the addition of the puromycin labelling data has addressed many of the concerns I had regarding the nature of the nucleolar dysfunction.

My only remaining suggestions relate to the bioinformatic analysis of the mutation databases.

1. The authors used the COSMIC database to assemble their lists of variants for analysis. This database catalogues somatic variants found in cancer some of which will be driver mutations but many will be passenger variants of no functional effect (necessarily). How did the authors decide which frameshifts and SNVs they found from this source were pathogenic or otherwise? I wonder if the COSMIC data should be excluded from this exercise.

The information used to annotate pathogenic variants in the catalog is solely from ClinVar. This means that a variant is annotated as “pathogenic” in the catalog only if ClinVar annotated the variant as pathogenic. The variants that originate from COSMIC indeed could be driver or passenger mutations, but some variants could be prioritized based on ClinVar information. For example, 3175 frameshifts are annotated in COSMIC, and 29 of these variants are found in ClinVar annotated as pathogenic. Also, the COSMIC variants that generate an HMGB1-like mutant tail could be prioritized as potential driver mutations and tested in further studies. For these reasons we would prefer keeping the COSMIC variants in the catalog.

2. The authors present a nice diagrammatic representation of the fraction of alleles located in C-terminal IDRs with different mutational mechanisms and variant sequences downstream of the variant and how many of these are assigned as pathogenic. The fraction of alleles that are assigned to be pathogenic is given as a percentage parenthetically, and there is an apparent increase in this fraction as the variant and derivative sequence is shown to have the characteristics of the HMGB1 frameshifts studied in detail here. The authors described this as “higher-than-average pathogenicity”. This series of data requires some statistical analysis. Each category will have a prior probability assignable to it according to a null hypothesis and therefore a P value will be able to be calculated for each category. These should be presented. Since the authors present data suggesting the significance of the appearance of a hydrophobic patch is important, I would like to understand probability of such a patch appearing by chance is as well.

We now include hypergeometric tests assessing the statistical significance of the enrichment of pathogenic variants among the variants that create sequences with the features shown in Figure 4b. In brief frameshifts, frameshifts that create arginine-rich sequences, and frameshifts that

create arginine-rich sequences and hydrophobic patches are all significantly enriched for pathogenic variants among all variants in C-terminal tails included in our catalog (Fig. 4b).

3. I still have difficulty in accepting the semiquantitative assertion that these data suggest "hundreds of disease-associated and common variants operate by this mechanism". It seems like a bit of a "shoot-from-the-hip" statement predicated on the functional evaluation of just 11 examples extracted from sets of data that have their imperfections and that no "common variants" are evaluated functionally. I think the significance of the data speak for themselves without the need for this speculative extrapolation.

We appreciate this comment. To reduce speculation, we removed said sentence in the summary paragraph, and instead, now write the following: "These data identify the cause of a rare complex syndrome, and suggest that a large number of genetic variants may dysregulate nucleoli and other biomolecular condensates in humans." (Highlighted in blue font in the revised manuscript).

My congratulations to the authors. This is fine work.

=====
Additional comments to referee 2's concerns:

1. Given the proposed nucleolar dysfunction is it plausible that patients with BPTAS are alive (casting into doubt the causative link between the observed variants and the BPTAS phenotype)? Many human syndromes exist that are similar to BPTAS in that they exert nucleolar dysfunction as their prime pathogenic mechanism. Like BPTAS, some are due to heterozygosity for a pathogenic allele. Although in the initial draft of this paper the evidence of nucleolar dysfunction was (to me) disconcertingly subtle, the addition of puromycilation experiments have solidified this contention to my satisfaction. I don't identify any incompatibility between the proposed pathogenic mechanism underlying BPTAS and its survivability, given these new data and the precedent set by other disorders with nucleolar dysfunction such as Treacher Collins syndrome (and the other conditions noted by the authors in their rebuttal).

2. Is a mouse model necessary to prove that heterozygosity for pathogenic variants in HMGB1 are causative of BPTAS?

I agree with Reviewer 2 that existing mouse models are not useful to answer this question. They propose a heterologous construct for such a mouse model where the C terminus of the protein is humanized with a relevant mutation introduced. This would be plausible. However, in my view the study of such a mouse model will add little to the evidence presented in this paper relating to the question whether or not the two different variants found in 5 BPTAS patient are causative of the phenotype. This is for at least three reasons:

1. The demonstration of precisely the same variant arising as de novo events in 4 independent families in addition to a variant that lies very close to the first and predicts that same functional consequence, is compelling evidence for a causal relationship between these variants and the

observed phenotype. The rarity of the phenotype (and its specificity) alongside these genetic observations further reinforces this confidence.

2. The absence of these variants in large population databases of healthy controls and furthermore the absence of variants that predict similar functional consequences bring orthogonal levels of evidence to this conclusion.

3. A mouse model enables the testing of a single variant on defined in singular background. Failure to replicate the same or a similar phenotype using this approach has plenty of precedence relating most likely to interspecies differences and certainly would not trump what I consider to be the unassailable human genetic evidence brought by the description of 2 alleles, all de novo, on 5 different “genetic backgrounds” delivering a highly concordant, extremely rare phenotype. In conclusion I think that the evidence that the two alleles described in this paper cause BPTAS is unassailable. The recurrency of one of these alleles is consistent with a mechanism that may only be induced by a very restricted range of variants at the HMGB1 locus. In this respect I disagree with Reviewer 2 in that just because only 2 alleles have been described, this speaks against the claim for the proposed pathogenic mechanism because a multiplicity of alternatives must exist if the phase-change hypothesis is supported. A large range of alternatives have been tested by these authors using in vitro mutagenesis and survey of standing variation in humans at this locus using large databases. A reasonable conclusion is that a restricted range of mutational alternatives are only available to convert the tail of HMGB1 from acidic to basic (and whatever other qualities are required such as length). The reviewer suggests that this is possible in many other ways but the presented bioinformatic and experimental evidence argues that such alternatives are not apparent. A study of a single mouse allele would not advance this matter any further.

In summary, I do not share Reviewer 2's concerns and insistence that a mouse model is required for the claims in this paper to be supported.

References

- 1 Gueiderikh, A. *et al.* Fanconi anemia A protein participates in nucleolar homeostasis maintenance and ribosome biogenesis. *Sci Adv* **7**, doi:10.1126/sciadv.abb5414 (2021).
- 2 Sondalle, S. B., Longerich, S., Ogawa, L. M., Sung, P. & Baserga, S. J. Fanconi anemia protein FANCI functions in ribosome biogenesis. *Proc Natl Acad Sci U S A* **116**, 2561-2570, doi:10.1073/pnas.1811557116 (2019).
- 3 Calo, E. *et al.* Tissue-selective effects of nucleolar stress and rDNA damage in developmental disorders. *Nature* **554**, 112-117, doi:10.1038/nature25449 (2018).
- 4 Trainor, P. A. & Merrill, A. E. Ribosome biogenesis in skeletal development and the pathogenesis of skeletal disorders. *Biochim Biophys Acta* **1842**, 769-778, doi:10.1016/j.bbadis.2013.11.010 (2014).
- 5 Alazami, A. M. *et al.* Mutations in C2orf37, encoding a nucleolar protein, cause hypogonadism, alopecia, diabetes mellitus, mental retardation, and extrapyramidal syndrome. *Am J Hum Genet* **83**, 684-691, doi:10.1016/j.ajhg.2008.10.018 (2008).
- 6 Ali, R. *et al.* Expanding on the phenotypic spectrum of Woodhouse-Sakati syndrome due to founder pathogenic variant in DCAF17: Report of 58 additional patients from Qatar and literature review. *Am J Med Genet A* **188**, 116-129, doi:10.1002/ajmg.a.62501 (2022).
- 7 Uguen, K. *et al.* Heterozygous HMGB1 loss-of-function variants are associated with developmental delay and microcephaly. *Clin Genet* **100**, 386-395, doi:10.1111/cge.14015 (2021).
- 8 Aviner, R. The science of puromycin: From studies of ribosome function to applications in biotechnology. *Comput Struct Biotechnol J* **18**, 1074-1083, doi:10.1016/j.csbj.2020.04.014 (2020).

Reviewer Reports on the Second Revision:

Referees' comments:

Referee #1 (Remarks to the Author):

The authors answered my concerns in a satisfactory manner. Thank you!

Referee #2 (Remarks to the Author):

The authors have submitted a revised version of their manuscript, and a detailed rebuttal.

As I previously indicated, I am convinced that (i) the two described frameshift mutations in HMGB1 in 5 different patients cause BPTAS, and that (ii) the replacement of the intrinsically disordered acidic tail of HMGB1 with a basic stretch causes mispartitioning of HMGB1 in the nucleolus and formation of condensates. The authors also provide at least partial evidence that (iii) other mutant proteins where a C-terminal Intrinsically Disordered Region is replaced by a Basic Stretch (IDR-BS) do mispartition/condensate in the nucleolus.

However, I am deeply skeptical that the facts (i) and (ii), which are well described in the manuscript, are causally related. The fact that an IDR-BS mutant protein mispartitions/condensates in the nucleolus does not mean that all IDR-BS mutants will cause a phenotype similar to that of BPTAS. In a stronger version of my opinion, the phenotype of BPTAS may not be caused by nucleolar mispartitioning/condensation at all, but by another feature associated to the HMGB1 IDR-BS mutant. In particular, HMGB1 is described to interact with a large number of proteins, many of them transcription factors. Any decrease in the concentration, the subnuclear localization or the chromatin dynamics of HMGB1 might affect its interaction with one or more of these proteins and the interactions of these proteins with chromatin, and as a consequence affect developmental decisions. As an example, the phenotype of BPTAS involves musculo-skeletal defects in the limbs, which might be connected to interaction of HMGB1 with HOX proteins, or modification of the chromatin accessibility of the HOX loci.

To prove that BPTAS is caused by nucleolar mispartitioning/condensation of proteins, the authors should show that one more (not just HMGB1) of the IDR-BS mutant proteins that cause nucleolar mispartitioning/condensation also cause a BPTAS- phenotype.

I previously suggested that the authors should express the mutant form of HMGB1 in mice and show that it causes a BPTAS-like phenotype. In fact, the other reviewers noted, this experiment might not be conclusive: even if did not cause a BPTAS-like phenotype, the result might be dictated by unspecified differences between human and mouse physiology and development. Moreover, following my argument in the previous paragraph, only expressing a SECOND IDR-BS mutant protein and showing that it also causes a similar phenotype to the HMGB1 IDR-BS mutant would nail the argument. I concede that this is undoable. Thus, I now think that the right test of the pudding is finding in the human genetics literature a second IDR-BS protein, and showing that it causes a BPTAS-like phenotype. The authors mention in the rebuttal that "Some of the [nucleolar] diseases have partially overlapping symptoms with BPTAS, which we described in detail in the Supplementary

Notes as per a previous request by Reviewer 3." I did not find these Supplementary Notes. However, such similarities would support the authors' argument, and as should be described in the main text.

I wish to point out that I am not arguing that the results of this manuscript are not interesting, I simply do not share the authors' conviction that they are causally connected, see lines 295-297 of discussion: "Our data identify the replacement of the disordered tail with an arginine-rich basic tail in HMGB1 as the pathomechanism underlying brachyphalangy-polydactyly-tibial aplasia syndrome (BPTAS)".

On the contrary, I partially subscribe to the statement in lines 287-289: "These results indicate that disease-associated frameshifts that generate an arginine-rich basic tail in C-terminal IDRs cause nucleolar mispartitioning and dysfunction." I would delete "disease-associated" in the phrase, because there is no proof that EVERY mutant that causes nucleolar mispartitioning and dysfunction would be pathogenic at the clinical level.

Other replies to the authors' rebuttal:

1. I do accept that reduction (but not abrogation) of nucleolar function is compatible with life. The authors should clarify what they mean by "nucleolar arrest" (repeated several times) or "HMGB1 poisons nucleoli". These words are rather dramatic and led me to think that the nucleoli do not work at all. Sorry for my misunderstanding.

2. I concede that the mouse model is outside the scope of this work (see above).

3. I do believe (and I made it clear in the previous review rounds) that the mutations described are the physical cause of BPTAS. On the other hand, I do find it statistically very improbable that 4 out of 5 independent de novo mutations involve the SAME 4 bp deletion. I did not argue that this compromises the conclusion that the IDR-BS mutation causes BPTAS (especially because there is a second, different mutation), but rather argued that the mechanism that gives origin to that 4 bp deletion 4 different times must be very specific and cannot be chance alone. Maybe the authors could elaborate on that.

4. The authors accept in the rebuttal my concerns on the likelihood of the formation of an alpha-helix from the basic stretch in the mutant, but have changed the main text in a very minor way. The impression that I get as a reader is that the formation of the alpha-helix is important. In my opinion, it is unlikely and unimportant: the basic stretch can have exactly the same pathogenetic features without necessarily forming an alpha-helix.

I also note that in panel 2a the second HMG box and the IDR overlap, while there is contrary evidence that the second HMG box is well structured and not an IDR (I pointed this out in a previous round of revision).

5. I am not convinced by the additional evidence provided on the purity of wt and mut HMGB1-GFP. Nonetheless, even if I would not bet on the reliability of the in vitro data, the data in cells (Fig. 3) show that mutHMGB1 precipitates. The precipitation, not the putative liquid liquid phase separation, appears to be important. That is a big change from the previous version.

6. I thank the authors for clarifying terminology. However, in Discussion the authors continue to argue that mispartition to the nucleolus occurs by phase separation, which in my opinion might not be the case: mispartition likely occurs through precipitation of mutHMGB1 around the nucleoli or parts of them. Mispartition appears to be due to the LACK of correct phase separation.

7. I thank the authors for the clarification on the puromycin incorporation experiment. I had misinterpreted the figure.

Referee #3 (Remarks to the Author):

The authors have responded to my questions well, including the statistical treatment of the catalog analysis. The exclusive reliance upon ClinVar to assign pathogenicity escaped me on my first reading and their approach is more clear to me now. I have one residual question for them (and it is not necessarily a criticism but might suggest an approach that could further bolster their claims). I did mention it in my first review.

Assignment of pathogenicity by a submitter to ClinVar would more likely occur if there is phenotypic congruence between the effects of other alleles at these same loci and the variant(s) under evaluation (in this instance the 3' frameshifts). In contrast it is clear that these two categories of variants give very different phenotypic effects for HMGB1. Because ClinVar is largely devoid of phenotypic data it is not possible to test the prediction that these IDR frameshifts operate by a starkly different mechanism resulting in an explanation for some of the phenotypic heterogeneity associated with variation at any of these loci. This heterogeneity may not have been fully appreciated to date because of a lessening of the confidence that pathogenicity will be pronounced for frameshifting variants within the IDR.

It is predicted that these IDR frameshifts should give phenotypes that are distinctive and usually not associated with what null alleles usually produce at these same loci. Could the authors perhaps widen their reliance from ClinVar to other databases that have phenotypic data such as Decipher to see if, for at least some of these loci, this prediction is borne out? I realise that there may not be enough data in Decipher to be definitive but testing a subset of genes with frameshifting variants at well described disease loci could be instructive.

One minor typo I noted:

p.6 remove the word "when" from "Nuclear inclusions were observed in several other human cell types expressing when the mutant HMGB1 (HEK293T, HCT116 and MCF7 cells) (Extended Data Fig. 5d)."

Author Rebuttals to Second Revision:

Referees' comments:

Referee #1 (Remarks to the Author):

The authors answered my concerns in a satisfactory manner.

Referee #2 (Remarks to the Author):

The authors have submitted a revised version of their manuscript, and a detailed rebuttal.

As I previously indicated, I am convinced that (i) the two described frameshift mutations in HMGB1 in 5 different patients cause BPTAS, and that (ii) the replacement of the intrinsically disordered acidic tail of HMGB1 with a basic stretch causes mispartitioning of HMGB1 in the nucleolus and formation of condensates. The authors also provide at least partial evidence that (iii) other mutant proteins where a C-terminal Intrinsically Disordered Region is replaced by a Basic Stretch (IDR-BS) do mispartition/condensate in the nucleolus.

However, I am deeply skeptical that the facts (i) and (ii), which are well described in the manuscript, are causally related. The fact that an IDR-BS mutant protein mispartitions/condensates in the nucleolus does not mean that all IDR-BS mutants will cause a phenotype similar to that of BPTAS. In a stronger version of my opinion, the phenotype of BPTAS may not be caused by nucleolar mispartitioning/condensation at all, but by another feature associated to the HMGB1 IDR-BS mutant. In particular, HMGB1 is described to interact with a large number of proteins, many of them transcription factors. Any decrease in the concentration, the subnuclear localization or the chromatin dynamics of HMGB1 might affect its interaction with one or more of these proteins and the interactions of these proteins with chromatin, and as a consequence affect developmental decisions. As an example, the phenotype of BPTAS involves musculo-skeletal defects in the limbs, which might be connected to interaction of HMGB1 with HOX proteins, or modification of the chromatin accessibility of the HOX loci.

To prove that BPTAS is caused by nucleolar mispartitioning/condensation of proteins, the authors should show that one more (not just HMGB1) of the IDR-BS mutant proteins that cause nucleolar mispartitioning/condensation also cause a BPTAS- phenotype.

I previously suggested that the authors should express the mutant form of HMGB1 in mice and show that it causes a BPTAS-like phenotype. In fact, the other reviewers noted, this experiment might not be conclusive: even if did not cause a BPTAS-like phenotype, the result might be dictated by unspecified differences between human and mouse physiology and development. Moreover, following my argument in the previous paragraph, only expressing a SECOND IDR-BS mutant protein and showing that it also causes a similar phenotype to the HMGB1 IDR-BS mutant would nail the argument. I concede that this is undoable. Thus, I now think that the right test of the pudding is finding in the human genetics literature a second IDR-BS protein, and showing that it causes a BPTAS-like phenotype. The authors mention in the rebuttal that "Some

of the [nucleolar] diseases have partially overlapping symptoms with BPTAS, which we described in detail in the Supplementary Notes as per a previous request by Reviewer 3." I did not find these Supplementary Notes. However, such similarities would support the authors' argument, and as should be described in the main text.

I wish to point out that I am not arguing that the results of this manuscript are not interesting, I simply do not share the authors' conviction that they are causally connected, see lines 295-297 of discussion: "Our data identify the replacement of the disordered tail with an arginine-rich basic tail in HMGB1 as the pathomechanism underlying brachyphalangy-polydactyly-tibial aplasia syndrome (BPTAS)". On the contrary, I partially subscribe to the statement in lines 287-289: "These results indicate that disease-associated frameshifts that generate an arginine-rich basic tail in C-terminal IDRs cause nucleolar mispartitioning and dysfunction." I would delete "disease-associated" in the phrase, because there is no proof that EVERY mutant that causes nucleolar mispartitioning and dysfunction would be pathogenic at the clinical level.

The data we present suggest that frameshift variants in IDRs that create arginine-rich basic sequences (IDR-BS) can lead to mispartitioning of the mutant protein into the nucleolus, and lead to nucleolar dysfunction. We do not expect that an IDR-BS mutation in every protein will lead to the same disease for at least two important reasons.

1. Genes have different tissue-specific expression patterns. Therefore, an IDR-BS variant is expected to have an effect in the cell types where the gene is expressed. HMGB1 is broadly expressed in virtually all cell types at a moderately high level, consistent with its IDR-BS mutation affecting multiple organ systems (BPTAS). In contrast, a closely related gene HMGB3 contains a similar IDR-BS mutation, but is expressed at relatively low level in a subset of tissues, and the variant is associated with microphthalmia and not BPTAS. Another example is FOXL2, whose IDR-BS mutation is associated with a rare disease with dysmorphic craniofacial features and ovarian insufficiency – where *FOXL2* is expressed (see Reviewer Figure 1 below).
2. Some genes are haplosufficient, and some genes are haploinsufficient in humans. HMGB1 for example has a pLI score of 0.82 (i.e. it is haploinsufficient). Therefore, BPTAS is caused – as argued by the reviewer – likely by a combination of the loss of one wild type allele, and the nucleolar dysfunction caused by the IDR-BS variant. As a corollary example, 16 genes affected by an IDR-BS variant in our catalog have a pLI<0.05 and are associated with autosomal dominant inheritance.

Furthermore, not every cell type is sensitive to the same degree to disruption of nucleolar function. For these reasons, similar IDR-BS mutations in different genes are not to be expected to cause the same disease. Nevertheless, we note that several IDR-BS variants are associated with microphthalmia (HMGB3, RAX, SOX2) suggesting some level of phenotypic consistency. We have made these points clear in the revised discussion section (highlighted in blue), and added analyses of pLI scores in Extended Data Fig. 7e.

Moreover, we have identified an IDR-BS variant in the *DVL1* gene, which is associated with a complex syndrome similar to BPTAS. IDR-BS mutations in *DVL1* cause Robinow syndrome (OMIM: 616331) characterized by skeletal, craniofacial and genitourinary anomalies. These organ systems are also affected in BPTAS. Mutations in *DVL1* show autosomal dominant inheritance, and were predicted to have an unknown gain-of-function effect (Bunn et al., 2015; White et al., 2015). We find that the variants create arginine-rich basic sequences with a hydrophobic patch (similar to the BPTAS *HMGB1* variant) (Extended Data Fig. 8). We include new data in the revised manuscript showing that the frameshift mutant *DVL1* mispartitions into the nucleolus in human pluripotent cells. In these cells, the Wnt pathway, that *DVL1* is a component of, is active (Extended Data Fig. 9f-i). As a further support for the importance of tissue-specificity of expression, *DVL1* shows a similarly broad expression pattern as *HMGB1* (see Reviewer Figure 1). We include the description of the *DVL1* variant and phenotype in the revised Supplementary Notes, and new Supplementary Figure 5.

We included information on the genotype-phenotype correlation of *HMGB1* variants, and several other IDR-BS variants in the Supplementary Notes. The reviewer noted a difficulty in locating the supplementary files, which might have been caused by issues of the online system. For the Reviewer's convenience, we include a link to an electronic version of the Supplementary Information here: <https://tinyurl.com/4uhvv6wb>.

Finally, we added the word "can" to the sentence in lines 287-289 commented on by the Reviewer, to read "disease-associated frameshift that generate arginine-rich basic tail in C-terminal IDRs can cause nucleolar mispartitioning and dysfunction."

We thank the reviewer for the thorough and useful insights and comments.

Reviewer Figure 1. Tissue-level expression data for *HMGB1*, *HMB3*, *FOXL2* and *DVL1* from GTEX (<https://gtexportal.org/home/>).

Other replies to the authors' rebuttal:

1. I do accept that reduction (but not abrogation) of nucleolar function is compatible with life. The authors should clarify what they mean by "nucleolar arrest" (repeated several times) or "HMGB1 poisons nucleoli". These words are rather dramatic and led me to think that the nucleoli do not work at all. Sorry for my misunderstanding.

We introduce the term arrest at the FRAP experiments of nucleolus (Fig. 3d), and use the term to refer to the lack of molecule turnover in the nucleolus revealed by FRAP. To reduce confusion, we changed the "HMGB1 poisons nucleoli" header to "Mutant HMGB1 causes nucleolar dysfunction" in the revised manuscript (highlighted in blue).

2. I concede that the mouse model is outside the scope of this work (see above).

Thank you! We are working on developing cellular and organismal models.

3. I do believe (and I made it clear in the previous review rounds) that the mutations described are the physical cause of BPTAS. On the other hand, I do find it statistically very improbable that 4 out of 5 independent de novo mutations involve the SAME 4 bp deletion. I did not argue that this compromises the conclusion that the IDR-BS mutation causes BPTAS (especially because there is a second, different mutation), but rather argued that the mechanism that gives origin to that 4 bp deletion 4 different times must be very specific and cannot be chance alone. Maybe the authors could elaborate on that.

The C-terminal tail of HMGB1 consists of a repeat of glutamates and aspartates, and is encoded by a sequence similar to a (CTT)_n-simple repeat on the DNA level. Such repeats are known to be error prone during DNA replication. A 4bp deletion could e.g. result from NHEJ which is a known mechanism in the repair of DNA-double strand breaks frequently observed in trinucleotide repeats (Khristich and Mirkin, 2020). We agree that investigation of the sequence features of DNA could be interesting.

4. The authors accept in the rebuttal my concerns on the likelihood of the formation of an alpha-helix from the basic stretch in the mutant, but have changed the main text in a very minor way. The impression that I get as a reader is that the formation of the alpha-helix is important. In my opinion, it is unlikely and unimportant: the basic stretch can have exactly the same pathogenetic features without necessarily forming an alpha-helix. I also note that in panel 2a the second HMG box and the IDR overlap, while there is contrary evidence that the second HMG box is well structured and not an IDR (I pointed this out in a previous round of revision).

We added a sentence to the discussion: "To what extent the minimal propensity of the mutant sequence to form a helix contributes to these effects remains to be tested." We have previously

addressed the concern on the IDR annotation: we show that an IDR that lacks the portion overlapping the HMG Box behaves the same way in cells as the ‘long’ IDR (Extended Data Fig. 5c).

5. I am not convinced by the additional evidence provided on the purity of wt and mut HMGB1-GFP. Nonetheless, even if I would not bet on the reliability of the in vitro data, the data in cells (Fig. 3) show that mutHMGB1 precipitates. The precipitation, not the putative liquid liquid phase separation, appears to be important. That is a big change from the previous version.

We have clarified these issues in the revised text, and in the discussion section in particular. In brief, wild type HMGB1 forms droplets that display features of LLPS, including fusion of droplets (Supplementary video 1). The mutant HMGB1 indeed appears to precipitate in the presence of crowding agent or RNA, supporting our conclusion that the mutant alters the phase separation capacity of the protein. In addition, the mutant HMGB1 mispartitions into the nucleolus in cells, and the FRAP experiments revealed that such nucleoli are characterized by negligible molecule exchange (Fig 3d, 3g), resembling “precipitates”. The mutant protein thus appears to alter the condensate properties of the nucleolus, which typically behaves as multi-layered, phase-separated liquid.

6. I thank the authors for clarifying terminology. However, in Discussion the authors continue to argue that mispartition to the nucleolus occurs by phase separation, which in my opinion might not be the case: mispartition likely occurs through precipitation of mutHMGB1 around the nucleoli or parts of them. Mispartition appears to be due to the LACK of correct phase separation.

We have clarified this in the revised discussion, and removed the clause in question. In brief, as also discussed at point 5, the mutation in HMGB1 indeed interferes with the phase separation of HMGB1, and also disrupts phase separation of the nucleolus (where the protein mispartitions).

7. I thank the authors for the clarification on the puromycin incorporation experiment. I had misinterpreted the figure.

We thank the reviewer for pointing us to previously unclear presentation of the data!

Referee #3 (Remarks to the Author):

The authors have responded to my questions well, including the statistical treatment of the catalog analysis. The exclusive reliance upon ClinVar to assign pathogenicity escaped me on my first reading and their approach is more clear to me now. I have one residual question for them (and it is not necessarily a criticism but might suggest an approach that could further bolster their

claims). I did mention it in my first review. Assignment of pathogenicity by a submitter to ClinVar would more likely occur if there is phenotypic congruence between the effects of other alleles at these same loci and the variant(s) under evaluation (in this instance the 3' frameshifts). In contrast it is clear that these two categories of variants give very different phenotypic effects for HMGB1. Because ClinVar is largely devoid of phenotypic data it is not possible to test the prediction that these IDR frameshifts operate by a starkly different mechanism resulting in an explanation for some of the phenotypic heterogeneity associated with variation at any of these loci. This heterogeneity may not have been fully appreciated to date because of a lessening of the confidence that pathogenicity will be pronounced for frameshifting variants within the IDR. It is predicted that these IDR frameshifts should give phenotypes that are distinctive and usually not associated with what null alleles usually produce at these same loci. Could the authors perhaps widen their reliance from ClinVar to other databases that have phenotypic data such as Decipher to see if, for at least some of these loci, this prediction is borne out? I realise that there may not be enough data in Decipher to be definitive but testing a subset of genes with frameshifting variants at well described disease loci could be instructive.

We thank the Reviewer for these insightful comments. We agree with the Reviewer's insight that the ClinVar annotation (or lack thereof) could bias against phenotypic incongruence of different variants that occur within the same gene.

Because phenotypic information is described in detail in OMIM, we queried OMIM (not DECIPHER). We found that 10 genes featuring pathogenic arginine rich frameshifts are associated with more than one syndrome. 15 genes with known pathogenic frameshift mutations adding hydrophobic patches to the amino acid sequence are associated with more than one disorder. ELN and TP53 are examples of genes where specifically C-terminal frameshifts result in a disorder distinct from that caused by other mutations of the gene including entire deletions. We added this information in the Supplementary Notes in the revised manuscript.

We also include further characterization of genes in which pathogenic arginine-rich frameshift variants occur. We included analyses of pLI scores in Extended Data Fig. 7e. We found 16 genes that contain a pathogenic arginine-rich frameshift variant, that have pLI < 0.05 and the gene is associated with autosomal dominant inheritance. These results argue that the effect of the variants likely cannot be explained by a loss-of-function. We note that the individual variants are rare, and as the reviewer noted, the clinical data are limited for more extensive genotype-phenotype correlation analysis.

Also, as described in detail above to the response to the general comment of Reviewer 2, we found an additional variant in DVL1, for which there is genetic evidence that the variant has a gain-of-function effect (Bunn et al., 2015; White et al., 2015).

We expanded the genotype-phenotype correlation analyses with *DVLI* and other genes in the Supplementary Notes. Reviewer 2 noted a difficulty in locating the supplementary files, which might have been caused by the online system. For the Reviewers' convenience, we include a link to an electronic version of the Supplementary Information here: <https://tinyurl.com/4uhvv6wb>.

One minor typo I noted:

p.6 remove the word "when" from "Nuclear inclusions were observed in several other human cell types expressing when the mutant HMGB1 (HEK293T, HCT116 and MCF7 cells) (Extended Data Fig. 5d).

Done.

References

Bunn, K.J., Daniel, P., Rosken, H.S., O'Neill, A.C., Cameron-Christie, S.R., Morgan, T., Brunner, H.G., Lai, A., Kunst, H.P., Markie, D.M., *et al.* (2015). Mutations in DVL1 cause an osteosclerotic form of Robinow syndrome. *Am J Hum Genet* 96, 623-630.

Khristich, A.N., and Mirkin, S.M. (2020). On the wrong DNA track: Molecular mechanisms of repeat-mediated genome instability. *J Biol Chem* 295, 4134-4170.

White, J., Mazzeu, J.F., Hoischen, A., Jhangiani, S.N., Gambin, T., Alcino, M.C., Penney, S., Saraiva, J.M., Hove, H., Skovby, F., *et al.* (2015). DVL1 frameshift mutations clustering in the penultimate exon cause autosomal-dominant Robinow syndrome. *Am J Hum Genet* 96, 612-622.

Reviewer Reports on the Third Revision:

Referees' comments:

Referee #2 (Remarks to the Author):

This manuscript shows that:

- frameshift mutations in HMGB1 that turn a disordered acidic tail into a basic tail are the cause of BPTAS, a rare syndrome
- wt HMGB1 can phase separate into liquid droplets in vivo, and associate with droplets of other proteins that phase separate; the frameshift HMGB1 mutant associates with droplets of other phase separating proteins at lower concentrations
- in cells, the frameshift HMGB1 mutant associates with nucleoli forming an outer shell that encapsulates them; the frameshift HMGB1 mutant exchanges slowly from the altered nucleoli and disrupts rRNA biosynthesis
- mispartitioning of frameshift HMGB1 mutant depends from the presence of Rs in the basic tail, and the reduction in exchange from nucleoli from the presence of a basic patch at the C terminus of the basic tail
- mining and integration of several databases shows that many proteins that contain an IDR may be subject to mutations that create a basic tail similar to that in the HMGB1 frameshift mutant. Thirteen of these frameshift proteins were expressed in cells and shown to mispartition. Individually, each of these points is well supported by data. Their integration in the complete story, though, still leads to overstatements and overinterpretations, despite several rounds of revision that progressively improved the manuscript.

In this revision, there is a clear improvement: the authors acknowledge that their proposed nucleolar grammar is subject to many variations in context. In particular, the HMGB1 frameshift mutation likely causes both haploinsufficiency and the gain-of-function in nucleolar mispartitioning. This point is now discussed well.

Other points are still weak. In particular:

- what is the role of IDRs? The whole flow of the story focuses on IDRs, but a basic tail could form due a frameshift in a gene region coding for a structured domain.
- what is the role of phase separation? The basic tail that arises from the frameshift in HMGB1 alters the liquid liquid phase separation properties of the wt HMGB1, but also causes the association of mutant HMGB1 as an amorphous layer on the outer part of the nucleolus. Is the alteration of the liquid liquid phase separation the cause of nucleolar dysfunction or the amorphous association with the nucleolus? The authors dismiss the issue by calling both the liquid liquid phase separation and the amorphous encapsulation of the nucleolus as "phase separation". In fact, liquid-solid phase separations and liquid-liquid phase separations are both phase separations, but deeply different ones. In my opinion, the liquid liquid phase separation plays a minor role here. This could be tested: appending a basic tail to a fully structured globular protein that resides in the nucleus (and does not undergo liquid liquid phase separations) might cause the same nucleolar disruption described here.
- what is an arrested nucleolus? As the authors define it, an arrested nucleolus is a nucleolus where an outer amorphous layer of mispartitioned mutant protein does not exchange fast with the liquid

phase. This is not informative, it is just the description of a property of the mispartitioned protein; it doesn't say much about the condition of the nucleolus. Depending on the nature of the mispartitioned protein and on its amount, the nucleolus could still function to an extent that causes little damage to the cell, or it could be severely damaged in one or more of its functions.

Referee #3 (Remarks to the Author):

The authors have addressed my previous comments well. I have just a couple of semantic comments to make about the new additions to the discussion:

1. The use of the word symptoms in the phrase "consistent with symptoms presenting in multiple organ systems" is incorrect. Symptoms are reported by the patient, signs are observed by the physician. An alternative phrase could be "phenotypic features" or "phenotypic components"
2. I am uncertain that the phrase "sharing underlying molecular principles" is the best terminology (the use of the word principles). Perhaps altering the sentence to "thus occur in a wide spectrum of genetic diseases with a shared underlying molecular pathogenic mechanism".

Congratulations to the authors for an excellent body of work.

Stephen Robertson

Author Rebuttals to Third Revision:

Referees' comments:

Referee #2 (Remarks to the Author):

This manuscript shows that:

- frameshift mutations in HMGB1 that turn a disordered acidic tail into a basic tail are the cause of BPTAS, a rare syndrome
- wt HMGB1 can phase separate into liquid droplets in vivo, and associate with droplets of other proteins that phase separate; the frameshift HMGB1 mutant associates with droplets of other phase separating proteins at lower concentrations
- in cells, the frameshift HMGB1 mutant associates with nucleoli forming an outer shell that encapsulates them; the frameshift HMGB1 mutant exchanges slowly from the altered nucleoli and disrupts rRNA biosynthesis
- mispartitioning of frameshift HMGB1 mutant depends from the presence of Rs in the basic tail, and the reduction in exchange from nucleoli from the presence of a basic patch at the C terminus of the basic tail
- mining and integration of several databases shows that many proteins that contain an IDR may be subject to mutations that create a basic tail similar to that in the HMGB1 frameshift mutant. Thirteen of these frameshift proteins were expressed in cells and shown to mispartition. Individually, each of these points is well supported by data. Their integration in the complete story, though, still leads to overstatements and overinterpretations, despite several rounds of revision that progressively improved the manuscript.

In this revision, there is a clear improvement: the authors acknowledge that their proposed nucleolar grammar is subject to many variations in context. In particular, the HMGB1 frameshift mutation likely causes both haploinsufficiency and the gain-of-function in nucleolar mispartitioning. This point is now discussed well.

We are grateful for the guidance, valuable comments and suggestions throughout the entire review process.

Other points are still weak. In particular:

- what is the role of IDRs? The whole flow of the story focuses on IDRs, but a basic tail could form due a frameshift in a gene region coding for a structured domain.

It is indeed plausible that a basic tail could form due to a frameshift in a structured domain. For HMGB1 however, the structured HMG boxes seem to partially contribute to the nucleolar mispartitioning (Figure 3a), suggesting that structured domains in the protein may contribute. The focus is on IDRs, because i) IDRs are known to be involved in phase separation, ii) we wanted to identify variants that are pathogenic not because of a loss-of-function effect (replacement of a structured domain is likely to lead to loss of function), and iii) the HMGB1 structured portion partially contributes to mislocalization.

- what is the role of phase separation? The basic tail that arises from the frameshift in HMGB1 alters the liquid liquid phase separation properties of the wt HMGB1, but also causes the association of mutant

HMGB1 as an amorphous layer on the outer part of the nucleolus. Is the alteration of the liquid liquid phase separation the cause of nucleolar dysfunction or the amorphous association with the nucleolus? The authors dismiss the issue by calling both the liquid liquid phase separation and the amorphous encapsulation of the nucleolus as "phase separation". In fact, liquid-solid phase separations and liquid-liquid phase separations are both phase separations, but deeply different ones. In my opinion, the liquid liquid phase separation plays a minor role here. This could be tested: appending a basic tail to a fully structured globular protein that resides in the nucleus (and does not undergo liquid liquid phase separations) might cause the same nucleolar disruption described here.

We discuss this issue at several places in the manuscript. We agree with the reviewer, that our data suggest that the effect of the mutation is two-fold: 1) it alters the phase separation capacity of HMGB1, and 2) it leads to mispartitioning of the mutant protein into the nucleolus where it interferes with nucleolar function. The function of the nucleolus is well understood to depend on the condensate features of the nucleolus (see reviewed e.g. in Lafontaine et al., Nat Rev Cell Bio 2021). The exact biophysical processes underlying the two effects (LLPS or LSPS?), are to be tested in future work. To further clarify the issue, we refer to the two effects now separately in the title.

- what is an arrested nucleolus? As the authors define it, an arrested nucleolus is a nucleolus where an outer amorphous layer of mispartitioned mutant protein does not exchange fast with the liquid phase. This is not informative, it is just the description of a property of the mispartitioned protein; it doesn't say much about the condition of the nucleolus. Depending on the nature of the mispartitioned protein and on its amount, the nucleolus could still function to an extent that causes little damage to the cell, or it could be severely damaged in one or more of its functions.

We indeed use the term "arrested" to describe a nucleolus "where an outer amorphous layer of mispartitioned mutant protein does not exchange fast with the liquid phase". We also show that the inhibited exchange is associated with altered rRNA biogenesis (Extended Data Fig. 6a), and global reduction of protein translation (Figure 3h-i, Extended Data Fig. 6a-b), suggesting nucleolar dysfunction.

Referee #3 (Remarks to the Author):

The authors have addressed my previous comments well. I have just a couple of semantic comments to make about the new additions to the discussion:

We are grateful for the guidance, valuable comments and suggestions throughout the entire review process.

1. The use of the word symptoms in the phrase "consistent with symptoms presenting in multiple organ systems" is incorrect. Symptoms are reported by the patient, signs are observed by the physician. An alternative phrase could be "phenotypic features" or "phenotypic components"

We changed the word symptom to "phenotypic features" and "features".

2. I am uncertain that the phrase "sharing underlying molecular principles" is the best terminology (the use of the word principles). Perhaps altering the sentence to "thus occur in a wide spectrum of genetic

diseases with a shared underlying molecular pathogenic mechanism".
Congratulations to the authors for an excellent body of work.

We changed the sentence to "thus occur in a wide spectrum of genetic diseases as a shared underlying molecular pathomechanism", as suggested.

Stephen Robertson